# ZEPHYRUS: AN AGENTIC FRAMEWORK FOR WEATHER SCIENCE

**Sumanth Varambally,  Marshall Fisher,  Jas Thakker,  Yiwei Chen,  Zhirui Xia,**
**Yasaman Jafari,  Ruijia Niu,  Manas Jain,  Veeramakali Vignesh Manivannan**
**Zachary Novack,  Luyu Han,  Srikar Eranky,  Salva Rühling Cachay**
**Taylor Berg-Kirkpatrick,  Duncan Watson-Parris,  Yi-An Ma,  Rose Yu** [*]
UC San Diego

## ABSTRACT

Foundation models for weather science are pre-trained on vast amounts of structured numerical data and outperform traditional weather forecasting systems. However, these models lack language-based reasoning capabilities, limiting their utility in interactive scientific workflows. Large language models (LLMs) excel at understanding and generating text but cannot reason about high-dimensional meteorological datasets. We bridge this gap by building the first agentic framework for weather science. Our framework includes a Python code-based environment for agents (ZEPHYRUSWORLD) to interact with weather data, featuring tools including a WeatherBench 2 dataset indexer, geolocator for geocoding from natural language, weather forecasting, climate simulation capabilities, and a climatology module for querying precomputed climatological statistics (e.g., means, extremes, and quantiles) across multiple timescales. We design ZEPHYRUS, a multi-turn LLM-based weather agent that iteratively analyzes weather datasets, observes results, and refines its approach through conversational feedback loops. We accompany the agent with a new benchmark, ZEPHYRUSBENCH, with a scalable data generation pipeline that constructs diverse question-answer pairs across weather-related tasks, from basic lookups to advanced forecasting, extreme event detection, and counterfactual reasoning. Experiments on this benchmark demonstrate the strong performance of ZEPHYRUS agents over text-only baselines, outperforming them by up to 44 percentage points in correctness. However, the hard tasks are still difficult even with frontier LLMs, highlighting the challenging nature of our benchmark and suggesting room for future development. Our codebase and benchmark are available at `https://github.com/Rose-STL-Lab/Zephyrus`.

## 1  INTRODUCTION

Large language models (LLMs) have demonstrated remarkable capabilities across diverse scientific domains (Birhane et al., 2023), revolutionizing fields from drug discovery (Zheng et al., 2024; Wu et al., 2024b) and materials science (Lei et al., 2024; Jablonka et al., 2023) to network biology (Theodoris et al., 2023). These models excel at processing textual content such as scientific literature, source code (Jiang et al., 2024), and structured data tables (Zhang et al., 2024). However, their application to domains requiring reasoning over high-dimensional numerical data remains limited (Wang et al., 2024).

Meteorology offers a compelling yet challenging case study, as combining natural language reasoning with complex atmospheric data has the potential to greatly advance weather research. Weather prediction is a critical scientific challenge, with profound implications spanning agriculture, disaster preparedness, transportation, and energy management (Alley et al., 2019). The field has witnessed remarkable progress through machine learning approaches, with foundation models (Nguyen et al., 2023; Kurth et al., 2023; Lam et al., 2023; Bi et al., 2023; Nguyen et al., 2024) now achieving state-of-the-art performance in medium-range forecasting, often surpassing traditional physics-based numerical simulations (Molteni et al., 1996; Bauer et al., 2015). However, current weather models

---

[*]Correspondence: `svarambally@ucsd.edu`, `roseyu@ucsd.edu`

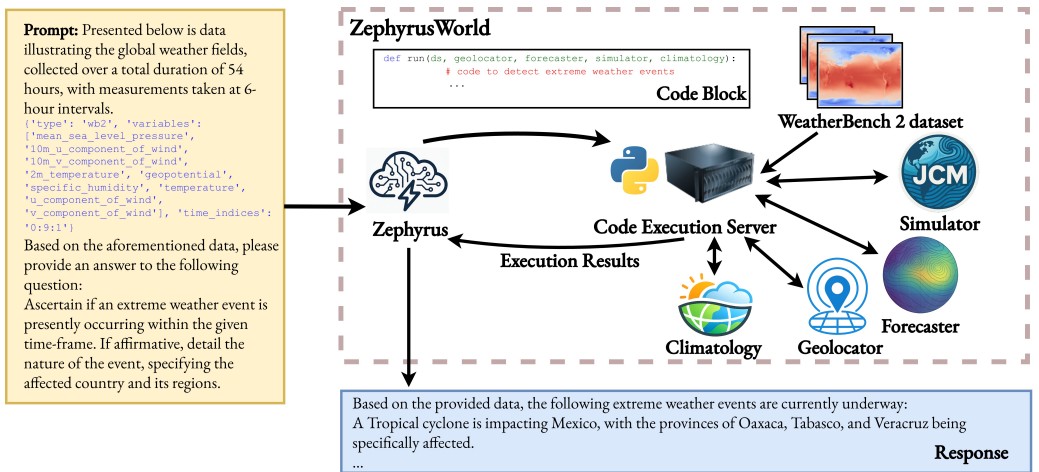

Figure 1: **Overview**: We develop ZEPHYRUS, an agentic framework for weather science. Given a query, the LLM-based agent ZEPHYRUS writes a code block which is sent to the code execution server. The server orchestrates several tools to execute the code block and returns the execution results to the agent. The agent either decides to execute more code to refine its output or respond back to the user. Refer to Appendix A.3 for the full prompt.

operate exclusively on structured numerical datasets such as reanalysis data, cannot incorporate valuable alternative modalities like textual weather bulletins or field station reports, and crucially, lack interactive natural language interfaces for querying or reasoning.

Weather science workflows require substantial technical expertise to orchestrate complex ecosystems of tools, datasets, and models. Researchers must navigate disparate data sources, integrate outputs from multiple forecasting systems, combine observational datasets with model predictions, and coordinate between different computational environments and APIs. This dependency on extensive technical knowledge creates barriers for domain non-experts, limiting broader participation in weather science. Traditional meteorological workflows therefore require expert interpretation to translate computational outputs into actionable insights, increasing costs and limiting their utility in human-in-the-loop decision-support systems.

Multimodal LLMs can handle data from diverse modalities and offer a potential pathway to address these challenges. Models capable of jointly processing text with images (Wang et al., 2022; Alayrac et al., 2022; Li et al., 2022; Liu et al., 2023c), video (Zhao et al., 2022; Zhang et al., 2023; Cheng et al., 2024; Lin et al., 2024; Zhang et al., 2025), and audio (Chu et al., 2023; Défossez et al., 2024; Wu et al., 2024a; 2025; Doh et al., 2025; Ghosh et al., 2025) have shown impressive cross-modal reasoning abilities. Yet atmospheric data poses unique challenges: its spatiotemporal, multi-channel structure is fundamentally different from conventional modalities, requiring specialized approaches for effective integration with language models. Initial attempts to bridge this gap have shown promise but remain limited in scope. Early vision-language approaches to meteorology (Chen et al., 2024a; Li et al., 2024; Ma et al., 2024) have focused on narrow applications like extreme weather prediction using restricted variable subsets, falling short of general-purpose meteorological reasoning. More recent multimodal weather-language models (Varambally et al., 2025) demonstrate the potential of this direction but still fail to match established baselines across many important meteorological tasks. This persistent gap highlights a fundamental challenge: despite significant progress in both weather foundation models and LLMs, no existing system successfully unifies meteorological data with natural language reasoning for broad, interactive scientific applications.

We address this challenge by first introducing an agentic environment that enables LLMs to interact programmatically with meteorological data and models. We setup ZEPHYRUSWORLD, a comprehensive execution environment that exposes weather-focused capabilities through easy-to-use Python APIs. The system includes interfaces to the WeatherBench 2 dataset (Rasp et al., 2024), geo-query functionality for translating between coordinates and named locations, state-of-the-art forecasting models (Nguyen et al., 2024), a physics-based climate simulator based on JCM (Davenport et al., 2026), and a climatology tool that provides precomputed statistics such as mean, median, extreme

values, and quantiles from historical weather data. A FastAPI backend parallelizes code execution from LLM-generated queries.

We then develop two workflows of increasing sophistication within this agentic framework. ZEPHYRUS-DIRECT generates Python code in a single step to solve weather problems directly (Gao et al., 2023). ZEPHYRUS-REFLECTIVE employs an iterative execution–refinement framework (Yao et al., 2023b): it executes code to manipulate weather data, analyzes the results, and refines both code and output before providing a final answer. Both approaches can automatically detect and correct errors produced during code execution. Figure 1 gives an overview of our agentic pipeline.

To systematically evaluate these approaches, we construct ZEPHYRUSBENCH, a comprehensive benchmark built on ERA5 reanalysis data (Hersbach et al., 2020) from WeatherBench 2 (Rasp et al., 2024). The benchmark combines human-authored and semi-synthetic tasks spanning 2230 question–answer pairs across 49 distinct tasks. Tasks range from basic data lookups and forecasting to challenging research problems involving extreme event detection, forecast report generation, and prediction and counterfactual analysis. We also implement robust evaluation schemes to assess the scientific accuracy of all generated answers across diverse meteorological reasoning tasks. We summarize our key contributions below.

- We develop ZEPHYRUSWORLD, an agentic environment providing unified Python APIs for meteorological data, forecasting models, climate simulation, and climatology tools.
- We introduce two code-generating systems that leverage ZEPHYRUSWORLD: ZEPHYRUS-DIRECT for single-step code generation and ZEPHYRUS-REFLECTIVE for iterative execution-refinement workflows to solve open-ended meteorological problems.
- We curate ZEPHYRUSBENCH, a challenging weather reasoning benchmark with 2230 question-answer pairs across 49 meteorological task types.
- Our evaluation shows that LLM agents achieve encouraging results on the benchmark, suggesting that they can be effective assistants to weather scientists.

## 2 RELATED WORK

**Agentic frameworks for scientific discovery.** Agentic frameworks implement the perceive–reason–plan–act loop by pairing LLMs with tools, memory, and feedback to pursue long-horizon goals. Core patterns include interleaving reasoning with tool calls (ReAct (Yao et al., 2023a)), self-critique with episodic memory (Reflexion Shinn et al. (2023)), and self-supervised learning of API use (Toolformer (Schick et al., 2023)). General-purpose libraries such as AutoGen provide a standard interface for multi-agent conversation and tool invocation, making these patterns reusable across tasks (Wu et al., 2024c).

In many scientific applications, these frameworks appear as domain agents and self-driving labs. In chemistry, ChemCrow couples an LLM controller with a curated set of expert tools for synthesis and analysis (Bran et al., 2024), while Coscientist integrates retrieval, code execution, and laboratory APIs to plan and run experiments end-to-end (Boiko et al., 2023). Biomedical agents extend the approach across literature, databases, and analysis workflows (e.g., Biomni (Huang et al., 2025)). Despite these advances across multiple scientific domains, weather science remains largely unexplored territory for agentic approaches.

**General-Purpose Vision-Language Models.** Multi-modal vision language models (Li et al., 2021; Alayrac et al., 2022; Li et al., 2022; 2023; Liu et al., 2023c;b;a; 2024) demonstrate strong visual reasoning capabilities on general-purpose evaluation benchmarks. However, adapting these models for applications in weather science presents considerable difficulties. Standard VLM architectures assume RGB visual inputs and exhibit weaknesses in quantitative analysis tasks (Lu et al., 2024; Yue et al., 2024). Meteorological data presents fundamentally different challenges. They comprise of high-dimensional, structured atmospheric measurements which require specialized integration approaches for language model compatibility. While weather-language hybrid models (Varambally et al., 2025) seem promising, they underperform relative to domain-specific baselines across critical meteorological applications.

**Weather Foundation Models.** Neural network-based weather forecasting systems (Lam et al., 2023; Price et al., 2025; Bi et al., 2023; Pathak et al., 2022; Nguyen et al., 2023; Bodnar et al., 2024; Nguyen

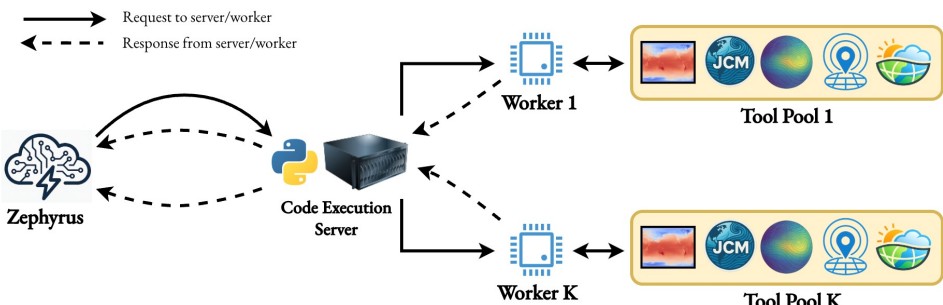

Figure 2: **Code Execution Server.** ZEPHYRUS sends parallel requests to the server, which distributes them to available workers. Each worker acquires resources from tool pools, loads datasets, injects tools into the execution environment, executes code, and returns results or errors to the agent.

et al., 2024) have revolutionized meteorological prediction by demonstrating superior performance compared to conventional physics-based approaches (Molteni et al., 1996) while being significantly more computationally efficient. Nevertheless, these architectures are predominantly trained for forecasting. In particular, they do not support conversational interfaces or cross-domain, multimodal reasoning capabilities.

**Multimodal Weather Datasets.** Recent research has developed several multimodal frameworks that combine weather observations with textual information. These include the Terra collection (Chen et al., 2024b), which integrates geographical imagery with descriptive text for general earth observation, and ClimateIQA (Chen et al., 2024a), which focuses on extreme weather detection through wind measurement analysis. Similarly, WeatherQA (Ma et al., 2024) specializes in severe weather interpretation using remote sensing data and expert commentary, while CLLMate (Li et al., 2024) connects media reports with ERA5 observations for weather event classification. Despite these valuable contributions, existing frameworks are narrow in scope. They concentrate on narrow applications or utilize only small subsets of atmospheric variables. This approach overlooks a fundamental characteristic of atmospheric dynamics: weather systems involve complex multi-scale interactions across numerous meteorological parameters. To address these limitations, our benchmark incorporates diverse weather reasoning tasks, both human-implemented and semi-synthetically generated, that span across most WeatherBench2 data channels.

## 3 ZEPHYRUS: AN AGENTIC FRAMEWORK FOR WEATHER SCIENCE

### 3.1 ZEPHYRUSWORLD: THE AGENTIC ENVIRONMENT

The fragmented nature of weather science tools makes it challenging for LLMs to effectively leverage them for scientific tasks. To address this, we introduce ZEPHYRUSWORLD, a comprehensive agentic environment that unifies weather science capabilities from diverse tools through a clean Pythonic interface. Given a question, we leverage LLMs' ability (Gao et al., 2023; Jimenez et al., 2024) to generate Python code and execute it in a sandboxed environment. The output is then fed back to the model along with any execution errors. We design high-level APIs for the tools for ease of use, and include documentation extracted from the docstrings in the models context at inference time.

The environment encompasses several essential weather science tools:

1. **WeatherBench 2 Data Indexer.** The environment provides the model access to the data through the `xarray` dataset interface.
2. **Geolocator.** This tool provides comprehensive geospatial functionality for weather data analysis. It handles forward geocoding (place names to coordinates) and reverse geocoding (coordinates to location names) using the Natural Earth dataset (Natural Earth, 2024). Key operations include finding geographic features at specific coordinates, retrieving boolean masks and area-weighted maps for regions, listing sublocations, and calculating geodistances. Built using `geopandas` and `shapely`, it maintains precomputed spatial caches for fast lookups.
3. **Forecaster.** We incorporate the Stormer model (Nguyen et al., 2024), a transformer-based neural weather prediction system trained on WeatherBench 2. We chose it for its strong performance at

short to medium range forecasts while being orders of magnitude more efficient than traditional numerical models. Our implementation abstracts checkpoint loading and preprocessing, providing a simple interface to run forecasts from arbitrary atmospheric initial conditions and return outputs as `xarray` datasets.

4. **Simulator.** We use the JCM simulator (Davenport et al., 2026), an intermediate complexity atmospheric model built on NeuralGCM's dynamical core (Kochkov et al., 2024). It incorporates physical parameterizations from the SPEEDY Fortran model (Molteni et al., 1996), including radiation, moist physics (clouds and convection), and vertical and horizontal diffusion. We use the default T32 configuration (approximately $3.5°$ resolution) with 8 vertical layers. Built on JAX, we can run 5-day simulations in only $\approx 25s$ on an A100 GPU.

5. **Climatology.** We precompute climatological baseline products for WeatherBench 2 variables across multiple aggregation timescales for the refernece period 1979-2000. Our tool exposes a simple query interface to retrieve summary statistics such as mean, extrema, standard deviation, and empirical quantiles. Supported timescales include all-time, seasonal, monthly, daily (day-of-year), and six-hourly climatologies.

**Code Execution Server.** ZEPHYRUSWORLD requires a system capable of handling multiple weather analysis tasks simultaneously without resource conflicts. We implement a FastAPI-based server-client architecture where clients send code execution requests to a dedicated execution server application that processes them in parallel. The system maintains resource pools for each tool component to prevent contention and enable true parallelism. Each pool contains one or more instances of the above tools. A resource manager implements acquire/release semantics to ensure each execution thread has exclusive access to a complete set of tools while preventing deadlocks. Each execution follows a strict protocol: acquire resources from pools, load requested datasets, inject tool instances into the execution environment, and execute user code with timeout protection. The system captures all outputs and error information, which are sent back to the client for further processing by the agent. Figure 2 provides an overview.

### 3.2 THE ZEPHYRUS FAMILY OF WEATHER AGENTS

We design agentic systems that leverage ZEPHYRUSWORLD to solve complex meteorological tasks. We construct prompts containing comprehensive documentation of ZEPHYRUSWORLDtools, variable descriptions, units, and coordinate systems. The models generate Python functions using these tools to solve the given questions, which execute on ZEPHYRUSWORLD's code execution server. Any execution errors or timeouts are returned to the models, which regenerate code until the error is resolved. We implement two distinct systems that differ in their execution strategy and refinement approach. Both systems intentionally maintain simple designs to isolate and measure the agentic capabilities of LLMs for solving weather science problems.

ZEPHYRUS-DIRECT generates a complete Python solution in one attempt and reports the execution output as the final answer. This model runs the error-correction loop for a maximum of 20 times.

ZEPHYRUS-REFLECTIVE implements a multi-turn workflow that alternates between code generation and execution phases. The agent executes individual code blocks and receives the output as observations. The execution results are fed back to the LLM, which analyzes the observations and decides on the next step. This ReAct-style (Yao et al. (2023b)) iterative process enables the model to assess the scientific plausibility of outputs, identify anomalies or mistakes in results, and refine subsequent code blocks to address logical errors. We run the interaction loop for a maximum of 20 times per question.

The complete prompts for both systems are presented in Appendix A.3.

## 4 ZEPHYRUSBENCH: A COMPREHENSIVE WEATHER BENCHMARK

Weather science problems require analyzing complex atmospheric patterns, modeling trends, and combining data from multiple sources. We introduce ZEPHYRUSBENCH, a comprehensive benchmark that evaluates how effectively LLMs can assist in real-world meteorological workflows. The benchmark comprises 49 distinct meteorological tasks with answers derived from curated weather reports and human-generated or verified code.

## 4.1 DATASET CURATION

We design our tasks based on the ERA5 reanalysis dataset (Hersbach et al., 2020), specifically from WeatherBench 2 (Rasp et al., 2024). The dataset provides global atmospheric data from 1979 to 2022. We use $1.5°$ spatial resolution with 6-hourly temporal resolution.

The capabilities measured by our curated tasks range from basic data lookups and computations to more advanced problems involving forecasting, challenging research problems including extreme event detection, forecast report generation, prediction analysis, and counterfactual reasoning. We design tasks with increasing difficulty levels (Easy, Medium, Hard) based on the complexity of tool usage required to answer them, from simple single-step data queries to multi-step analytical workflows. Table 2 lists the full set of tasks in ZEPHYRUSBENCH, while table 1 provides an overview of the different task types.

For each task-type, we define natural language templates with placeholders such as location, variable, and time window. To create task-specific examples, these placeholders are filled by randomly sampling inputs, and the corresponding ground truth is computed deterministically using human-written or human-verified synthetic code applied to the raw ERA5 data. Figure 7 shows an example template, and a sample generated from it.

Using our framework, we construct a benchmark dataset comprising 2230 test samples spread across 49 tasks. For a detailed breakdown of dataset statistics, please refer to Appendix A.1. We provide more details about how the tasks are implemented in the subsequent sections.

| Difficulty | Human-Gen. Tasks | Human-Gen. Samples | Synthetic Tasks | Synthetic Samples | Total Tasks | Total Samples |
|---|---|---|---|---|---|---|
| Easy | 7 | 793 | 0 | 0 | 7 | 793 |
| Medium | 4 | 182 | 29 | 290 | 33 | 472 |
| Hard | 8 | 765 | 1 | 200 | 9 | 965 |
| Total | 19 | 1,740 | 30 | 490 | 49 | 2,230 |

Table 1: **ZEPHYRUSBENCH Statistics**: Number of unique tasks and samples, grouped by difficulty and generation method.

### 4.1.1 HUMAN-GENERATED TASKS

The human-generated tasks span across the Easy, Medium and Hard difficulty levels and represent realistic meteorological queries curated in conjunction with a domain expert. For each task, a graduate student created a question template and wrote Python code to answer the query. Easy tasks focus on basic data retrieval operations like finding extrema, querying specific values, and identifying locations with particular weather conditions. Medium-difficulty tasks introduce forecasting elements, asking for future weather predictions at specific locations and times, and/or implementing complex data analysis pipelines. Hard tasks demand substantial meteorological expertise and mirror real-world operational workflows. These include extreme weather event detection, comprehensive weather assessments, and generation of detailed forecast discussions that span regional to global scales. For instance, ENSO outlook reports require synthesizing complex interactions between multiple atmospheric and oceanic variables to produce coherent, scientifically grounded forecasts. We source the expert-generated weather discussion reports from several online sources, such as the NOAA website[1] and IRI Seasonal Climate Forecasts/Outlooks[2]. For extreme weather event tasks, we use records from the EM-DAT international disaster database (Delforge et al., 2025), matching event entries by date and location to the ERA5 data.

### 4.1.2 SEMI-SYNTHETIC TASK GENERATION

To increase task diversity, we implement a semi-synthetic pipeline that transforms unstructured weather-related text into verifiable benchmark tasks. Figure 3 provides an overview of the procedure. The process begins with a claim extraction agent that analyzes weather texts from NOAA meteorological reports, using an LLM (specifically, `gpt-4.1`) to identify scientifically meaningful observational

---

[1] `https://www.wpc.ncep.noaa.gov/discussions/hpcdiscussions.php?disc=pmdepd`

[2] `https://iri.columbia.edu/our-expertise/climate/forecasts/seasonal-climate-forecasts/`

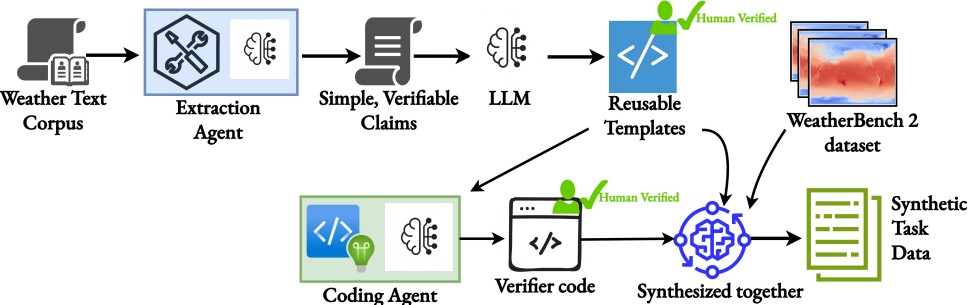

Figure 3: **Semi-synthetic task generation pipeline**: Weather-related texts are processed by a claim extraction agent to identify scientifically meaningful observational claims. Verified claims are transformed into reusable templates and manually reviewed. Code is generated by an LLM and verified by humans to validate each sample from a template against ERA5 meteorological data. We combine the verifier code with the templates and WeatherBench data to produce novel samples.

claims about weather phenomena. The agent focuses on quantifiable changes, trends, extremes, and relationships between variables.

These claims are then converted into question templates where we can substitute different locations, time periods, and weather variables to generate multiple benchmark examples from each original claim. For each template, an LLM writes a verification code block that can validate any instance generated from that template against the ERA5 data. This verification step ensures that the generated questions are not only linguistically coherent but also scientifically accurate when tested against actual meteorological observations. We generate multiple candidate instances from each template through this approach. Finally, we manually review them for scientific interest and code correctness. In this way, we generate 30 distinct human-validated synthetic task types.

We also include a semi-synthetic meteorological claim verification task to test whether models are capable of validating claims extracted from meteorological reports against the weather data; more details on this task are included in Appendix A.1.2.

## 4.2 EVALUATION METRICS

Since all our benchmark questions are designed around weather tasks with objectively correct answers, we develop an evaluation pipeline that can assess the scientific correctness of the answers produced by the models. The model answers fall into the following primary categories: **numerical**, **temporal**, **boolean**, **spatial (location-based)** and **descriptive**. Given that model outputs are in natural language, we evaluate them through a multi-stage process:

1. **Extraction:** Extract a structured answer from the model's natural language response using an LLM (`gpt-4.1-mini`). The expected format varies by question type.
2. **Verification:** Programmatically verify that the extracted answer is well-formed. If it is not, re-attempt extraction up to three times. At this stage, we merely assess whether or not the response has a valid answer to the given question, and not its correctness.
3. **Scoring:** Apply scoring methods specific to the type of question, which are detailed below.

**Numerical Answers.** For numerical responses, we record the Standardized Median Absolute Error between the predicted and reference values, standardized by the standard deviation of the corresponding variable in the dataset. This enables us to compare across variables with different scales and units. In addition, we also report the $25\%, 75\%$ and $99\%$ quantiles of the standarized absolute error to provide a more complete picture of the error distribution. We use quantiles rather than means because large outliers can significantly skew mean values, obscuring typical model performance patterns.

**Time-based Answers.** We evaluate tasks with time values as responses using Median Absolute Error. We omit the standarization step, since all the answers are in the same units (that is, hours). Like the numerical answers case, we also report the $25\%, 75\%$ and $99\%$ quantiles.

**Boolean Answers.** For tasks with binary True/False responses, we evaluate predictions using F1 score and correctness.

**Location-based Answers.** For questions whose answers are geographic locations, we first match the extracted location name to one of the expected entries from the NaturalEarth dataset (e.g., mapping "USA" to "United States of America"). For countries, we use the `country_converter` library (Stadler, 2017). For other geographic entities such as continents and water bodies, we apply fuzzy string matching (Bachmann et al., 2023), accepting matches above a predefined similarity threshold.

To quantitatively assess the geographic deviation between predicted and reference locations, we employ the Earth Mover's Distance (EMD) (Monge, 1781) as a primary evaluation metric. We begin by generating surface area-weighted masks over a latitude–longitude grid for both the predicted and reference locations. These masks are normalized to form probability distributions. To account for the curvature of the Earth, we compute pairwise distances between grid points using geodesic distance. The EMD is then calculated using the `POT` library (Flamary et al., 2021). As a complementary metric, we also report Location Accuracy, which simply measures whether the predicted and reference location strings are an exact match.

**Descriptive Answers.** To evaluate descriptive answers, we extract individual discussion points from both the model's response and the reference answer. We then classify each extracted claim from the model's response as either SUPPORTED, REFUTED, or NEUTRAL against the reference answer, obtaining logit scores from the language model and applying softmax normalization. Similarly, we perform the same procedure for claims from the reference text compared against the model response.

We then define two complementary metrics, precision and recall. We define precision as the validity of the model's claims by computing the proportion that are supported rather than refuted by the reference answer, excluding neutral classifications.

$$\text{Precision} = \frac{\sum_{i \in S} P_{\text{model}\to\text{ref}}(\text{Supported}_i)}{\sum_{i \in S} P_{\text{model}\to\text{ref}}(\text{Supported}_i) + \sum_{i \in S} P_{\text{model}\to\text{ref}}(\text{Refuted}_i)}$$

where $S = \{i : P_{\text{model}\to\text{ref}}(\text{Neutral}_i) < 0.5\}$ and $P_{\text{model}\to\text{ref}}(\text{Supported}_i)$ denotes the probability that model claim $i$ is supported by the reference answer, calculated from the model logits.

Recall measures coverage by evaluating how well the model response addresses the reference claims, computed as the average support probability across all reference points:

$$\text{Recall} = \frac{1}{N} \sum_{i=1}^{N} P_{\text{ref}\to\text{model}}(\text{Supported}_i)$$

where $N$ is the number of reference claims and $P_{\text{ref}\to\text{model}}(\text{Supported}_i)$ denotes the probability that reference claim $i$ is supported by the model answer.

Finally, we define the **discussion score** as the F1 score $= \dfrac{2 \cdot \text{Precision} \cdot \text{Recall}}{\text{Precision} + \text{Recall}}$.

**Extreme Weather Tasks.** In order to evaluate the extreme-weather tasks, we report two metrics: (1) F1 score, which only assesses whether the model correctly predicts the *occurrence* of an extreme event anywhere in the world, without considering event type or exact location. (2) EMD, which measures the agreement between the reference and predicted list of countries.

**Correctness.** For ease of presentation, we define correctness criteria that vary depending on the task type. Rather than requiring exact matches, we consider an answer correct if it falls within an acceptable range of the target response. The precise criteria for determining correctness for each task type are detailed in Appendix A.2.

## 5 EXPERIMENTAL RESULTS

We evaluate model performance across all task types from Section 4 using five LLM backbones: three proprietary models OpenAI GPT-5.2, GPT-5-Mini, Google Gemini 2.5 Flash (Comanici et al., 2025), and two open-source models OpenAI `gpt-oss-120b` (Agarwal et al., 2025) and `Qwen3-30B-A3B-Thinking-2507`. All the thinking models are evaluated with the reasoning

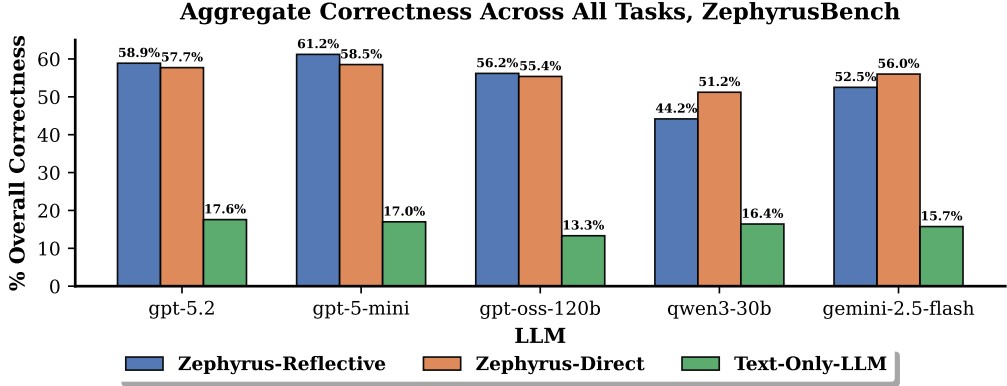

Figure 4: Percentage of questions in the complete dataset answered correctly by each LLM and model type. Definitions of correctness for each question type are detailed in Appendix A.2.

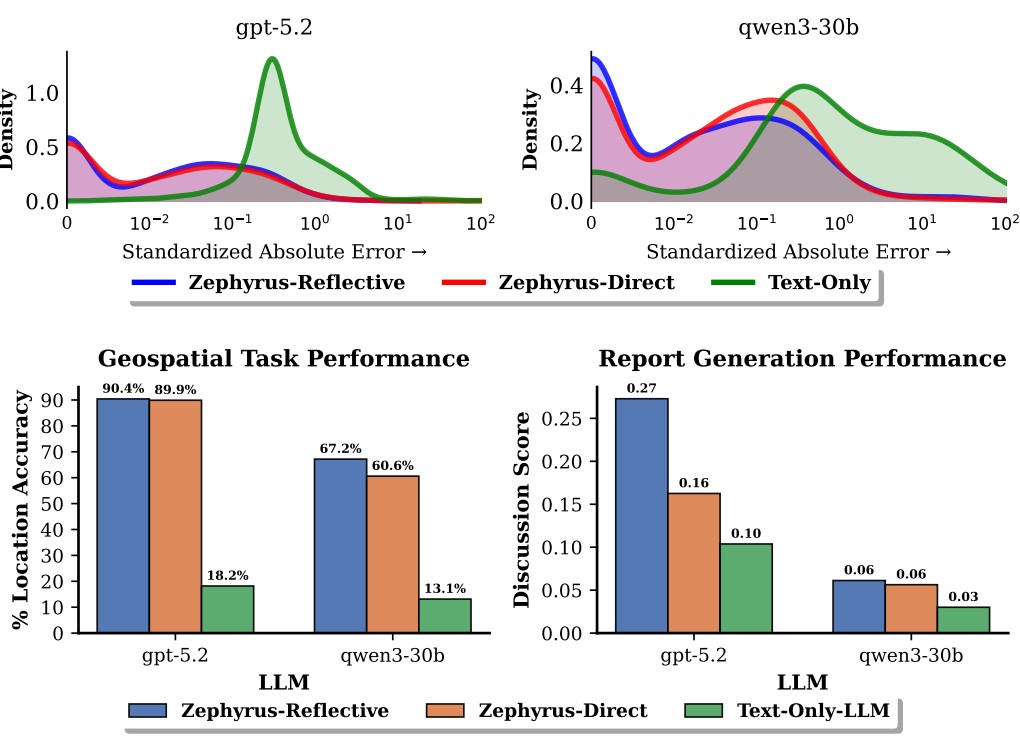

Figure 5: Plots showing (top) error distribution on numerical tasks (bottom-left) location accuracy (bottom-right) discussion scores for GPT-5.2 and Qwen3-30B.

effort set to `high`. We compare three experimental settings: (1) a text-only baseline that attempts to answer weather reasoning questions using only natural language metadata without access to structured weather data or numerical inputs, (2) ZEPHYRUS-DIRECT, and (3) ZEPHYRUS-REFLECTIVE. The text-only baseline measures the extent to which models can utilize their prior meteorological knowledge.

The correctness results across all models and tasks are presented in Figure 4. We observe that the ZEPHYRUS agents significantly outperform the text-only baseline across all models, demonstrating the agentic framework's ability to effectively ground answers by leveraging meteorological data from WeatherBench. For GPT-5-Mini, ZEPHYRUS-DIRECT and ZEPHYRUS-REFLECTIVE achieve 58.5%

and 61.2% correctness respectively, compared to only 17% for the text-only baseline. This substantial improvement holds consistently across other LLMs, with ZEPHYRUS agents achieving 27.8-44.2% higher correctness than their text-only counterparts. With the OpenAI models, the multi-turn execute-observe-solution framework implemented in ZEPHYRUS-REFLECTIVE enables it to outperform ZEPHYRUS-DIRECT by 0.8-2.7%. However, these gains are not consistent, with ZEPHYRUS-DIRECT outperforming ZEPHYRUS-REFLECTIVEwith Qwen3-30B and Gemini 2.5 Flash.

Figure 5 enables fine-grained analysis of error distributions for numerical tasks, location accuracy, and discussion scores for descriptive answers using GPT-5.2 and Qwen3-30B as the LLMs. The agents particularly excel at numerical and location prediction tasks, achieving substantially lower Standarized Absolute Errors and higher location accuracies compared to text-only baselines. For location prediction, ZEPHYRUS-REFLECTIVE with GPT-5.2 achieves strong performance with 90.4% accuracy. Once again, the reflective variant enjoys a small benefit in performance over the Direct approach. The difference between ZEPHYRUS-DIRECT and ZEPHYRUS-REFLECTIVE is significant on numerical tasks for Qwen3-30B, while both variants perform similarly with GPT-5.2.

However, all models struggle with the challenging task of generating textual weather reports. The best performing model (ZEPHYRUS-REFLECTIVE with GPT-5.2) only achieves a discussion score of 0.27. Nevertheless, ZEPHYRUS-REFLECTIVE demonstrates significant advantages over both ZEPHYRUS-DIRECT and text-only variants for these descriptive tasks. While the text-only variant lacks access to meteorological information, ZEPHYRUS-DIRECT produces rigid answers by directly outputting program results, making it ill-suited for nuanced textual generation. The execute-observe-solution framework in ZEPHYRUS-REFLECTIVE proves more effective.

Performance breakdown by difficulty level reveals interesting patterns (detailed results in Appendices A.4.6 and A.4.10). On easy tasks, which primarily involve data analysis questions, ZEPHYRUS agents perform well with 76.2-90.9% correctness. Medium difficulty tasks show moderate performance with 49.3-63.5% correctness. However, on hard tasks, all models struggle significantly, with ZEPHYRUS agents achieving 14.2%-37.7% correctness. This suggests that while current LLMs can effectively solve simple data analysis problems that arise in meteorology, they do not yet possess the capability to reason about weather phenomena directly from data even when provided with relevant tools. For generating meteorological discussions and forecasts for the continental United States, models show promise with ZEPHYRUS-REFLECTIVE + GPT-5.2 achieving an average discussion score of 0.43. This contrasts sharply with global climate forecasting tasks spanning three months, where all models fail completely, highlighting the current limitations in long-term, large-scale weather reasoning.

### 5.1 ABLATION STUDIES AND ADDITIONAL RESULTS

We conduct several ablation studies to better understand agent behavior and failure modes. These include analysis of representative failure cases (Appendix A.4.1), tool-use and runtime error-correction statistics (Appendix A.4.2), the impact of withholding individual tools from the agent's tool set (Appendix A.4.3), the performance upper bound when ground-truth forecasts are provided directly (Appendix A.4.4), and run-to-run variance across repeated trials (Appendix A.4.5).

## 6 CONCLUSION

We tackled the challenging problem of enabling LLMs to reason over high-dimensional weather data by developing, to our knowledge, the first agentic model for meteorology. Our contributions include: (1) ZEPHYRUSWORLD, an agentic environment with comprehensive meteorological tools, (2) the ZEPHYRUS family of agents that leverage these tools, and (3) a scalable data pipeline producing a large, diverse benchmark dataset (ZEPHYRUSBENCH). Our empirical evaluation shows that the agentic framework enables effective reasoning about meteorological data, significantly outperforming text-only baselines. The agents excel at most tasks but struggle with complex challenges like forecast report generation. Looking ahead, ZEPHYRUSWORLD's modular design makes it a good candidate to incorporate new tools, data sources, and domain-specific workflows, positioning it as a unified platform to support related fields such as hydrology, geosensing, and climate science. More broadly, our framework serves as a sandbox for developing and refining agentic workflows beyond meteorology. Future work could leverage more diverse and larger-scale datasets to further improve the scientific accuracy and generalizability of trained agent models.

## ACKNOWLEDGMENTS

Sumanth and Zachary would like to thank Chinmay Talegaonkar for several useful discussions about the work. This work was supported in part by the U.S. Army Research Office under Army-ECASE award W911NF-07-R-0003-03, the U.S. Department Of Energy, Office of Science, Google Academic Research Award, IARPA HAYSTAC Program, NSF Grants #2205093, #2146343, #2134274, and CCF-2112665, CDC-RFA-FT-23-0069, as well as DARPA AIE FoundSci and DARPA YFA.

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

## A  APPENDIX

### A.1  DATASET DETAILS

Table 2 details all the tasks in ZEPHYRUSBENCH, and table 1 reports the number of samples generated grouped by difficulty and type.

For weather tasks, we leverage the ERA5 reanalysis dataset (Hersbach et al., 2020), specifically from WeatherBench 2 (Rasp et al., 2024), which provides global atmospheric data from 1979 to 2022. We use $1.5°$ spatial resolution with 6-hourly temporal resolution, and include 4 surface variables and 5 atmospheric variables at 13 pressure levels. For each task-type, we define natural language templates with placeholders such as location, variable, and time window. For example, Task 1 is defined as 'Which {geofeature} experienced the {extremum_direction} average {variable}?'. To create task-specific examples, these placeholders are filled by randomly sampled inputs. Ground truth answers are obtained by a deterministic procedure: we apply human-written or human-verified synthetic code to the raw ERA5 data and other supplementary data.

### A.1.1  HUMAN-GENERATED TASKS

Tasks 1 through 7 rely entirely on the raw ERA5 data. Tasks 1-5 include basic data lookups and computations, while Tasks 6 and 7 involve simple comparisons with the climatological quantities. Tasks 8-9 are forecasting questions; however, we use the ground-truth answer as the response while curating the dataset. We extensively use the Geolocator tool to map natural language names to masks that are applied to spatiotemporal data.

For Tasks 11, 12 and 48, which involve extreme event detection, we use records from the EM-DAT international disaster database (Delforge et al., 2025), matching event entries by date and location to ERA5 data. We consider the metereological events, comprising storms, heat and cold waves.

| ID | Natural Language Description | Answer Type | Difficulty | Type |
|---|---|---|---|---|
| 1 | Which geographic feature experienced the highest/lowest average value of a weather variable | Location | Easy | Human |
| 2 | What is the min/max/average/median value of a weather variable at a specific location | Numerical | Easy | Human |
| 3 | Which sublocation has the highest/lowest recorded variable value | Location | Easy | Human |
| 4 | How many hours from start did a location experience extremum | Temporal | Easy | Human |
| 5 | What is the weather variable value at a location at a specific time | Numerical | Easy | Human |
| 6 | Identify geofeatures where variable differs from mean by $N$ standard deviations | List of locations | Easy | Human |
| 7 | Identify geofeatures where variable lies outside climatological percentile envelope | List of locations | Easy | Human |
| 8 | What will the variable be at a location after time interval (forecast) | Numerical | Medium | Human |
| 9 | When will location experience its extremum in future period (forecast) | Temporal | Medium | Human |
| 10 | Which geofeatures are experiencing exceedance in variable values | List of locations | Medium | Human |
| 11 | Identify extreme weather events that will occur in the next $N$ hours (forecast) | List of locations | Hard | Human |
| 12 | Check if extreme weather events are currently happening | List of locations | Hard | Human |
| 13 | Which geographic features experienced unusual weather anomalies compared to baseline | List of locations | Medium | Human |
| 14 | Does maximum weather variable occur at same or adjacent grid point as another variable (forecast) | Yes/No | Medium | Synthetic |
| 15 | Does maximum weather variable in region remain lower than future maximum (forecast) | Yes/No | Medium | Synthetic |
| 16 | Does maximum weather variable occur at higher latitude than in another region (forecast) | Yes/No | Medium | Synthetic |
| 17 | Does mean weather variable in one region exceed another by specified amount (forecast) | Yes/No | Medium | Synthetic |
| 18 | Does mean weather variable exceed threshold while maximum of another stays below (forecast) | Yes/No | Medium | Synthetic |
| 19 | Does mean weather variable within region exceed specified threshold (forecast) | Yes/No | Medium | Synthetic |
| 20 | Does weather variable exceed threshold within any part of region (forecast) | Yes/No | Medium | Synthetic |
| 21 | Does weather variable exceed threshold in more grid points in one region than another (forecast) | Yes/No | Medium | Synthetic |
| 22 | Does area where weather variable exceeds threshold cover more than percentage of region (forecast) | Yes/No | Medium | Synthetic |
| 23 | Does area-averaged weather variable exceed threshold while another stays below (forecast) | Yes/No | Medium | Synthetic |
| 24 | Does maximum weather variable in one region exceed threshold while another stays below (forecast) | Yes/No | Medium | Synthetic |
| 25 | Does maximum weather variable within region exceed specified threshold (forecast) | Yes/No | Medium | Synthetic |
| 26 | Does maximum weather variable occur at latitude farther north than in another region (forecast) | Yes/No | Medium | Synthetic |
| 27 | Does maximum weather variable stay above threshold while another stays below (forecast) | Yes/No | Medium | Synthetic |
| 28 | Does maximum weather variable in one region exceed another by specified amount (forecast) | Yes/No | Medium | Synthetic |
| 29 | Does minimum weather variable within region remain above threshold (forecast) | Yes/No | Medium | Synthetic |
| 30 | What is the area where multiple weather variables exceed their percentile values (forecast) | Numerical | Medium | Synthetic |
| 31 | What is the area where weather variable exceeds its median value (forecast) | Numerical | Medium | Synthetic |
| 32 | What is the displacement between centroids of areas with above-median values (forecast) | Numerical | Medium | Synthetic |
| 33 | What is the distance between centroids of maximum weather variable value areas (forecast) | Numerical | Medium | Synthetic |
| 34 | What is the maximum difference in weather variable between grid points within region (forecast) | Numerical | Medium | Synthetic |
| 35 | What is the minimum weather variable where another variable exceeds median (forecast) | Numerical | Medium | Synthetic |
| 36 | What is the difference between maximum weather variables in two regions (forecast) | Numerical | Medium | Synthetic |
| 37 | What is the difference in area-weighted mean weather variable between two regions (forecast) | Numerical | Medium | Synthetic |
| 38 | What is the displacement of minimum weather variable location after time window (forecast) | Numerical | Medium | Synthetic |
| 39 | What is the latitude difference between centroids of high weather variable areas (forecast) | Numerical | Medium | Synthetic |
| 40 | What is the maximum weather variable difference between two regions (forecast) | Numerical | Medium | Synthetic |
| 41 | What is the mean weather variable where another variable exceeds median (forecast) | Numerical | Medium | Synthetic |
| 42 | What is the weather variable value where another variable reaches maximum (forecast) | Numerical | Medium | Synthetic |
| 43 | Generate comprehensive global climate forecast for temperature and precipitation for next 3 months (forecast) | Description | Hard | Human |
| 44 | Provide detailed meteorological discussion and forecast for continental United States (forecast) | Description | Hard | Human |
| 45 | Generate ENSO climate update and outlook based on atmospheric data (forecast) | Description | Hard | Human |
| 46 | How will weather variable change after specified time given an intervention (counterfactual) | Numerical | Hard | Human |
| 47 | What is the value of the input parameter of the simulator model that produces the simulation output | Numerical | Hard | Human |
| 48 | Where is a target disaster currently happening | List of locations | Hard | Human |
| 49 | Check whether the given claim extracted from meteorological report is supported by the data | Yes/No | Hard | Synthetic |

Table 2: Complete set of Weather Tasks, grouped by difficulty.

The anomaly detection tasks (Tasks 6, 10 and 13) capture the model's ability to compare global data with precomputed quantile statistics derived from a historical reference period and accessed through the Climatology tool. Locations where the recent values significantly exceed or fall below the reference quantile are flagged as anomalous. We then use the Geolocator to map flagged grid points to natural language region names.

Report generation tasks (ID 43, 44, 45) are designed to evaluate forecasting and interpretation capabilities based on ERA5 atmospheric datasets. They all use global weather fields over the given

time duration as context. Task 43 requires generating a comprehensive global forecast report for temperature and precipitation for a three month horizon into the future. The task instructs the report to be structured into separate sections for precipitation and temperature, and to provide region-specific forecasts. Task 44 focuses on the continental United States, where the model must provide a detailed meteorological discussion and forecast, including current weather system positions and movements, temperature trends and expected changes over the coming days, precipitation patterns and likelihood of significant events, pressure system evolution and impacts, and notable atmospheric features such as fronts and jet stream positioning. Task 45 requires an ENSO (El Niño–Southern Oscillation) climate update and outlook. Models are tasked to analyze atmospheric variables to assess the current ENSO phase, evaluate strength and persistence indicators, forecast evolution over the next $3 - 6$ months, and discuss global implications using uncertainty-aware language and standard ENSO terminology. Ground truth reports for these tasks are obtained from authoritative government and research institutes, which provide validated assessments of global and regional climate outlooks and ENSO conditions. The answer sources for these three tasks are NOAA Global Climate Reports, NOAA National Weather Service Area Forecast Discussions, and WMO ENSO Reports, respectively.

Task 46 and Task 47 both rely on the JAX-GCM simulator, an intermediate-complexity atmospheric model built on NeuralGCM's dynamical core (Kochkov et al., 2024). Task 46 assesses the causal impact of localized perturbations on atmospheric states. To obtain each specific sample, a variable, location, and perturbation magnitude are first sampled, and a Gaussian mask is applied to induce the desired perturbation at the chosen location. The simulator is then run twice, once starting from the unperturbed initial state and once with the imposed perturbation. At the specified simulation end time, the target variable from both simulations is extracted and compared, with the difference quantifying the perturbation's impact.

Task 47 is a black-box optimization task based on the simulator. The input consists of two components: (i) a segment of recent global data spanning a specified duration and interval, and (ii) simulated data generated by the simulator. In the simulation, we vary one input parameter of the model by sampling its value randomly from the range $[0, 1]$, then save the resulting simulation output. The objective of the task is to estimate the original value of the underlying input parameter from observable simulation outputs. Since the climate simulator is presented as a black box, the model must infer the parameter solely from the input-output mapping, which can be highly nonlinear and sensitive to small parameter changes. By evaluating the model on novel simulator outputs, we benchmark its general handling of a domain-specific optimization problem. Performance is assessed by comparing the predicted and ground-truth parameter values.

### A.1.2  METEOROLOGICAL CLAIM VERIFICATION

In the Meteorological Claim Verification task (Task 49), we extract metereological claims from NOAA monthly meteorological reports (1988–2024) into timestamped text using an LLM (`gpt-4.1-mini`). We then select individual claims and pair them with the 24-hour slice of WeatherBench2 data corresponding to the report's date. Negative instances are generated by systematically negating claims using an LLM. All examples are human-verified to ensure clarity, verifiability, and correctness of negation. Figure 6 demonstrates an example from this task.

### A.2  DEFINITION OF TASK CORRECTNESS

Different task types in ZEPHYRUSBENCH are evaluated using relevant metrics. To create a unified definition of correctness, we employ the following requirements for each metric type:

- **Numerical:** Standardized difference $\frac{|\hat{y}-y|}{\sigma} < 0.05$, where $\sigma$ is the standard deviation of the relevant task variable in the WeatherBench2 dataset.
- **Distance/Area/Coordinate/Simulation:** Relative error $\frac{|\hat{y}-y|}{|y|} < 0.05$. For true values of 0, we require $|\hat{y}| < 0.05$.
- **Location:** Exact locations string match, using fuzzy string matching logic.
- **Extreme Weather/Anomaly:** Earth Mover's Distance (EMD) score $< 100$ km. If both true and predicted values are empty lists, the answer is considered correct.
- **Boolean:** Exact match between model answer and ground truth boolean value.

The following data shows meteorological conditions over a 24-hour period:

{'type': 'wb2', 'variables': ['mean_sea_level_pressure', '10m_u_component_of_wind', '10m_v_component_of_wind', '2m_temperature', 'geopotential', 'specific_humidity', 'temperature', 'u_component_of_wind', 'v_component_of_wind'], 'time_indices': '47007:47011:1', 'start_idx': 47007}

Based on the provided data, answer the following question: Does this data support the provided meteorological claim? Answer with True or False.

Claim: A very strong jet in excess of 150 knots will be across the east central U.S. with a favorable left exit region of the jet over New England.

**Positive Claim Type**

**Original Claim:** A very strong jet in excess of 150 knots will be across the east central U.S. with a favorable left exit region of the jet over New England.

The following data shows meteorological conditions over a 24-hour period:

{'type': 'wb2', 'variables': ['mean_sea_level_pressure', '10m_u_component_of_wind', '10m_v_component_of_wind', '2m_temperature', 'geopotential', 'specific_humidity', 'temperature', 'u_component_of_wind', 'v_component_of_wind'], 'time_indices': '79179:79183:1', 'start_idx': 79179}

Based on the provided data, answer the following question: Does this data support the provided meteorological claim? Answer with True or False.

Claim: Dry conditions with minimal precipitation are expected across western Washington and into British Columbia.

**Negative Claim Type**

**Original Claim**: Heavy precipitation is expected across western Washington and into British Columbia.

Figure 6: (left) Positive claim and (right) negative example for meteorological claim verification

The following data shows a snapshot of the global weather fields.
{data}
Based on the above data, answer the following question:

Which {geofeature} experienced the {extremum_direction} average {variable}?","Based on the provided data, {answer} experienced the {extremum_direction} average {variable} over the specified time-period, with an average {variable} of {answer_numeric}."

**Example Template**

The following data shows a snapshot of the global weather fields.

{'type': 'wb2', 'variables': ['mean_sea_level_pressure', '10m_u_component_of_wind', '10m_v_component_of_wind', '2m_temperature', 'geopotential', 'specific_humidity', 'temperature', 'u_component_of_wind', 'v_component_of_wind'], 'time_indices': '54746:54747:1'}

Based on the above data, answer the following question: Which continent experienced the highest average Surface temperature?

Based on the provided data, Africa experienced the highest average Surface temperature over the specified time-period, with an average Surface temperature of 303.5 K.

**Generated Sample**

Figure 7: (left) Example template from which samples are generated and (right) a sample generated using the template.

- **Discussion:** Overall discussion score $> 0.5$.

- **Time:** Exact match required (absolute error $= 0.0$).

## A.3 MODEL PROMPTS

We use the following core Instruction prompt for ZEPHYRUS-REFLECTIVE:

## Zephyrus-Reflective Instruction Prompt

```
You are an AI weather expert agent. You will use an interactive coding environment with
↪  tool functions, data, and softwares to solve the user's task.

At each turn, you should first provide your thinking and reasoning given the
↪  conversation history (which might include output from executed code within
↪  <observation></observation>).
After that, you must do exactly one of the following:
1) Write code based on problem and/or observation. Your code should be enclosed using
↪  "<execute>" tag, for example: <execute> return "Hello World!" </execute>. IMPORTANT:
↪  You must end the code block with </execute> tag.
2) When you think you have a solution ready, directly provide a solution that adheres to
↪  the required format for the given task to the user.
Your solution should be enclosed using "<solution>" tag, for example: The answer is
↪  <solution> A </solution>. IMPORTANT: You must end the solution block with </solution>
↪  tag. When answering numerical questions, always use SI base unit (standard units of
↪  measurement) unless the problem specifically asks for a certain unit. For example,
↪  some questions may require you to answer in hours. Enclose ONLY the final answer to
↪  the question in these tags, do NOT include any other information.

In each response, you must include <execute> or <solution> tag. Not both at the same
↪  time. Do not generate code outside <execute>. Do not output answers outside
↪  <solution>. Do not respond with messages without any tags. No empty messages.

- Geolocator Documentation:

The detailed documentation for the Geolocator class, including its available methods, is
↪  provided below:

{geolocator_documentation}

------------------------------------------------------------------------
- Forecaster API Documentation:

{forecaster_documentation}

The Forecaster can reliably forecast at most 2 weeks into the future.
- IMPORTANT: If the question is about the future, you **will need to** use the Forecaster
↪  object to answer the question and solve the task.
             The input data **will not** contain the answer to questions about the
             ↪  future.
------------------------------------------------------------------------
- Simulator API Documentation:

{simulator_documentation}

- The Simulator provides atmospheric modeling and can be used for climate simulations,
↪  answering counterfactuals, sensitivity studies, or generating synthetic weather
↪  data.
- The Simulator can handle extended time periods (months to years) in a SINGLE call. DO
↪  NOT create loops or multiple simulator instances. Set total_time to the desired
↪  duration and call simulate() once.
------------------------------------------------------------------------

- Variable Descriptions:

A comprehensive description of every variable contained in the xarray datasets is given
↪  here:
{var_desc}

- Dataset Keys Explanation:

An explanation of what each key in the datasets represents is provided below:
{keys}

...(continued)
```

**Zephyrus-Reflective Instruction Prompt (cont.)**

```
(continued)...

- Units:

Always use the following SI units when reasoning and coding:
{units_desc}
Answer in SI units unless the problem specifically asks for a different unit. For
↪  example, some questions may require hours.

- Time Indices:
You should NOT slice the provided dataset according to the time indices. The datasets
↪  are already sliced to the correct time indices.
For any question that asks about the time offset, only provide the time indices relative
↪  to the provided dataset.
If the question asks for the time offset, return the answer in hours from the initial
↪  time index.
For example, if the question asks about a dataset with time interval 6 hours and time
↪  indices 12345:12351:1, and you think the answer is index 12350, you should return 30
↪  hours.
Do NOT return the time index as a timestamp or datetime object.

**Execution code requirements:**
- The code MUST all be defined with a function called `run`.
- The `run` function should accept four parameters:
  a. A list of one or more xarray datasets.
  b. A Geolocator object (which comes with a set of predefined helpful functions).
  c. A Forecaster object (which comes with a set of predefined helpful functions).
  c. A Simulator object (which comes with a set of predefined helpful functions).
- DO NOT write any code outside of the `run` function.

**IMPORTANT:**
- The Geolocator object is already constructed and passed in as `geolocator`.
- **Never open files, use `xr.open_dataset`, or import Geolocator.**
- If you are subsetting, make sure to subset carefully considering runtime. It is too
↪  slow to select the entire xarray dataset. If you are subsetting over multiple
↪  dimensions (e.g. spatially and temporally), make sure to apply the smaller subset
↪  operation first.
- By following these detailed instructions, your code should clearly use the provided
↪  datasets and tools to produce the correct result.

- Coordinate System:
The WeatherBench2 (WB2) dataset uses an equiangular grid with the following
↪  specifications:
- Latitude: 121 grid points ranging from -90° to +90° in 1.5° increments
- Longitude: 240 grid points ranging from 0° to 358.5° in 1.5° increments
- The latitude coordinates are: [-90, -88.5, -87, ..., 87, 88.5, 90]
- The longitude coordinates are: [0, 1.5, 3, ..., 355.5, 357, 358.5]

**Other Requirements:**
- Under NO circumstances should you loop over the grid points (i.e. you should NOT loop
↪  over latitudes and longitudes), but rather try to leverage vectorized operations,
↪  built-in functions or the Geolocator class as appropriate. This is a key requirement.
↪  DO NOT loop over the latitudes and longitudes ANYWHERE in your generated code.
- Ensure that you call and use the functions from the Geolocator object correctly as per
↪  its documentation.

**Question:**
{question}
```

For the reflective stage of ZEPHYRUS-REFLECTIVE, we use the following Observation prompt:

**Zephyrus-Reflective Observation Prompt**

```
The executed code produced the output above. Reason about your next step and either (1)
↪  output the final result based on this observation. Enclose your answer in
↪  <solution></solution> tags., or (2) generate another code block to execute. Enclose
↪  your code in <execute></execute> tags.
If you choose to give a solution, enclose ONLY the final answer to the question in these
↪  tags, do NOT include any other information.
You should execute code if you think you need more information before providing a final
↪  answer.
```

For ZEPHYRUS-DIRECT, we use the following direct Instruction prompt:

---

**Zephyrus-Direct Instruction Prompt**

```
Your objective is to write a Python function called 'run' that solves a specified
↪   problem using provided data and Toolset APIs. The function should be designed
↪   according to the following guidelines:

1. Function Definition:

- The function must be named run.
- It should accept four parameters:
  a. A list of one or more xarray datasets.
  b. A Geolocator object (which comes with a set of predefined helpful functions).
  c. A Forecaster object (which comes with a set of predefined helpful functions).
  c. A Simulator object (which comes with a set of predefined helpful functions).

2. Data Descriptions:

- Variable Descriptions:

A comprehensive description of every variable contained in the xarray datasets is given
↪   here:
{var_desc}

- Dataset Keys Explanation:

An explanation of what each key in the datasets represents is provided below:
{keys}

- Units:

Always use the following SI units when reasoning and coding:
{units_desc}

- Time Indices:

The datasets provided have been converted from using a time dimension to simple integer
↪   indices starting from 0. Each index step represents 6 hours of time in the original
↪   dataset.
You should NOT slice the provided dataset according to the provided indices. The
↪   datasets are already sliced to the correct indices.
For any question that asks about the time offset, only provide the time indices relative
↪   to the provided dataset.
If the question asks for the time offset, you should return the answer in hours from the
↪   initial time index.
For example, if the question asks about a dataset with time interval 6 hours and indices
↪   0:6:1, and you think the answer is index 5, you should return 30 hours.
Do NOT return the index directly.

3. Toolset APIs

You are given access to the following code tools. Please use them as needed inside your
↪   `run` function:

- Geolocator Documentation:

The detailed documentation for the Geolocator class, including its available methods, is
↪   provided below:

{geolocator_documentation}

------------------------------------------------------------------------
- Forecaster API Documentation:

{forecaster_documentation}
The Forecaster can reliably forecast at most 2 weeks into the future.
- IMPORTANT: If the question is about the future, you **will need to** use the Forecaster
↪   object to answer the question and solve the task.
            The input data **will not** contain the answer to questions about the
                ↪   future.
------------------------------------------------------------------------

...(continued)
```

**Zephyrus-Direct Instruction Prompt (cont.)**

```
(continued)...

- Simulator API Documentation:

{simulator_documentation}

- The Simulator provides atmospheric modeling and can be used for climate simulations,
↪  answering counterfactuals, sensitivity studies, or generating synthetic weather
↪  data.
- The Simulator can handle extended time periods (months to years) in a SINGLE call. DO
↪  NOT create loops or multiple simulator instances. Set total_time to the desired
↪  duration and call simulate() once.
-----------------------------------------------------------------------

4. Task Details:

- The function should process the datasets using the pertinent variables as specified
↪  within the question.
- Under NO circumstances should you loop over the grid points (i.e. you should NOT loop
↪  over latitudes and longitudes), but rather try to leverage vectorized operations,
↪  built-in functions or the Geolocator class as appropriate. This is a key requirement.
↪  DO NOT loop over the latitudes and longitudes ANYWHERE in your generated code.
- Ensure that you call and use the functions from the Geolocator object correctly as per
↪  its documentation.

5. Returning the Answer:

- The final result should be returned by the function.
- Make sure to encapsulate your run function in triple backticks for clarity. For
↪  example:
```
def run(...):
    return "Hello"
```
- If the answer is a time value, make sure to return it in a unit of time rather than as
↪  a timestamp or datetime object. For example, return `5 hours` instead of `2022-01-01
↪  05:00:00`.
- Always return the answer in the same unit as the one used in the weatherbench dataset.
↪  Do not convert any units.

6. Problem Statement:

By following these detailed instructions, your code should clearly use the provided
↪  datasets and the Toolset APIs to produce the correct result.
The specific question that your function needs to answer is provided at the end of this
↪  prompt: {question}
```

We use the following prompt for the text-only LLM:

**Text-Only LLM Prompt**

```
Provide an answer to the following question.
Do not complain about insufficient data.
Be brief, and provide a 1-2 sentence answer.
Always provide your answer in SI units.
Do not give approximate answers.
If the question asks for the time offset, you should return the answer in hours from the
↪  initial time index.
For example, if the question asks about a dataset with time interval 6 hours and time
↪  indices 12345:12351:1, and you think the answer is index 12350, you should return 30
↪  hours.
Do NOT return the time index as a timestamp or datetime object.
Provide a concrete final answer. It is unacceptable to say "Without data I cannot answer
↪  this question", or, "I am sorry but I cannot provide this answer".
{question}
```

Figure 8: Example failure mode of ZEPHYRUS-REFLECTIVE (GPT-5.2) on the extreme weather event detection task. The expected ground truth (left) is a storm in southern USA, while the model (right) predicts a storm in Alaska and Russia.

## A.4 ADDITIONAL RESULTS

### A.4.1 EXAMPLE FAILURE MODE

We illustrate an example failure mode of ZEPHYRUS-REFLECTIVE (GPT-5.2) on a challenging extreme weather event prediction task. The task is to forecast whether any extreme weather event will develop in the 42 hours following the end of the data window (December 13, 2003, 06:00am UTC), and to identify the affected countries and regions. The ground truth is that a general storm is expected to affect a large swathe of the continental United States, including Texas (Jefferson, Fort Bend, Harris, and Dallas districts), California (Los Angeles), Mississippi (Smith district), and the states of Alabama, Arkansas, Florida, Georgia, Louisiana, Maryland, Missouri, North Carolina, Oklahoma, South Carolina, Tennessee, Virginia, and West Virginia, with the event expected approximately 42 to 66 hours out. Figure 8 illustrates the expected and predicted model answers.

We analyze the agent's behavior across its four turns to understand how it arrived at its final answer.

**Turn 1.** The agent uses the forecaster to produce a 7-step (42-hour) autoregressive forecast using the Forecaster tool. At each step, it extracts the global MSLP minimum and the global maximum 10m wind speed, then checks whether these exceed hardcoded absolute thresholds (MSLP < 98,000 Pa and wind $\geq$ 25 m/s, or wind $\geq$ 30 m/s). No extreme event is detected, as wind speeds peak at approximately 21 m/s, just below the threshold.

**Turn 2.** The agent retries with loosened thresholds, now flagging a storm if MSLP < 96,000 Pa and wind $\geq$ 20.8 m/s, or if wind $\geq$ 25 m/s at any point. While the MSLP values (peaking at approximately 95,200 Pa) satisfy the pressure condition, the global wind maximum of 21 m/s does not co-occur with the deepest pressure at the same timestep, and again no extreme event is detected.

**Turn 3.** The agent switches to a climatology-relative criterion. It retrieves the all-time 1st percentile MSLP from the climatology object at the grid point nearest to the forecast pressure minimum, and flags an extreme event if the forecast falls below that local threshold. This criterion is now satisfied (forecast 95,220 Pa vs. climatological 1st percentile 96,839 Pa), and the storm is classified as an extra-tropical storm tracking across the North Pacific into the Bering Sea. However, when the agent samples the gale-wind footprint (wind $\geq$ 17 m/s) to identify affected land regions, it finds only water bodies: the North Pacific Ocean, the Norwegian Sea, and the Labrador Sea.

**Turn 4.** Noticing the absence of any land regions in its output, the agent attempts to recover by manually probing a set of hardcoded coordinates near the storm track, covering the Aleutian Islands, southwest Alaska, and the Kamchatka Peninsula, using the Geolocator. This returns Alaska (United States) and Kamchatka (Russia) as the nearest land regions, along with the Bering Sea and Gulf of Alaska as relevant water bodies.

**Final Answer.** The agent incorrectly reports an extra-tropical storm centered over the Bering Sea, peaking at 24 hours, with Alaska and Kamchatka identified as the regions at risk. The ground truth answer, a general storm affecting more than 15 states across the continental United States, had a comparatively modest pressure signature and never registered as the global MSLP minimum during the forecast window. Because the agent's detection pipeline was anchored throughout to the

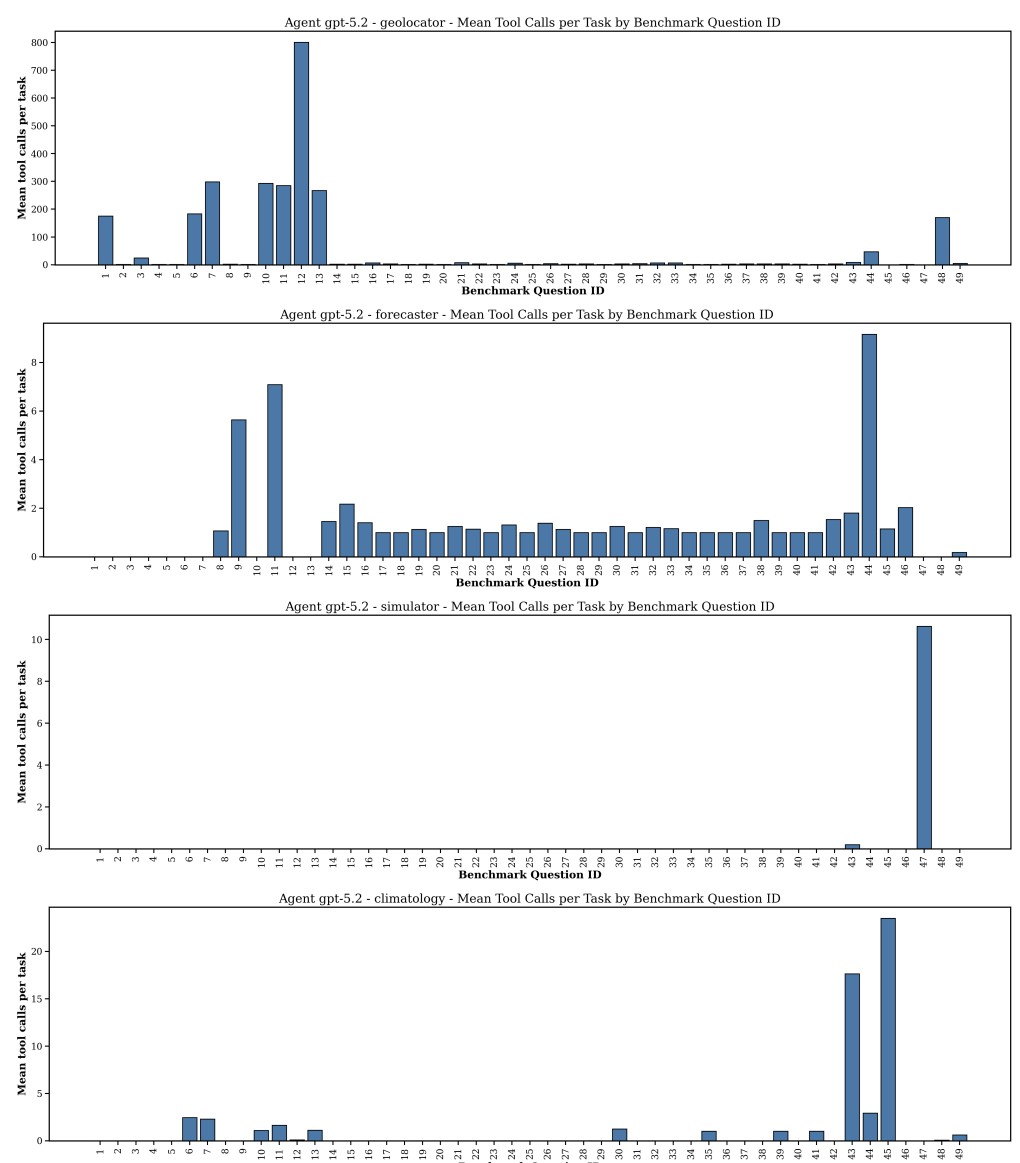

Figure 9: Tool usage statistics for ZEPHYRUS-REFLECTIVE (GPT-5.2) grouped by Question ID.

| Model | Tool | Coverage (%) | Total Calls | Mean Calls/Task | Error Rate (%) | Errors Corrected | Avg. # Turns | Err. Attempts Exceeded |
|---|---|---|---|---|---|---|---|---|
| ZEPHYRUS-REFLECTIVE | Geolocator | 75.4 | 132,273 | 59.3 | 7.6 | 170 | 1.4 | 0 |
| | Forecaster | 43.0 | 2,682 | 1.2 | | | | |
| | Simulator | 4.9 | 1,082 | 0.5 | | | | |
| | Climatology | 20.1 | 3,535 | 1.6 | | | | |
| ZEPHYRUS-DIRECT | Geolocator | 89.4 | 101,526 | 45.5 | 6.1 | 137 | 1.1 | 0 |
| | Forecaster | 44.1 | 1,982 | 0.9 | | | | |
| | Simulator | 5.3 | 324 | 0.1 | | | | |
| | Climatology | 25.9 | 3,372 | 1.5 | | | | |

Table 3: Tool usage/calls and aggregate runtime error statistics for ZEPHYRUS-REFLECTIVE (GPT-5.2) on ZEPHYRUSBENCH.

single deepest pressure center on the globe, the far more consequential continental system was never examined.

### A.4.2    TOOL USE AND ERROR-CORRECTION STATISTICS

In this section, we examine the tool-usage patterns of ZEPHYRUS and examine the run-time error correction statistics. Table 3 details the aggregate tool-use statistics and runtime-error correction

| Tools | % Correct | Easy | Medium | Hard | Location Acc | EMD | Valid % |
|---|---|---|---|---|---|---|---|
| All Tools | **56.7%** | 86.3% | 62.0% | **28.7%** | **76.8%** | 1,472 | **99.2%** |
| Exclude Climatology | 50.8% | 74.1% | 58.6% | 27.2% | 74.8% | 2,870 | 97.9% |
| Exclude Forecaster | 44.3% | 84.9% | 28.4% | 16.0% | 73.7% | 1,744 | 93.4% |
| Exclude Geolocator | 37.2% | 42.9% | 49.5% | 27.0% | 22.2% | 4,497 | 98.5% |
| Exclude Simulator | 55.6% | **87.0%** | **62.5%** | 25.4% | 73.7% | 1,489 | 96.9% |
| Exclude Dataset, Forecaster, Simulator | 18.6% | 13.4% | 37.0% | 15.2% | 29.3% | 4,922 | 91.2% |

Table 4: Tool ablation study on ZephyrusBench.

| Tools | % Correct | Valid % |
|---|---|---|
| All Tools | 33.3% | **100.0%** |
| Exclude Climatology | **38.1%** | **100.0%** |
| Exclude Forecaster | 0.00% | 71.4% |
| Exclude Geolocator | 33.3% | **100.0%** |
| Exclude Simulator | **38.1%** | **100.0%** |
| Exclude Dataset, Forecaster, Simulator | 0.00% | 90.5% |

Table 5: Tool ablation for Template 8: Forecast Value After Time Interval.

statistics for ZEPHYRUS-DIRECT and ZEPHYRUS-REFLECTIVE on ZEPHYRUSBENCH. Figure 9 details the average number of tool calls segmented by question ID.

As expected from its execution-observation-answer loop, ZEPHYRUS-REFLECTIVE makes more tool calls and uses more turns on average than ZEPHYRUS-DIRECT. Notably, despite a significant runtime error rate (6.1–7.6% of tasks encountered at least one error), both ZEPHYRUS-DIRECT and ZEPHYRUS-REFLECTIVE successfully self-corrected in all cases, with no task exceeding the 20-attempt limit.

Examining the tool use segmented by question ID, we observe that most of the tool use aligns with the expected behavior. The Geolocator is used for most of the tasks (>75% for ZEPHYRUS-REFLECTIVE and almost 90% of the tasks for ZEPHYRUS-DIRECT). For the remaining tasks, we observed that the LLM uses an approximate memorized latitude and longitude to answer questions. We find that the forecaster and climatology tools are extensively used in tasks 43, 44 and 45, which are free-form report generation tasks with no preset procedure. One notable exception is task 46, a counterfactual question answering task, where the simulator is not invoked, explaining the model's poor performance on that task. Overall, these patterns suggest that ZEPHYRUS makes reasonable tool selection decisions across most of the benchmark.

### A.4.3 ABLATION WITH EXCLUDED TOOLS

To test the effect of our different tools on model performance, we evaluated ZEPHYRUS-REFLECTIVE with a "leave-one-out" test for each tool. We also evaluated the model with only the climatology and geolocator available, as a baseline to test how well the ZEPHYRUSBENCHquestions could be answered from historical climatology. Test runs were conducted with LLM `gpt-oss-120b` on a 50% subsample of ZEPHYRUSBENCH.

The results of this ablation are largely intuitive: performance decreases when tools are removed, especially tools critical for a large number of tasks, like the forecaster or geolocator. We do observe a very slight uptick in performance on Easy and Medium tasks when the climate simulator tool is removed. To understand this phenomenon, we examined a few sample tasks where performance improved without the simulator; these are shown in Table 5 and Table 6. These are forecasting tasks, which are exceedingly difficult (0% correctness) when the forecaster tool is removed. When both the forecaster and the simulator are available, however, ZEPHYRUS-REFLECTIVE performs worse than when only the forecaster is available. This result suggests that the model sometimes reaches for the wrong tool and tries to use the climate simulator for short-term forecasting, even though tool prompt instructions specifically recommend the forecaster. This highlights the shortcoming of current LLM agents in instruction following and intelligent tool use ability, especially on long-context and reasoning tasks.

| Tools | % Correct | Valid % |
|---|---|---|
| All Tools | 33.3% | **100.0%** |
| Exclude Climatology | 33.3% | **100.0%** |
| Exclude Forecaster | 0.00% | 83.3% |
| Exclude Geolocator | 0.00% | **100.0%** |
| Exclude Simulator | **50.0%** | **100.0%** |
| Exclude Dataset, Forecaster, Simulator | 0.00% | **100.0%** |

Table 6: Tool ablation for Template 37: Difference in Forecast Area-Weighted Mean.

| Forecaster | % Correct | Easy | Medium | Hard | SAE (Median) | Location Acc | EMD | Valid % | Extreme F1 | Boolean F1 |
|---|---|---|---|---|---|---|---|---|---|---|
| Ground-Truth | 63.3% | 86.3% | 83.2% | 34.7% | 0.00 | 77.8% | 1,841 | 99.1% | 0.04 | 0.58 |
| Stormer | 61.3% | 88.7% | 60.5% | 37.8% | 0.00 | 87.9% | 1,486 | 99.3% | 0.00 | 0.61 |

Table 7: Ablation: Ground-Truth vs. Stormer forecaster on ZephyrusBench.

### A.4.4 ABLATION WITH GROUND-TRUTH FORECASTER

Table 7 contains a comparison of running ZEPHYRUS-REFLECTIVE with the default Stormer (Nguyen et al., 2024) forecaster tool against a "Ground-Truth" forecaster setting, which returns the exact ground-truth WeatherBench2 data through the tool API. The test was conducted using LLM `gpt-oss-120b` on a 50% subsample of ZEPHYRUSBENCH.

Using the Ground-Truth forecaster, model performance increases significantly (60.5% to 83.2% correctness) on Medium tasks, many of which are straightforward forecast-and-return questions. On Easy tasks, which do not require forecasting to solve correctly, performance does not improve. Most interestingly, model performance also does not improve on Hard tasks. Hard tasks, like report generation, require additional capability in order to convert accurate forecast data into a correct task answer. This result highlights the improvements in weather science reasoning and domain knowledge still necessary for LLMs to succeed thoroughly on ZEPHYRUSBENCH.

### A.4.5 VARIANCE ACROSS MULTIPLE RUNS

In order to ensure that the stochasticity in sampling from LLMs does not have a major impact on the scores of ZEPHYRUSBENCH, we ran ZEPHYRUS-REFLECTIVE three times with identical settings, using LLM `gpt-oss-120b` on a 50% random subsample of ZEPHYRUSBENCH. Results of this variance test are presented in Table 8. From this test we establish that the variance in overall benchmark performance induced by LLM sampling stochasticity is low.

### A.4.6 PERFORMANCE BY DIFFICULTY LEVEL

We report the difficulty-wise breakdown of model performance on ZEPHYRUSBENCH in Figure 10. The difficulty levels for the tasks are defined in Table 2.

| Run | % Correct | Easy | Medium | Hard | SAE (Median) | Location Acc | EMD | Valid % | Extreme F1 | Boolean F1 |
|---|---|---|---|---|---|---|---|---|---|---|
| Run 1 | 56.7% | 86.3% | 62.0% | 28.7% | 0.03 | 76.8% | 1,472 | 99.2% | 0.00 | 0.61 |
| Run 2 | 55.4% | 85.6% | 62.5% | 26.2% | 0.03 | 70.7% | 1,821 | 99.0% | 0.04 | 0.57 |
| Run 3 | 54.2% | 87.0% | 57.7% | 24.3% | 0.04 | 72.7% | 1,770 | 98.6% | 0.04 | 0.57 |
| Mean ± Std | 55.45% ± 1.00 | 86.33% ± 0.59 | 60.74% ± 2.16 | 26.40% ± 1.79 | 0.03 ± 0.00 | 73.40% ± 2.52 | 1687.62 ± 154.10 | 98.95% ± 0.23 | 0.02 ± 0.02 | 0.58 ± 0.02 |

Table 8: Per-run metrics across 3 independent runs, with cross-run variance summary (mean ± std) for ZEPHYRUS-REFLECTIVE on ZEPHYRUSBENCH.

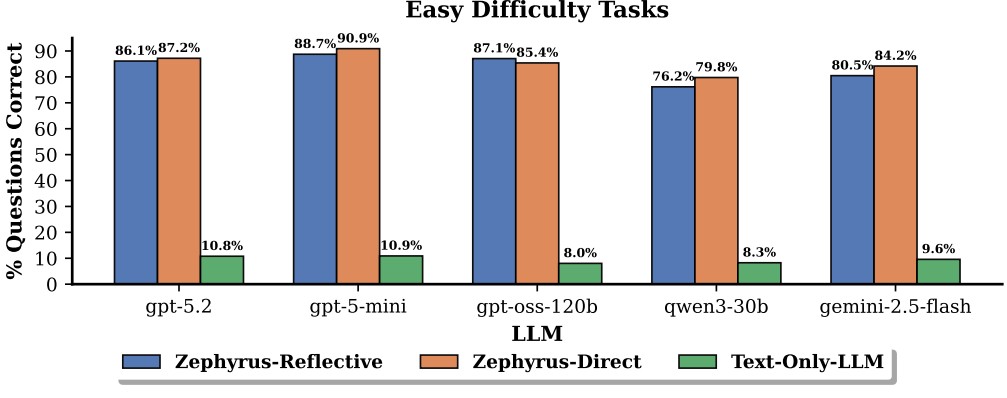

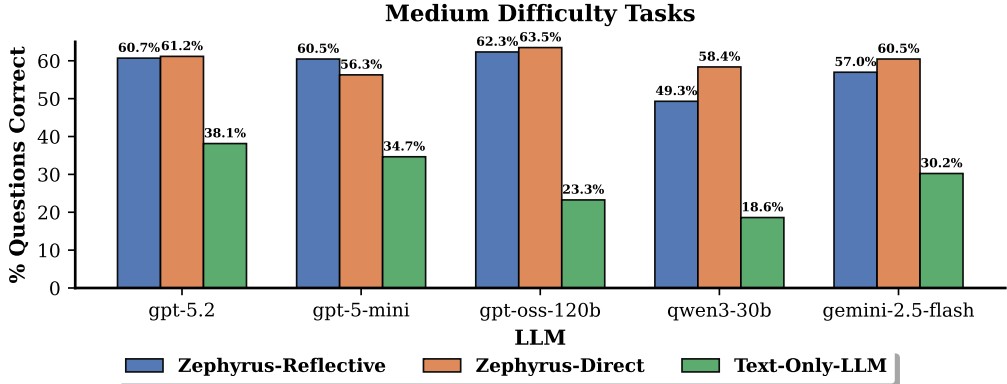

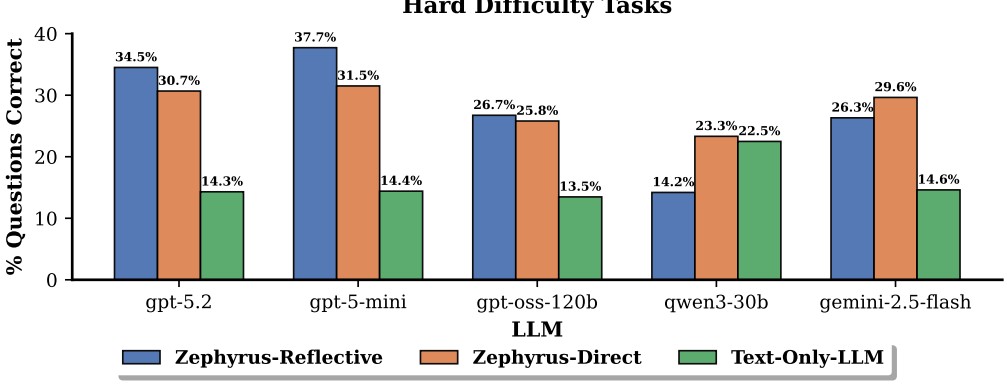

Figure 10: Performance on ZEPHYRUSBENCH segmented by difficulty level.

### A.4.7 Detailed Performance Metrics

We include detailed performance metrics from running several LLMs across all three modes on the entire ZEPHYRUSBENCH dataset in Tables 10, 11, 12 and 9.

| Model | LLM | SAE (Q25) ($\downarrow$) | SAE (Q50) ($\downarrow$) | SAE (Q75) ($\downarrow$) | SAE (Q99) ($\downarrow$) |
|---|---|---|---|---|---|
| ZEPHYRUS-REFLECTIVE | gpt-5.2 | 0.00 | 0.02 | 0.09 | **0.95** |
| ZEPHYRUS-DIRECT | gpt-5.2 | **0.00** | **0.01** | **0.08** | 472.8 |
| Text Only LLM | gpt-5.2 | 0.24 | 0.33 | 0.67 | 17.5 |
| ZEPHYRUS-REFLECTIVE | gpt-5-mini | **0.00** | 0.00 | **0.06** | 0.96 |
| ZEPHYRUS-DIRECT | gpt-5-mini | **0.00** | 0.00 | 0.07 | **0.94** |
| Text Only LLM | gpt-5-mini | 0.23 | 0.32 | 0.59 | 4.87 |
| ZEPHYRUS-REFLECTIVE | gemini-2.5-flash | **0.00** | 0.02 | 0.14 | 5.38 |
| ZEPHYRUS-DIRECT | gemini-2.5-flash | **0.00** | **0.02** | **0.09** | **1.00** |
| Text Only LLM | gemini-2.5-flash | 0.24 | 0.34 | 0.93 | 77.4 |
| ZEPHYRUS-REFLECTIVE | gpt-oss-120b | 0.00 | 0.03 | 0.12 | 1.56 |
| ZEPHYRUS-DIRECT | gpt-oss-120b | **0.00** | **0.03** | **0.11** | **1.27** |
| Text Only LLM | gpt-oss-120b | 0.24 | 0.30 | 0.37 | 9.03 |
| ZEPHYRUS-REFLECTIVE | qwen3-30b | **0.00** | **0.02** | **0.13** | **9.70** |
| ZEPHYRUS-DIRECT | qwen3-30b | 0.00 | 0.03 | 0.18 | 18.1 |
| Text Only LLM | qwen3-30b | 0.23 | 0.49 | 7.25 | 165,992 |

Table 9: Output validity and error metric quantiles for numerical tasks. SAE stands for standardized absolute error, the absolute error divided by the standard deviation of the relevant variable in the data. Q25, Q50, Q75, and Q99 respectively represent the 25th, 50th, 75th, and 99th percentile of error across all numerical dataset examples.

| Model | LLM | AE (Q25) (↓) | AE (Q50) (↓) | AE (Q75) (↓) | AE (Q99) (↓) |
|---|---|---|---|---|---|
| ZEPHYRUS-REFLECTIVE | gpt-5.2 | **0.00** | **0.00** | **12.0** | **99.7** |
| ZEPHYRUS-DIRECT | gpt-5.2 | **0.00** | **0.00** | **12.0** | **99.7** |
| Text Only LLM | gpt-5.2 | 6.00 | 18.0 | 36.0 | 135.7 |
| ZEPHYRUS-REFLECTIVE | gpt-5-mini | **0.00** | **0.00** | **3.00** | 73.4 |
| ZEPHYRUS-DIRECT | gpt-5-mini | **0.00** | **0.00** | 6.00 | **66.0** |
| Text Only LLM | gpt-5-mini | 6.00 | 12.0 | 36.0 | 103.4 |
| ZEPHYRUS-REFLECTIVE | gemini-2.5-flash | **0.00** | **0.00** | **0.00** | **105.8** |
| ZEPHYRUS-DIRECT | gemini-2.5-flash | **0.00** | **0.00** | **0.00** | 160.1 |
| Text Only LLM | gemini-2.5-flash | 6.00 | 18.0 | 42.0 | 129.1 |
| ZEPHYRUS-REFLECTIVE | gpt-oss-120b | **0.00** | **0.00** | **0.00** | 45.7 |
| ZEPHYRUS-DIRECT | gpt-oss-120b | **0.00** | **0.00** | **0.00** | **39.8** |
| Text Only LLM | gpt-oss-120b | 6.00 | 18.0 | 36.0 | 114.0 |
| ZEPHYRUS-REFLECTIVE | qwen3-30b | **0.00** | **0.00** | 6.00 | 87.6 |
| ZEPHYRUS-DIRECT | qwen3-30b | **0.00** | **0.00** | **0.00** | **60.0** |
| Text Only LLM | qwen3-30b | 6.00 | 24.0 | 54.0 | 151.4 |

Table 10: Absolute error quantiles for time tasks, in units of hours.

| Model | LLM | Location Accuracy (%)(↑) | EMD (km) (↓) | Extreme Weather F1 (%) (↑) |
|---|---|---|---|---|
| ZEPHYRUS-REFLECTIVE | gpt-5.2 | **90.4** | 1,452 | **0.00** |
| ZEPHYRUS-DIRECT | gpt-5.2 | 89.9 | **1,277** | **0.00** |
| Text Only LLM | gpt-5.2 | 18.2 | 5,445 | **0.00** |
| ZEPHYRUS-REFLECTIVE | gpt-5-mini | 87.9 | 1,486 | 0.00 |
| ZEPHYRUS-DIRECT | gpt-5-mini | **88.4** | **1,353** | **1.83** |
| Text Only LLM | gpt-5-mini | 18.2 | 5,772 | 0.00 |
| ZEPHYRUS-REFLECTIVE | gemini-2.5-flash | **76.8** | 1,651 | **3.85** |
| ZEPHYRUS-DIRECT | gemini-2.5-flash | 76.3 | 1,890 | 1.79 |
| Text Only LLM | gemini-2.5-flash | 16.2 | 3,750 | 0.00 |
| ZEPHYRUS-REFLECTIVE | gpt-oss-120b | **78.3** | **1,472** | **0.00** |
| ZEPHYRUS-DIRECT | gpt-oss-120b | 67.2 | 1,870 | **0.00** |
| Text Only LLM | gpt-oss-120b | 9.09 | 2,823 | **0.00** |
| ZEPHYRUS-REFLECTIVE | qwen3-30b | **67.2** | 1,597 | **0.00** |
| ZEPHYRUS-DIRECT | qwen3-30b | 60.6 | **1,370** | **0.00** |
| Text Only LLM | qwen3-30b | 13.1 | 3,899 | **0.00** |

Table 11: Location metrics for location answer-based questions. EMD stands for Earth Mover's Distance.

| Model | LLM | % Correct (↑) | % Valid Outputs (↑) | Discussion Score (↑) | Boolean F1 (%) (↑) |
|---|---|---|---|---|---|
| ZEPHYRUS-REFLECTIVE | gpt-5.2 | **58.9** | 98.9 | **0.27** | 58.0 |
| ZEPHYRUS-DIRECT | gpt-5.2 | 57.7 | **99.5** | 0.16 | **58.5** |
| Text Only LLM | gpt-5.2 | 17.6 | 97.5 | 0.10 | 42.5 |
| ZEPHYRUS-REFLECTIVE | gpt-5-mini | **61.2** | **99.2** | **0.25** | 60.6 |
| ZEPHYRUS-DIRECT | gpt-5-mini | 58.5 | **99.2** | 0.09 | **61.5** |
| Text Only LLM | gpt-5-mini | 17.0 | 98.6 | 0.07 | 53.0 |
| ZEPHYRUS-REFLECTIVE | gemini-2.5-flash | 52.5 | 97.2 | **0.10** | 61.8 |
| ZEPHYRUS-DIRECT | gemini-2.5-flash | **56.0** | **99.1** | 0.07 | 61.6 |
| Text Only LLM | gemini-2.5-flash | 15.7 | 90.8 | 0.06 | 24.3 |
| ZEPHYRUS-REFLECTIVE | gpt-oss-120b | **56.2** | **99.1** | **0.11** | **59.3** |
| ZEPHYRUS-DIRECT | gpt-oss-120b | 55.4 | 98.3 | 0.08 | 57.1 |
| Text Only LLM | gpt-oss-120b | 13.3 | 68.8 | 0.02 | 25.0 |
| ZEPHYRUS-REFLECTIVE | qwen3-30b | 44.2 | 83.6 | **0.06** | 48.6 |
| ZEPHYRUS-DIRECT | qwen3-30b | **51.2** | **96.3** | **0.06** | **59.8** |
| Text Only LLM | qwen3-30b | 16.4 | 93.0 | 0.03 | 8.97 |

Table 12: Overall percentage of questions correct, percentage of valid outputs, numerical score (0-1) for discussion questions, and F1 score for boolean questions.

### A.4.8 PERFORMANCE METRICS, HUMAN-GENERATED TASKS

We also include top-level performance metrics broken out specifically for the 19 human-generated tasks in ZEPHYRUSBENCH dataset. The results are reported in Tables 13, 14, 15, and 16.

| Model | LLM | SAE (Q25) (↓) | SAE (Q50) (↓) | SAE (Q75) (↓) | SAE (Q99) (↓) |
|---|---|---|---|---|---|
| ZEPHYRUS-REFLECTIVE | gpt-5.2 | 0.00 | 0.01 | 0.07 | **0.60** |
| ZEPHYRUS-DIRECT | gpt-5.2 | **0.00** | **0.00** | **0.07** | 2,164 |
| Text Only LLM | gpt-5.2 | 0.24 | 0.32 | 0.58 | 4.60 |
| ZEPHYRUS-REFLECTIVE | gpt-5-mini | **0.00** | 0.00 | **0.05** | 0.44 |
| ZEPHYRUS-DIRECT | gpt-5-mini | **0.00** | **0.00** | 0.06 | **0.42** |
| Text Only LLM | gpt-5-mini | 0.23 | 0.31 | 0.47 | 2.84 |
| ZEPHYRUS-REFLECTIVE | gemini-2.5-flash | **0.00** | 0.02 | 0.12 | 3.82 |
| ZEPHYRUS-DIRECT | gemini-2.5-flash | **0.00** | **0.01** | **0.08** | **0.56** |
| Text Only LLM | gemini-2.5-flash | 0.25 | 0.33 | 0.70 | 476.1 |
| ZEPHYRUS-REFLECTIVE | gpt-oss-120b | 0.00 | 0.02 | 0.10 | 1.34 |
| ZEPHYRUS-DIRECT | gpt-oss-120b | **0.00** | **0.02** | **0.09** | **0.98** |
| Text Only LLM | gpt-oss-120b | 0.24 | 0.30 | 0.36 | 7.38 |
| ZEPHYRUS-REFLECTIVE | qwen3-30b | **0.00** | **0.00** | **0.10** | **14.3** |
| ZEPHYRUS-DIRECT | qwen3-30b | 0.00 | 0.03 | 0.16 | 19.7 |
| Text Only LLM | qwen3-30b | 0.23 | 0.40 | 5.43 | 108,610 |

Table 13: Output validity and error metric quantiles for numerical tasks. SAE stands for standardized absolute error, the absolute error divided by the standard deviation of the relevant variable in the data. Q25, Q50, Q75, and Q99 respectively represent the 25th, 50th, 75th, and 99th percentile of error across all numerical dataset examples in human-generated tasks.

| Model | LLM | AE (Q25) (↓) | AE (Q50) (↓) | AE (Q75) (↓) | AE (Q99) (↓) |
|---|---|---|---|---|---|
| ZEPHYRUS-REFLECTIVE | gpt-5.2 | **0.00** | **0.00** | **12.0** | **99.7** |
| ZEPHYRUS-DIRECT | gpt-5.2 | **0.00** | **0.00** | **12.0** | **99.7** |
| Text Only LLM | gpt-5.2 | 6.00 | 18.0 | 36.0 | 135.7 |
| ZEPHYRUS-REFLECTIVE | gpt-5-mini | **0.00** | **0.00** | **3.00** | 73.4 |
| ZEPHYRUS-DIRECT | gpt-5-mini | **0.00** | **0.00** | 6.00 | **66.0** |
| Text Only LLM | gpt-5-mini | 6.00 | 12.0 | 36.0 | 103.4 |
| ZEPHYRUS-REFLECTIVE | gemini-2.5-flash | **0.00** | **0.00** | **0.00** | **105.8** |
| ZEPHYRUS-DIRECT | gemini-2.5-flash | **0.00** | **0.00** | **0.00** | 160.1 |
| Text Only LLM | gemini-2.5-flash | 6.00 | 18.0 | 42.0 | 129.1 |
| ZEPHYRUS-REFLECTIVE | gpt-oss-120b | **0.00** | **0.00** | **0.00** | 45.7 |
| ZEPHYRUS-DIRECT | gpt-oss-120b | **0.00** | **0.00** | **0.00** | **39.8** |
| Text Only LLM | gpt-oss-120b | 6.00 | 18.0 | 36.0 | 114.0 |
| ZEPHYRUS-REFLECTIVE | qwen3-30b | **0.00** | **0.00** | 6.00 | 87.6 |
| ZEPHYRUS-DIRECT | qwen3-30b | **0.00** | **0.00** | **0.00** | **60.0** |
| Text Only LLM | qwen3-30b | 6.00 | 24.0 | 54.0 | 151.4 |

Table 14: Absolute error quantiles for time tasks, in units of hours, on human-generated tasks in human-generated tasks.

| Model | LLM | Location Accuracy (%)(↑) | EMD (km) (↓) | Extreme Weather F1 (%) (↑) |
|---|---|---|---|---|
| ZEPHYRUS-REFLECTIVE | gpt-5.2 | **90.4** | 1,452 | **0.00** |
| ZEPHYRUS-DIRECT | gpt-5.2 | 89.9 | **1,277** | **0.00** |
| Text Only LLM | gpt-5.2 | 18.2 | 5,445 | **0.00** |
| ZEPHYRUS-REFLECTIVE | gpt-5-mini | 87.9 | 1,486 | 0.00 |
| ZEPHYRUS-DIRECT | gpt-5-mini | **88.4** | **1,353** | 1.83 |
| Text Only LLM | gpt-5-mini | 18.2 | 5,772 | 0.00 |
| ZEPHYRUS-REFLECTIVE | gemini-2.5-flash | **76.8** | 1,651 | 3.85 |
| ZEPHYRUS-DIRECT | gemini-2.5-flash | 76.3 | 1,890 | 1.79 |
| Text Only LLM | gemini-2.5-flash | 16.2 | 3,750 | 0.00 |
| ZEPHYRUS-REFLECTIVE | gpt-oss-120b | **78.3** | 1,472 | 0.00 |
| ZEPHYRUS-DIRECT | gpt-oss-120b | 67.2 | 1,870 | 0.00 |
| Text Only LLM | gpt-oss-120b | 9.09 | 2,823 | 0.00 |
| ZEPHYRUS-REFLECTIVE | qwen3-30b | **67.2** | 1,597 | 0.00 |
| ZEPHYRUS-DIRECT | qwen3-30b | 60.6 | **1,370** | 0.00 |
| Text Only LLM | qwen3-30b | 13.1 | 3,899 | 0.00 |

Table 15: Location metrics for location answer-based questions in human generated tasks. EMD stands for Earth mover's Distance.

| Model | LLM | % Correct (↑) | % Valid Outputs (↑) | Discussion Score (↑) |
|---|---|---|---|---|
| ZEPHYRUS-REFLECTIVE | gpt-5.2 | **57.5** | 98.7 | **0.27** |
| ZEPHYRUS-DIRECT | gpt-5.2 | 56.0 | **99.3** | 0.16 |
| Text Only LLM | gpt-5.2 | 8.91 | 97.3 | 0.10 |
| ZEPHYRUS-REFLECTIVE | gpt-5-mini | **60.4** | 99.0 | **0.25** |
| ZEPHYRUS-DIRECT | gpt-5-mini | 57.1 | **99.1** | 0.09 |
| Text Only LLM | gpt-5-mini | 8.56 | 98.5 | 0.07 |
| ZEPHYRUS-REFLECTIVE | gemini-2.5-flash | 50.9 | 97.8 | **0.10** |
| ZEPHYRUS-DIRECT | gemini-2.5-flash | **54.1** | **98.9** | 0.07 |
| Text Only LLM | gemini-2.5-flash | 8.28 | 89.1 | 0.06 |
| ZEPHYRUS-REFLECTIVE | gpt-oss-120b | **54.5** | **98.9** | **0.11** |
| ZEPHYRUS-DIRECT | gpt-oss-120b | 53.1 | 97.8 | 0.08 |
| Text Only LLM | gpt-oss-120b | 6.61 | 67.7 | 0.02 |
| ZEPHYRUS-REFLECTIVE | qwen3-30b | 42.1 | 82.0 | **0.06** |
| ZEPHYRUS-DIRECT | qwen3-30b | **48.4** | **95.9** | 0.06 |
| Text Only LLM | qwen3-30b | 11.4 | 94.9 | 0.03 |

Table 16: Overall percentage of questions correct, percentage of valid outputs, and numerical score (0-1) for discussion questions in human-generated tasks.

### A.4.9 Performance Metrics, Semi-Synthetic Tasks

We also include top-level performance metrics broken out specifically for the 30 semi-synthetic (LLM-generated, human-validated) tasks in ZEPHYRUSBENCH dataset. The results are reported in Tables 17 and 18.

| Model | LLM | SAE (Q25) ($\downarrow$) | SAE (Q50) ($\downarrow$) | SAE (Q75) ($\downarrow$) | SAE (Q99) ($\downarrow$) |
|---|---|---|---|---|---|
| ZEPHYRUS-REFLECTIVE | gpt-5.2 | **0.02** | **0.07** | **0.30** | 3.24 |
| ZEPHYRUS-DIRECT | gpt-5.2 | **0.02** | **0.07** | 0.30 | **2.68** |
| Text Only LLM | gpt-5.2 | 0.25 | 0.65 | 1.62 | 333.6 |
| ZEPHYRUS-REFLECTIVE | gpt-5-mini | **0.02** | **0.07** | 0.26 | **1.32** |
| ZEPHYRUS-DIRECT | gpt-5-mini | **0.02** | **0.07** | 0.24 | 1.41 |
| Text Only LLM | gpt-5-mini | 0.27 | 0.78 | 1.68 | 24.3 |
| ZEPHYRUS-REFLECTIVE | gemini-2.5-flash | 0.03 | 0.10 | 0.34 | 9.47 |
| ZEPHYRUS-DIRECT | gemini-2.5-flash | **0.02** | **0.08** | **0.29** | **1.31** |
| Text Only LLM | gemini-2.5-flash | 0.23 | 0.85 | 2.09 | 31.0 |
| ZEPHYRUS-REFLECTIVE | gpt-oss-120b | **0.02** | **0.07** | **0.29** | **1.89** |
| ZEPHYRUS-DIRECT | gpt-oss-120b | **0.02** | **0.07** | **0.29** | 2.61 |
| Text Only LLM | gpt-oss-120b | 0.28 | 1.02 | 1.92 | 16.2 |
| ZEPHYRUS-REFLECTIVE | qwen3-30b | **0.02** | **0.07** | **0.29** | **2.34** |
| ZEPHYRUS-DIRECT | qwen3-30b | **0.02** | 0.11 | 0.36 | 4.36 |
| Text Only LLM | qwen3-30b | 0.40 | 2.13 | 17.1 | 3.39e+06 |

Table 17: Output validity and error metric quantiles for semi-synthetically generated numerical tasks. SAE stands for standardized absolute error, the absolute error divided by the standard deviation of the relevant variable in the data. Q25, Q50, Q75, and Q99 respectively represent the 25th, 50th, 75th, and 99th percentile of error across all numerical dataset examples.

| Model | LLM | % Correct ($\uparrow$) | % Valid Outputs ($\uparrow$) | Boolean F1 (%) ($\uparrow$) |
|---|---|---|---|---|
| ZEPHYRUS-REFLECTIVE | gpt-5.2 | 63.7 | 99.8 | 58.0 |
| ZEPHYRUS-DIRECT | gpt-5.2 | **63.9** | **100.0** | **58.5** |
| Text Only LLM | gpt-5.2 | 48.4 | 98.2 | 42.5 |
| ZEPHYRUS-REFLECTIVE | gpt-5-mini | **64.1** | **99.8** | 60.6 |
| ZEPHYRUS-DIRECT | gpt-5-mini | 63.7 | 99.4 | **61.5** |
| Text Only LLM | gpt-5-mini | 46.9 | 99.0 | 53.0 |
| ZEPHYRUS-REFLECTIVE | gemini-2.5-flash | 58.2 | 94.9 | **61.8** |
| ZEPHYRUS-DIRECT | gemini-2.5-flash | **62.9** | **99.8** | 61.6 |
| Text Only LLM | gemini-2.5-flash | 42.2 | 96.7 | 24.3 |
| ZEPHYRUS-REFLECTIVE | gpt-oss-120b | 62.2 | **100.0** | **59.3** |
| ZEPHYRUS-DIRECT | gpt-oss-120b | **63.5** | **100.0** | 57.1 |
| Text Only LLM | gpt-oss-120b | 37.1 | 72.7 | 25.0 |
| ZEPHYRUS-REFLECTIVE | qwen3-30b | 51.4 | 89.4 | 48.6 |
| ZEPHYRUS-DIRECT | qwen3-30b | **61.2** | **98.0** | **59.8** |
| Text Only LLM | qwen3-30b | 34.1 | 86.5 | 8.97 |

Table 18: Overall percentage of questions correct, percentage of valid outputs, numerical score (0-1) for discussion questions, and F1 score for boolean questions on the semi-synthetically generated subset of tasks.

### A.4.10 PERFORMANCE METRICS BY TASK

At the most granular level of detail, we provide performance metrics of all models' performance on each unique template ID in ZEPHYRUSBENCH. The reported performance metrics are tailored individually to each question type. These results are reported in Tables 19 through 67.

| Model | LLM | Location Accuracy (%)(↑) | EMD (km)(↓) |
|---|---|---|---|
| ZEPHYRUS-REFLECTIVE | gpt-5-mini | 96.8 | 179.8 |
| ZEPHYRUS-DIRECT | gpt-5-mini | **100.0** | **0.00** |
| Text Only LLM | gpt-5-mini | 17.0 | 8,551 |
| ZEPHYRUS-REFLECTIVE | gemini-2.5-flash | 83.0 | 1,132 |
| ZEPHYRUS-DIRECT | gemini-2.5-flash | **84.0** | **1,126** |
| Text Only LLM | gemini-2.5-flash | 18.1 | 6,372 |
| ZEPHYRUS-REFLECTIVE | gpt-oss-120b | 83.0 | 911.2 |
| ZEPHYRUS-DIRECT | gpt-oss-120b | **88.3** | **889.3** |
| Text Only LLM | gpt-oss-120b | 13.8 | 6,954 |
| ZEPHYRUS-REFLECTIVE | gpt-5.2 | **100.0** | **0.00** |
| ZEPHYRUS-DIRECT | gpt-5.2 | **100.0** | **0.00** |
| Text Only LLM | gpt-5.2 | 17.0 | 7,968 |
| ZEPHYRUS-REFLECTIVE | qwen3-30b | 76.6 | 1,502 |
| ZEPHYRUS-DIRECT | qwen3-30b | **89.4** | **855.3** |
| Text Only LLM | qwen3-30b | 16.0 | 7,183 |

Table 19: Location prediction metrics for Template ID 1: `Which geographic feature experienced the highest/lowest average value of a weather variable`

| Model | LLM | SAE (Q25)(↓) | SAE (Q50)(↓) | SAE (Q75)(↓) | SAE (Q99)(↓) | Err Std(↓) |
|---|---|---|---|---|---|---|
| ZEPHYRUS-REFLECTIVE | gpt-5-mini | **0.00** | **0.00** | **0.00** | 0.13 | 20.7 |
| ZEPHYRUS-DIRECT | gpt-5-mini | **0.00** | **0.00** | **0.00** | **0.06** | **5.34** |
| Text Only LLM | gpt-5-mini | 0.16 | 0.40 | 0.77 | 3.24 | 1,390 |
| ZEPHYRUS-REFLECTIVE | gemini-2.5-flash | **0.00** | **0.00** | 0.01 | 0.41 | **413.0** |
| ZEPHYRUS-DIRECT | gemini-2.5-flash | **0.00** | **0.00** | **0.00** | **0.39** | 422.0 |
| Text Only LLM | gemini-2.5-flash | 0.26 | 0.72 | 2.87 | 75.8 | 30,300 |
| ZEPHYRUS-REFLECTIVE | gpt-oss-120b | **0.00** | **0.00** | 0.01 | **0.47** | **413.0** |
| ZEPHYRUS-DIRECT | gpt-oss-120b | **0.00** | **0.00** | **0.00** | **0.47** | 460.0 |
| Text Only LLM | gpt-oss-120b | 0.12 | 0.46 | 1.15 | 116.0 | 2,330 |
| ZEPHYRUS-REFLECTIVE | gpt-5.2 | **0.00** | **0.00** | **0.00** | **0.13** | **376.0** |
| ZEPHYRUS-DIRECT | gpt-5.2 | **0.00** | **0.00** | **0.00** | 0.39 | 562.0 |
| Text Only LLM | gpt-5.2 | 0.20 | 0.56 | 1.34 | 13.5 | 21,000 |
| ZEPHYRUS-REFLECTIVE | qwen3-30b | **0.00** | **0.00** | 0.03 | 0.89 | 391.0 |
| ZEPHYRUS-DIRECT | qwen3-30b | **0.00** | **0.00** | **0.02** | **0.48** | **271.0** |
| Text Only LLM | qwen3-30b | 0.26 | 0.92 | 6.86 | 76.2 | 39,000 |

Table 20: Standardized Absolute Error (SAE) quantiles for Template ID 2: `What is the min/max/average/median value of a weather variable at a specific location`

| Model | LLM | Location Accuracy (%)(↑) | EMD (km)(↓) |
|---|---|---|---|
| ZEPHYRUS-REFLECTIVE | gpt-5-mini | **79.8** | **472.1** |
| ZEPHYRUS-DIRECT | gpt-5-mini | 77.9 | 784.8 |
| Text Only LLM | gpt-5-mini | 19.2 | 3,611 |
| ZEPHYRUS-REFLECTIVE | gemini-2.5-flash | **71.2** | **1,224** |
| ZEPHYRUS-DIRECT | gemini-2.5-flash | 69.2 | 1,442 |
| Text Only LLM | gemini-2.5-flash | 14.4 | 2,962 |
| ZEPHYRUS-REFLECTIVE | gpt-oss-120b | **74.0** | **842.2** |
| ZEPHYRUS-DIRECT | gpt-oss-120b | 48.1 | 2,290 |
| Text Only LLM | gpt-oss-120b | 4.80 | 3,290 |
| ZEPHYRUS-REFLECTIVE | gpt-5.2 | **81.7** | 403.6 |
| ZEPHYRUS-DIRECT | gpt-5.2 | 80.8 | **211.2** |
| Text Only LLM | gpt-5.2 | 19.2 | 3,196 |
| ZEPHYRUS-REFLECTIVE | qwen3-30b | **58.7** | **2,307** |
| ZEPHYRUS-DIRECT | qwen3-30b | 34.6 | 2,656 |
| Text Only LLM | qwen3-30b | 10.6 | 3,312 |

Table 21: Location prediction metrics for Template ID 3: `Which sublocation has the highest/lowest recorded variable value`

| Model | LLM | AE (Q25)(↓) | AE (Q50)(↓) | AE (Q75)(↓) | AE (Q99)(↓) |
|---|---|---|---|---|---|
| ZEPHYRUS-REFLECTIVE | gpt-5-mini | **0.00** | **0.00** | **0.00** | 82.9 |
| ZEPHYRUS-DIRECT | gpt-5-mini | **0.00** | **0.00** | **0.00** | **73.0** |
| Text Only LLM | gpt-5-mini | 6.00 | 18.0 | 42.0 | 109.0 |
| ZEPHYRUS-REFLECTIVE | gemini-2.5-flash | **0.00** | **0.00** | **0.00** | **125.0** |
| ZEPHYRUS-DIRECT | gemini-2.5-flash | **0.00** | **0.00** | **0.00** | 200.0 |
| Text Only LLM | gemini-2.5-flash | 6.00 | 24.0 | 48.0 | 138.0 |
| ZEPHYRUS-REFLECTIVE | gpt-oss-120b | **0.00** | **0.00** | **0.00** | 48.0 |
| ZEPHYRUS-DIRECT | gpt-oss-120b | **0.00** | **0.00** | **0.00** | **37.2** |
| Text Only LLM | gpt-oss-120b | 6.00 | 18.0 | 48.0 | 119.0 |
| ZEPHYRUS-REFLECTIVE | gpt-5.2 | **0.00** | **0.00** | **12.0** | **103.0** |
| ZEPHYRUS-DIRECT | gpt-5.2 | **0.00** | **0.00** | **12.0** | 105.0 |
| Text Only LLM | gpt-5.2 | 6.00 | 24.0 | 48.0 | 138.0 |
| ZEPHYRUS-REFLECTIVE | qwen3-30b | **0.00** | **0.00** | **0.00** | 90.0 |
| ZEPHYRUS-DIRECT | qwen3-30b | **0.00** | **0.00** | **0.00** | **60.7** |
| Text Only LLM | qwen3-30b | 12.0 | 30.0 | 72.0 | 156.0 |

Table 22: Absolute Error (AE) quantiles for Template ID 4: `How many hours from start did a location experience extremum`

| Model | LLM | SAE (Q25)(↓) | SAE (Q50)(↓) | SAE (Q75)(↓) | SAE (Q99)(↓) | Err Std(↓) |
|---|---|---|---|---|---|---|
| ZEPHYRUS-REFLECTIVE | gpt-5-mini | **0.00** | **0.00** | **0.00** | 1.34 | 553.0 |
| ZEPHYRUS-DIRECT | gpt-5-mini | **0.00** | **0.00** | **0.00** | 0.36 | **551.0** |
| Text Only LLM | gpt-5-mini | 0.19 | 0.45 | 0.90 | 4.09 | 2,030 |
| ZEPHYRUS-REFLECTIVE | gemini-2.5-flash | **0.00** | **0.00** | **0.00** | 2.41 | **634.0** |
| ZEPHYRUS-DIRECT | gemini-2.5-flash | **0.00** | **0.00** | **0.00** | 0.42 | 2,890 |
| Text Only LLM | gemini-2.5-flash | 0.31 | 1.10 | 3.22 | 5,440 | 21,500 |
| ZEPHYRUS-REFLECTIVE | gpt-oss-120b | **0.00** | **0.00** | 0.01 | 1.46 | 6,970 |
| ZEPHYRUS-DIRECT | gpt-oss-120b | **0.00** | **0.00** | 0.01 | 0.37 | **343.0** |
| Text Only LLM | gpt-oss-120b | 0.15 | 0.45 | 1.28 | 269.0 | 2,670 |
| ZEPHYRUS-REFLECTIVE | gpt-5.2 | **0.00** | **0.00** | **0.00** | 0.39 | **232.0** |
| ZEPHYRUS-DIRECT | gpt-5.2 | **0.00** | **0.00** | **0.00** | 0.36 | 232.0 |
| Text Only LLM | gpt-5.2 | 0.18 | 0.51 | 1.08 | 7.14 | 2,530 |
| ZEPHYRUS-REFLECTIVE | qwen3-30b | **0.00** | **0.00** | 0.01 | 1.60 | **2,060** |
| ZEPHYRUS-DIRECT | qwen3-30b | **0.00** | **0.00** | **0.00** | 0.83 | 2,900 |
| Text Only LLM | qwen3-30b | 0.82 | 5.08 | 15.1 | 173,000 | 38,700 |

Table 23: Standardized Absolute Error (SAE) quantiles for Template ID 5: `What is the weather variable value at a location at a specific time`

| Model | LLM | Correctness (%)(↑) | EMD (km)(↓) | IoU (%)(↑) |
|---|---|---|---|---|
| ZEPHYRUS-REFLECTIVE | gpt-5-mini | **100.0** | **0.00** | **100.0** |
| ZEPHYRUS-DIRECT | gpt-5-mini | **100.0** | **0.00** | **100.0** |
| Text Only LLM | gpt-5-mini | 20.0 | 7,580 | 0.00 |
| ZEPHYRUS-REFLECTIVE | gemini-2.5-flash | **82.2** | 1,243 | **100.0** |
| ZEPHYRUS-DIRECT | gemini-2.5-flash | 73.3 | **832.3** | **100.0** |
| Text Only LLM | gemini-2.5-flash | 37.8 | 10,864 | 0.00 |
| ZEPHYRUS-REFLECTIVE | gpt-oss-120b | **100.0** | **0.00** | **100.0** |
| ZEPHYRUS-DIRECT | gpt-oss-120b | **100.0** | **0.00** | **100.0** |
| Text Only LLM | gpt-oss-120b | 57.8 | 10,270 | **100.0** |
| ZEPHYRUS-REFLECTIVE | gpt-5.2 | 95.6 | **558.7** | **100.0** |
| ZEPHYRUS-DIRECT | gpt-5.2 | **97.8** | 565.0 | **100.0** |
| Text Only LLM | gpt-5.2 | 15.6 | 7,750 | 0.00 |
| ZEPHYRUS-REFLECTIVE | qwen3-30b | 82.2 | 887.7 | **100.0** |
| ZEPHYRUS-DIRECT | qwen3-30b | **88.9** | **419.8** | **100.0** |
| Text Only LLM | qwen3-30b | 46.7 | 10,569 | 0.00 |

Table 24: Extreme weather detection metrics for Template ID 6: `Identify geofeatures where variable differs from mean by N standard deviations`

| Model | LLM | Correctness (%)(↑) | EMD (km)(↓) | IoU (%)(↑) |
|---|---|---|---|---|
| ZEPHYRUS-REFLECTIVE | gpt-5-mini | 95.8 | 9.30 | **100.0** |
| ZEPHYRUS-DIRECT | gpt-5-mini | **100.0** | **0.00** | **100.0** |
| Text Only LLM | gpt-5-mini | 0.00 | 5,519 | 2.30 |
| ZEPHYRUS-REFLECTIVE | gemini-2.5-flash | 79.2 | **225.4** | **100.0** |
| ZEPHYRUS-DIRECT | gemini-2.5-flash | **81.2** | 227.5 | **100.0** |
| Text Only LLM | gemini-2.5-flash | 4.20 | 5,609 | 0.00 |
| ZEPHYRUS-REFLECTIVE | gpt-oss-120b | **100.0** | 0.70 | **100.0** |
| ZEPHYRUS-DIRECT | gpt-oss-120b | 95.8 | **0.00** | **100.0** |
| Text Only LLM | gpt-oss-120b | 4.20 | 6,137 | 0.00 |
| ZEPHYRUS-REFLECTIVE | gpt-5.2 | **97.9** | 74.7 | **100.0** |
| ZEPHYRUS-DIRECT | gpt-5.2 | **97.9** | **11.3** | **100.0** |
| Text Only LLM | gpt-5.2 | 0.00 | 5,750 | 1.50 |
| ZEPHYRUS-REFLECTIVE | qwen3-30b | 83.3 | 292.6 | **100.0** |
| ZEPHYRUS-DIRECT | qwen3-30b | **87.5** | **62.6** | **100.0** |
| Text Only LLM | qwen3-30b | 2.10 | 7,287 | 0.00 |

Table 25: Extreme weather detection metrics for Template ID 7: `Identify geofeatures where variable lies outside climatological percentile envelope`

| Model | LLM | SAE (Q25)(↓) | SAE (Q50)(↓) | SAE (Q75)(↓) | SAE (Q99)(↓) | Err Std(↓) |
|---|---|---|---|---|---|---|
| ZEPHYRUS-REFLECTIVE | gpt-5-mini | **0.01** | **0.06** | **0.12** | **0.67** | **173.0** |
| ZEPHYRUS-DIRECT | gpt-5-mini | 0.04 | 0.07 | 0.17 | 1.36 | 180.0 |
| Text Only LLM | gpt-5-mini | 0.22 | 0.46 | 0.97 | 3.76 | 257.0 |
| ZEPHYRUS-REFLECTIVE | gemini-2.5-flash | **0.01** | **0.05** | 0.10 | 2.12 | **31.1** |
| ZEPHYRUS-DIRECT | gemini-2.5-flash | **0.01** | **0.05** | **0.09** | **0.37** | 66.2 |
| Text Only LLM | gemini-2.5-flash | 0.31 | 0.82 | 2.15 | 58.4 | 20,100 |
| ZEPHYRUS-REFLECTIVE | gpt-oss-120b | **0.01** | **0.06** | **0.10** | **0.67** | 182.0 |
| ZEPHYRUS-DIRECT | gpt-oss-120b | **0.01** | **0.06** | 0.11 | **0.67** | 186.0 |
| Text Only LLM | gpt-oss-120b | 0.72 | 0.95 | 1.54 | 2.12 | 1,810 |
| ZEPHYRUS-REFLECTIVE | gpt-5.2 | **0.01** | **0.05** | **0.10** | **0.67** | 172.0 |
| ZEPHYRUS-DIRECT | gpt-5.2 | **0.01** | 0.06 | 0.11 | **0.67** | 173.0 |
| Text Only LLM | gpt-5.2 | 0.28 | 0.66 | 1.67 | 3.22 | 432.0 |
| ZEPHYRUS-REFLECTIVE | qwen3-30b | **0.00** | **0.03** | **0.09** | **0.54** | 205.0 |
| ZEPHYRUS-DIRECT | qwen3-30b | **0.00** | 0.05 | **0.09** | 0.56 | **188.0** |
| Text Only LLM | qwen3-30b | 1.54 | 12.2 | 24.3 | 726,000 | 28,100 |

Table 26: Standardized Absolute Error (SAE) quantiles for Template ID 8: `What will the variable be at a location after time interval (forecast)`

| Model | LLM | AE (Q25)(↓) | AE (Q50)(↓) | AE (Q75)(↓) | AE (Q99)(↓) |
|---|---|---|---|---|---|
| ZEPHYRUS-REFLECTIVE | gpt-5-mini | **0.00** | **0.00** | 6.00 | 39.2 |
| ZEPHYRUS-DIRECT | gpt-5-mini | **0.00** | 6.00 | 18.0 | 39.3 |
| Text Only LLM | gpt-5-mini | 6.00 | 6.00 | 12.0 | **24.5** |
| ZEPHYRUS-REFLECTIVE | gemini-2.5-flash | **0.00** | **0.00** | 6.00 | **39.2** |
| ZEPHYRUS-DIRECT | gemini-2.5-flash | **0.00** | **0.00** | **4.50** | **39.2** |
| Text Only LLM | gemini-2.5-flash | 1.00 | 6.00 | 15.0 | **39.2** |
| ZEPHYRUS-REFLECTIVE | gpt-oss-120b | **0.00** | **0.00** | **0.00** | **39.2** |
| ZEPHYRUS-DIRECT | gpt-oss-120b | **0.00** | **0.00** | **0.00** | **39.2** |
| Text Only LLM | gpt-oss-120b | **0.00** | 12.0 | 18.0 | 40.5 |
| ZEPHYRUS-REFLECTIVE | gpt-5.2 | **0.00** | **0.00** | 12.0 | 39.2 |
| ZEPHYRUS-DIRECT | gpt-5.2 | **0.00** | **0.00** | 6.00 | 39.2 |
| Text Only LLM | gpt-5.2 | **0.00** | 6.00 | 12.0 | **21.2** |
| ZEPHYRUS-REFLECTIVE | qwen3-30b | **0.00** | 6.00 | 21.0 | **39.7** |
| ZEPHYRUS-DIRECT | qwen3-30b | **0.00** | **0.00** | **12.0** | 42.0 |
| Text Only LLM | qwen3-30b | 6.00 | 12.0 | 18.0 | 45.2 |

Table 27: Absolute Error (AE) quantiles for Template ID 9: `When will location experience its extremum in future period (forecast)`

| Model | LLM | Correctness (%)(↑) | EMD (km)(↓) | IoU (%)(↑) |
|---|---|---|---|---|
| ZEPHYRUS-REFLECTIVE | gpt-5-mini | **13.7** | 3,839 | 40.0 |
| ZEPHYRUS-DIRECT | gpt-5-mini | **13.7** | **3,210** | **45.5** |
| Text Only LLM | gpt-5-mini | 7.80 | 6,732 | 0.00 |
| ZEPHYRUS-REFLECTIVE | gemini-2.5-flash | 11.8 | **2,520** | **37.1** |
| ZEPHYRUS-DIRECT | gemini-2.5-flash | **15.7** | 2,693 | **37.1** |
| Text Only LLM | gemini-2.5-flash | 9.80 | 8,383 | 0.00 |
| ZEPHYRUS-REFLECTIVE | gpt-oss-120b | **15.7** | **2,772** | 38.9 |
| ZEPHYRUS-DIRECT | gpt-oss-120b | 13.7 | 3,087 | **42.6** |
| Text Only LLM | gpt-oss-120b | 11.8 | – | 0.00 |
| ZEPHYRUS-REFLECTIVE | gpt-5.2 | **13.7** | 4,644 | 34.4 |
| ZEPHYRUS-DIRECT | gpt-5.2 | **13.7** | **4,031** | **35.2** |
| Text Only LLM | gpt-5.2 | 11.8 | 6,561 | 0.00 |
| ZEPHYRUS-REFLECTIVE | qwen3-30b | 11.8 | 3,582 | 5.00 |
| ZEPHYRUS-DIRECT | qwen3-30b | **13.7** | **3,359** | **17.1** |
| Text Only LLM | qwen3-30b | 9.80 | 8,215 | 0.00 |

Table 28: Extreme weather detection metrics for Template ID 10: `Which geofeatures are experiencing exceedance in variable values`

| Model | LLM | Correctness (%)(↑) | EMD (km)(↓) | IoU (%)(↑) |
|---|---|---|---|---|
| ZEPHYRUS-REFLECTIVE | gpt-5-mini | 17.1 | 9,422 | **0.00** |
| ZEPHYRUS-DIRECT | gpt-5-mini | 20.0 | **8,410** | **0.00** |
| Text Only LLM | gpt-5-mini | **28.6** | – | **0.00** |
| ZEPHYRUS-REFLECTIVE | gemini-2.5-flash | 22.9 | **5,761** | **0.00** |
| ZEPHYRUS-DIRECT | gemini-2.5-flash | 17.1 | 8,696 | **0.00** |
| Text Only LLM | gemini-2.5-flash | **28.6** | – | **0.00** |
| ZEPHYRUS-REFLECTIVE | gpt-oss-120b | 14.3 | **7,586** | **0.00** |
| ZEPHYRUS-DIRECT | gpt-oss-120b | 25.7 | 8,409 | **0.00** |
| Text Only LLM | gpt-oss-120b | **28.6** | – | **0.00** |
| ZEPHYRUS-REFLECTIVE | gpt-5.2 | 17.1 | **10,402** | **0.00** |
| ZEPHYRUS-DIRECT | gpt-5.2 | **28.6** | 11,612 | **0.00** |
| Text Only LLM | gpt-5.2 | **28.6** | – | **0.00** |
| ZEPHYRUS-REFLECTIVE | qwen3-30b | **28.6** | 10,114 | **0.00** |
| ZEPHYRUS-DIRECT | qwen3-30b | 25.7 | 11,073 | **0.00** |
| Text Only LLM | qwen3-30b | **28.6** | – | **0.00** |

Table 29: Extreme weather detection metrics for Template ID 11: `Identify extreme weather events that will occur in the next N hours (forecast)`

| Model | LLM | Correctness (%)(↑) | EMD (km)(↓) | IoU (%)(↑) |
|---|---|---|---|---|
| ZEPHYRUS-REFLECTIVE | gpt-5-mini | 8.30 | **6,912** | **0.00** |
| ZEPHYRUS-DIRECT | gpt-5-mini | 8.30 | 7,871 | **0.00** |
| Text Only LLM | gpt-5-mini | **16.7** | – | **0.00** |
| ZEPHYRUS-REFLECTIVE | gemini-2.5-flash | 14.6 | 7,462 | **0.00** |
| ZEPHYRUS-DIRECT | gemini-2.5-flash | 4.20 | **6,824** | **0.00** |
| Text Only LLM | gemini-2.5-flash | **16.7** | – | **0.00** |
| ZEPHYRUS-REFLECTIVE | gpt-oss-120b | 8.30 | 9,589 | **0.00** |
| ZEPHYRUS-DIRECT | gpt-oss-120b | 6.20 | **8,357** | **0.00** |
| Text Only LLM | gpt-oss-120b | **16.7** | – | **0.00** |
| ZEPHYRUS-REFLECTIVE | gpt-5.2 | 12.5 | **7,556** | **0.00** |
| ZEPHYRUS-DIRECT | gpt-5.2 | 10.4 | 9,214 | **0.00** |
| Text Only LLM | gpt-5.2 | **16.7** | – | **0.00** |
| ZEPHYRUS-REFLECTIVE | qwen3-30b | **16.7** | 8,183 | **0.00** |
| ZEPHYRUS-DIRECT | qwen3-30b | **16.7** | **7,785** | **0.00** |
| Text Only LLM | qwen3-30b | **16.7** | – | **0.00** |

Table 30: Extreme weather detection metrics for Template ID 12: `Check if extreme weather events are currently happening`

| Model | LLM | Correctness (%)(↑) | EMD (km)(↓) | IoU (%)(↑) |
|---|---|---|---|---|
| ZEPHYRUS-REFLECTIVE | gpt-5-mini | **100.0** | **0.00** | 100.0 |
| ZEPHYRUS-DIRECT | gpt-5-mini | **100.0** | **0.00** | 100.0 |
| Text Only LLM | gpt-5-mini | 4.80 | 6,248 | 2.00 |
| ZEPHYRUS-REFLECTIVE | gemini-2.5-flash | 71.4 | 1,136 | **100.0** |
| ZEPHYRUS-DIRECT | gemini-2.5-flash | **85.7** | **559.8** | **100.0** |
| Text Only LLM | gemini-2.5-flash | 4.80 | 7,554 | 0.00 |
| ZEPHYRUS-REFLECTIVE | gpt-oss-120b | 97.6 | 0.10 | **100.0** |
| ZEPHYRUS-DIRECT | gpt-oss-120b | **100.0** | **0.00** | **100.0** |
| Text Only LLM | gpt-oss-120b | 9.50 | 5,879 | 0.00 |
| ZEPHYRUS-REFLECTIVE | gpt-5.2 | **100.0** | **0.00** | 100.0 |
| ZEPHYRUS-DIRECT | gpt-5.2 | **100.0** | **0.00** | 100.0 |
| Text Only LLM | gpt-5.2 | 2.40 | 6,656 | 0.00 |
| ZEPHYRUS-REFLECTIVE | qwen3-30b | 76.2 | 598.4 | **100.0** |
| ZEPHYRUS-DIRECT | qwen3-30b | **97.6** | **0.00** | **100.0** |
| Text Only LLM | qwen3-30b | 4.80 | 6,763 | 0.00 |

Table 31: Extreme weather detection metrics for Template ID 13: `Which geographic features experienced unusual weather anomalies compared to baseline`

| Model | LLM | F1 Score(↑) | Precision (%)(↑) | Recall (%)(↑) |
|---|---|---|---|---|
| ZEPHYRUS-REFLECTIVE | gpt-5-mini | **93.3** | **100.0** | **87.5** |
| ZEPHYRUS-DIRECT | gpt-5-mini | **93.3** | **100.0** | **87.5** |
| Text Only LLM | gpt-5-mini | 61.5 | 80.0 | 50.0 |
| ZEPHYRUS-REFLECTIVE | gemini-2.5-flash | **93.3** | **100.0** | **87.5** |
| ZEPHYRUS-DIRECT | gemini-2.5-flash | 76.9 | **100.0** | 62.5 |
| Text Only LLM | gemini-2.5-flash | 50.0 | 75.0 | 37.5 |
| ZEPHYRUS-REFLECTIVE | gpt-oss-120b | **93.3** | **100.0** | **87.5** |
| ZEPHYRUS-DIRECT | gpt-oss-120b | 76.9 | **100.0** | 62.5 |
| Text Only LLM | gpt-oss-120b | 57.1 | **100.0** | 40.0 |
| ZEPHYRUS-REFLECTIVE | gpt-5.2 | 76.9 | **100.0** | 62.5 |
| ZEPHYRUS-DIRECT | gpt-5.2 | **93.3** | **100.0** | **87.5** |
| Text Only LLM | gpt-5.2 | 76.9 | **100.0** | 62.5 |
| ZEPHYRUS-REFLECTIVE | qwen3-30b | 93.3 | 100.0 | 87.5 |
| ZEPHYRUS-DIRECT | qwen3-30b | **94.1** | 88.9 | **100.0** |
| Text Only LLM | qwen3-30b | 75.0 | 100.0 | 60.0 |

Table 32: Boolean classification metrics for Template ID 14: `Does maximum weather variable occur at same or adjacent grid point as another variable (forecast)`

| Model | LLM | F1 Score(↑) | Precision (%)(↑) | Recall (%)(↑) |
|---|---|---|---|---|
| ZEPHYRUS-REFLECTIVE | gpt-5-mini | **85.7** | **75.0** | **100.0** |
| ZEPHYRUS-DIRECT | gpt-5-mini | **85.7** | **75.0** | **100.0** |
| Text Only LLM | gpt-5-mini | 0.00 | 0.00 | 0.00 |
| ZEPHYRUS-REFLECTIVE | gemini-2.5-flash | **85.7** | **75.0** | **100.0** |
| ZEPHYRUS-DIRECT | gemini-2.5-flash | **85.7** | **75.0** | **100.0** |
| Text Only LLM | gemini-2.5-flash | 0.00 | 0.00 | 0.00 |
| ZEPHYRUS-REFLECTIVE | gpt-oss-120b | **85.7** | **75.0** | **100.0** |
| ZEPHYRUS-DIRECT | gpt-oss-120b | **85.7** | **75.0** | **100.0** |
| Text Only LLM | gpt-oss-120b | 50.0 | 50.0 | 50.0 |
| ZEPHYRUS-REFLECTIVE | gpt-5.2 | **85.7** | **75.0** | **100.0** |
| ZEPHYRUS-DIRECT | gpt-5.2 | **85.7** | **75.0** | **100.0** |
| Text Only LLM | gpt-5.2 | 66.7 | 50.0 | 100.0 |
| ZEPHYRUS-REFLECTIVE | qwen3-30b | 85.7 | 75.0 | 100.0 |
| ZEPHYRUS-DIRECT | qwen3-30b | **100.0** | **100.0** | 100.0 |
| Text Only LLM | qwen3-30b | 0.00 | 0.00 | 0.00 |

Table 33: Boolean classification metrics for Template ID 15: `Does maximum weather variable in region remain lower than future maximum (forecast)`

| Model | LLM | F1 Score(↑) | Precision (%)(↑) | Recall (%)(↑) |
|---|---|---|---|---|
| ZEPHYRUS-REFLECTIVE | gpt-5-mini | **100.0** | **100.0** | **100.0** |
| ZEPHYRUS-DIRECT | gpt-5-mini | **100.0** | **100.0** | **100.0** |
| Text Only LLM | gpt-5-mini | 75.0 | **100.0** | 60.0 |
| ZEPHYRUS-REFLECTIVE | gemini-2.5-flash | 88.9 | 80.0 | **100.0** |
| ZEPHYRUS-DIRECT | gemini-2.5-flash | **100.0** | **100.0** | **100.0** |
| Text Only LLM | gemini-2.5-flash | 33.3 | **100.0** | 20.0 |
| ZEPHYRUS-REFLECTIVE | gpt-oss-120b | **100.0** | **100.0** | **100.0** |
| ZEPHYRUS-DIRECT | gpt-oss-120b | **100.0** | **100.0** | **100.0** |
| Text Only LLM | gpt-oss-120b | 50.0 | **100.0** | 33.3 |
| ZEPHYRUS-REFLECTIVE | gpt-5.2 | **100.0** | **100.0** | **100.0** |
| ZEPHYRUS-DIRECT | gpt-5.2 | **100.0** | **100.0** | **100.0** |
| Text Only LLM | gpt-5.2 | 83.3 | 71.4 | **100.0** |
| ZEPHYRUS-REFLECTIVE | qwen3-30b | 50.0 | **100.0** | 33.3 |
| ZEPHYRUS-DIRECT | qwen3-30b | **75.0** | **100.0** | **60.0** |
| Text Only LLM | qwen3-30b | 40.0 | 50.0 | 33.3 |

Table 34: Boolean classification metrics for Template ID 16: `Does maximum weather variable occur at higher latitude than in another region (forecast)`

| Model | LLM | F1 Score(↑) | Precision (%)(↑) | Recall (%)(↑) |
|---|---|---|---|---|
| ZEPHYRUS-REFLECTIVE | gpt-5-mini | **88.9** | **100.0** | **80.0** |
| ZEPHYRUS-DIRECT | gpt-5-mini | **88.9** | **100.0** | **80.0** |
| Text Only LLM | gpt-5-mini | 62.5 | 83.3 | 50.0 |
| ZEPHYRUS-REFLECTIVE | gemini-2.5-flash | **100.0** | **100.0** | **100.0** |
| ZEPHYRUS-DIRECT | gemini-2.5-flash | 94.7 | **100.0** | 90.0 |
| Text Only LLM | gemini-2.5-flash | 46.2 | **100.0** | 30.0 |
| ZEPHYRUS-REFLECTIVE | gpt-oss-120b | **94.7** | **100.0** | **90.0** |
| ZEPHYRUS-DIRECT | gpt-oss-120b | **94.7** | **100.0** | **90.0** |
| Text Only LLM | gpt-oss-120b | 66.7 | **100.0** | 50.0 |
| ZEPHYRUS-REFLECTIVE | gpt-5.2 | **90.0** | **90.0** | **90.0** |
| ZEPHYRUS-DIRECT | gpt-5.2 | **90.0** | **90.0** | **90.0** |
| Text Only LLM | gpt-5.2 | 58.8 | 71.4 | 50.0 |
| ZEPHYRUS-REFLECTIVE | qwen3-30b | 87.5 | 87.5 | 87.5 |
| ZEPHYRUS-DIRECT | qwen3-30b | **90.0** | **90.0** | **90.0** |
| Text Only LLM | qwen3-30b | 0.00 | 0.00 | 0.00 |

Table 35: Boolean classification metrics for Template ID 17: `Does mean weather variable in one region exceed another by specified amount (forecast)`

| Model | LLM | F1 Score(↑) | Precision (%)(↑) | Recall (%)(↑) |
|---|---|---|---|---|
| ZEPHYRUS-REFLECTIVE | gpt-5-mini | **0.00** | **0.00** | **0.00** |
| ZEPHYRUS-DIRECT | gpt-5-mini | **0.00** | **0.00** | **0.00** |
| Text Only LLM | gpt-5-mini | **0.00** | **0.00** | **0.00** |
| ZEPHYRUS-REFLECTIVE | gemini-2.5-flash | **0.00** | **0.00** | **0.00** |
| ZEPHYRUS-DIRECT | gemini-2.5-flash | **0.00** | **0.00** | **0.00** |
| Text Only LLM | gemini-2.5-flash | **0.00** | **0.00** | **0.00** |
| ZEPHYRUS-REFLECTIVE | gpt-oss-120b | **0.00** | **0.00** | **0.00** |
| ZEPHYRUS-DIRECT | gpt-oss-120b | **0.00** | **0.00** | **0.00** |
| Text Only LLM | gpt-oss-120b | **0.00** | **0.00** | **0.00** |
| ZEPHYRUS-REFLECTIVE | gpt-5.2 | **0.00** | **0.00** | **0.00** |
| ZEPHYRUS-DIRECT | gpt-5.2 | **0.00** | **0.00** | **0.00** |
| Text Only LLM | gpt-5.2 | **0.00** | **0.00** | **0.00** |
| ZEPHYRUS-REFLECTIVE | qwen3-30b | **0.00** | **0.00** | **0.00** |
| ZEPHYRUS-DIRECT | qwen3-30b | **0.00** | **0.00** | **0.00** |
| Text Only LLM | qwen3-30b | **0.00** | **0.00** | **0.00** |

Table 36: Boolean classification metrics for Template ID 18: `Does mean weather variable exceed threshold while maximum of another stays below (forecast)`

| Model | LLM | F1 Score(↑) | Precision (%)(↑) | Recall (%)(↑) |
|---|---|---|---|---|
| ZEPHYRUS-REFLECTIVE | gpt-5-mini | **100.0** | **100.0** | **100.0** |
| ZEPHYRUS-DIRECT | gpt-5-mini | **100.0** | **100.0** | **100.0** |
| Text Only LLM | gpt-5-mini | 0.00 | 0.00 | 0.00 |
| ZEPHYRUS-REFLECTIVE | gemini-2.5-flash | **100.0** | **100.0** | **100.0** |
| ZEPHYRUS-DIRECT | gemini-2.5-flash | **100.0** | **100.0** | **100.0** |
| Text Only LLM | gemini-2.5-flash | 0.00 | 0.00 | 0.00 |
| ZEPHYRUS-REFLECTIVE | gpt-oss-120b | **100.0** | **100.0** | **100.0** |
| ZEPHYRUS-DIRECT | gpt-oss-120b | **100.0** | **100.0** | **100.0** |
| Text Only LLM | gpt-oss-120b | 0.00 | 0.00 | 0.00 |
| ZEPHYRUS-REFLECTIVE | gpt-5.2 | **100.0** | **100.0** | **100.0** |
| ZEPHYRUS-DIRECT | gpt-5.2 | **100.0** | **100.0** | **100.0** |
| Text Only LLM | gpt-5.2 | 0.00 | 0.00 | 0.00 |
| ZEPHYRUS-REFLECTIVE | qwen3-30b | **100.0** | **100.0** | **100.0** |
| ZEPHYRUS-DIRECT | qwen3-30b | **100.0** | **100.0** | **100.0** |
| Text Only LLM | qwen3-30b | 0.00 | 0.00 | 0.00 |

Table 37: Boolean classification metrics for Template ID 19: `Does mean weather variable within region exceed specified threshold (forecast)`

| Model | LLM | F1 Score(↑) | Precision (%)(↑) | Recall (%)(↑) |
|---|---|---|---|---|
| ZEPHYRUS-REFLECTIVE | gpt-5-mini | **83.3** | 83.3 | **83.3** |
| ZEPHYRUS-DIRECT | gpt-5-mini | **83.3** | 83.3 | **83.3** |
| Text Only LLM | gpt-5-mini | 66.7 | **100.0** | 50.0 |
| ZEPHYRUS-REFLECTIVE | gemini-2.5-flash | **83.3** | 83.3 | **83.3** |
| ZEPHYRUS-DIRECT | gemini-2.5-flash | **83.3** | 83.3 | **83.3** |
| Text Only LLM | gemini-2.5-flash | 50.0 | **100.0** | 33.3 |
| ZEPHYRUS-REFLECTIVE | gpt-oss-120b | **83.3** | 83.3 | **83.3** |
| ZEPHYRUS-DIRECT | gpt-oss-120b | **83.3** | 83.3 | **83.3** |
| Text Only LLM | gpt-oss-120b | 28.6 | **100.0** | 16.7 |
| ZEPHYRUS-REFLECTIVE | gpt-5.2 | **83.3** | 83.3 | **83.3** |
| ZEPHYRUS-DIRECT | gpt-5.2 | **83.3** | 83.3 | **83.3** |
| Text Only LLM | gpt-5.2 | **83.3** | 83.3 | **83.3** |
| ZEPHYRUS-REFLECTIVE | qwen3-30b | 80.0 | 80.0 | 80.0 |
| ZEPHYRUS-DIRECT | qwen3-30b | **83.3** | **83.3** | **83.3** |
| Text Only LLM | qwen3-30b | 0.00 | 0.00 | 0.00 |

Table 38: Boolean classification metrics for Template ID 20: `Does weather variable exceed threshold within any part of region (forecast)`

| Model | LLM | F1 Score(↑) | Precision (%)(↑) | Recall (%)(↑) |
|---|---|---|---|---|
| ZEPHYRUS-REFLECTIVE | gpt-5-mini | **100.0** | **100.0** | **100.0** |
| ZEPHYRUS-DIRECT | gpt-5-mini | **100.0** | **100.0** | **100.0** |
| Text Only LLM | gpt-5-mini | 88.9 | 80.0 | **100.0** |
| ZEPHYRUS-REFLECTIVE | gemini-2.5-flash | **100.0** | **100.0** | **100.0** |
| ZEPHYRUS-DIRECT | gemini-2.5-flash | **100.0** | **100.0** | **100.0** |
| Text Only LLM | gemini-2.5-flash | 40.0 | **100.0** | 25.0 |
| ZEPHYRUS-REFLECTIVE | gpt-oss-120b | **100.0** | **100.0** | **100.0** |
| ZEPHYRUS-DIRECT | gpt-oss-120b | **100.0** | **100.0** | **100.0** |
| Text Only LLM | gpt-oss-120b | 0.00 | 0.00 | 0.00 |
| ZEPHYRUS-REFLECTIVE | gpt-5.2 | **100.0** | **100.0** | **100.0** |
| ZEPHYRUS-DIRECT | gpt-5.2 | **100.0** | **100.0** | **100.0** |
| Text Only LLM | gpt-5.2 | 88.9 | 80.0 | **100.0** |
| ZEPHYRUS-REFLECTIVE | qwen3-30b | 85.7 | **100.0** | 75.0 |
| ZEPHYRUS-DIRECT | qwen3-30b | **100.0** | **100.0** | **100.0** |
| Text Only LLM | qwen3-30b | 0.00 | 0.00 | 0.00 |

Table 39: Boolean classification metrics for Template ID 21: `Does weather variable exceed threshold in more grid points in one region than another (forecast)`

| Model | LLM | F1 Score(↑) | Precision (%)(↑) | Recall (%)(↑) |
|---|---|---|---|---|
| ZEPHYRUS-REFLECTIVE | gpt-5-mini | **100.0** | **100.0** | **100.0** |
| ZEPHYRUS-DIRECT | gpt-5-mini | 0.00 | 0.00 | 0.00 |
| Text Only LLM | gpt-5-mini | 0.00 | 0.00 | 0.00 |
| ZEPHYRUS-REFLECTIVE | gemini-2.5-flash | **100.0** | **100.0** | **100.0** |
| ZEPHYRUS-DIRECT | gemini-2.5-flash | **100.0** | **100.0** | **100.0** |
| Text Only LLM | gemini-2.5-flash | 66.7 | 50.0 | **100.0** |
| ZEPHYRUS-REFLECTIVE | gpt-oss-120b | **100.0** | **100.0** | **100.0** |
| ZEPHYRUS-DIRECT | gpt-oss-120b | **100.0** | **100.0** | **100.0** |
| Text Only LLM | gpt-oss-120b | 33.3 | 20.0 | **100.0** |
| ZEPHYRUS-REFLECTIVE | gpt-5.2 | **100.0** | **100.0** | **100.0** |
| ZEPHYRUS-DIRECT | gpt-5.2 | **100.0** | **100.0** | **100.0** |
| Text Only LLM | gpt-5.2 | 0.00 | 0.00 | 0.00 |
| ZEPHYRUS-REFLECTIVE | qwen3-30b | **100.0** | **100.0** | **100.0** |
| ZEPHYRUS-DIRECT | qwen3-30b | **100.0** | **100.0** | **100.0** |
| Text Only LLM | qwen3-30b | 0.00 | 0.00 | 0.00 |

Table 40: Boolean classification metrics for Template ID 22: `Does area where weather variable exceeds threshold cover more than percentage of region (forecast)`

| Model | LLM | F1 Score(↑) | Precision (%)(↑) | Recall (%)(↑) |
|---|---|---|---|---|
| ZEPHYRUS-REFLECTIVE | gpt-5-mini | **100.0** | **100.0** | **100.0** |
| ZEPHYRUS-DIRECT | gpt-5-mini | **100.0** | **100.0** | **100.0** |
| Text Only LLM | gpt-5-mini | 66.7 | **100.0** | 50.0 |
| ZEPHYRUS-REFLECTIVE | gemini-2.5-flash | **100.0** | **100.0** | **100.0** |
| ZEPHYRUS-DIRECT | gemini-2.5-flash | **100.0** | **100.0** | **100.0** |
| Text Only LLM | gemini-2.5-flash | 0.00 | 0.00 | 0.00 |
| ZEPHYRUS-REFLECTIVE | gpt-oss-120b | **100.0** | **100.0** | **100.0** |
| ZEPHYRUS-DIRECT | gpt-oss-120b | **100.0** | **100.0** | **100.0** |
| Text Only LLM | gpt-oss-120b | 66.7 | **100.0** | 50.0 |
| ZEPHYRUS-REFLECTIVE | gpt-5.2 | **100.0** | **100.0** | **100.0** |
| ZEPHYRUS-DIRECT | gpt-5.2 | **100.0** | **100.0** | **100.0** |
| Text Only LLM | gpt-5.2 | 80.0 | 66.7 | **100.0** |
| ZEPHYRUS-REFLECTIVE | qwen3-30b | **100.0** | **100.0** | **100.0** |
| ZEPHYRUS-DIRECT | qwen3-30b | 66.7 | **100.0** | 50.0 |
| Text Only LLM | qwen3-30b | 0.0 | 0.0 | 0.0 |

Table 41: Boolean classification metrics for Template ID 23: `Does area-averaged weather variable exceed threshold while another stays below (forecast)`

| Model | LLM | F1 Score(↑) | Precision (%)(↑) | Recall (%)(↑) |
|---|---|---|---|---|
| ZEPHYRUS-REFLECTIVE | gpt-5-mini | **80.0** | **100.0** | **66.7** |
| ZEPHYRUS-DIRECT | gpt-5-mini | 72.7 | 80.0 | **66.7** |
| Text Only LLM | gpt-5-mini | 50.0 | 50.0 | 50.0 |
| ZEPHYRUS-REFLECTIVE | gemini-2.5-flash | **80.0** | **100.0** | **66.7** |
| ZEPHYRUS-DIRECT | gemini-2.5-flash | **80.0** | **100.0** | **66.7** |
| Text Only LLM | gemini-2.5-flash | 28.6 | 33.3 | 25.0 |
| ZEPHYRUS-REFLECTIVE | gpt-oss-120b | **80.0** | **100.0** | **66.7** |
| ZEPHYRUS-DIRECT | gpt-oss-120b | **80.0** | **100.0** | **66.7** |
| Text Only LLM | gpt-oss-120b | 60.0 | 75.0 | 50.0 |
| ZEPHYRUS-REFLECTIVE | gpt-5.2 | 80.0 | **100.0** | 66.7 |
| ZEPHYRUS-DIRECT | gpt-5.2 | 80.0 | **100.0** | 66.7 |
| Text Only LLM | gpt-5.2 | **83.3** | 83.3 | **83.3** |
| ZEPHYRUS-REFLECTIVE | qwen3-30b | 75.0 | **100.0** | 60.0 |
| ZEPHYRUS-DIRECT | qwen3-30b | **80.0** | **100.0** | **66.7** |
| Text Only LLM | qwen3-30b | 0.00 | 0.00 | 0.00 |

Table 42: Boolean classification metrics for Template ID 24: `Does maximum weather variable in one region exceed threshold while another stays below (forecast)`

| Model | LLM | F1 Score(↑) | Precision (%)(↑) | Recall (%)(↑) |
|---|---|---|---|---|
| ZEPHYRUS-REFLECTIVE | gpt-5-mini | **90.9** | **100.0** | 83.3 |
| ZEPHYRUS-DIRECT | gpt-5-mini | **90.9** | **100.0** | 83.3 |
| Text Only LLM | gpt-5-mini | 83.3 | 83.3 | **83.3** |
| ZEPHYRUS-REFLECTIVE | gemini-2.5-flash | 88.9 | **100.0** | 80.0 |
| ZEPHYRUS-DIRECT | gemini-2.5-flash | **90.9** | **100.0** | **83.3** |
| Text Only LLM | gemini-2.5-flash | 50.0 | **100.0** | 33.3 |
| ZEPHYRUS-REFLECTIVE | gpt-oss-120b | **90.9** | **100.0** | **83.3** |
| ZEPHYRUS-DIRECT | gpt-oss-120b | **90.9** | **100.0** | **83.3** |
| Text Only LLM | gpt-oss-120b | 75.0 | **100.0** | 60.0 |
| ZEPHYRUS-REFLECTIVE | gpt-5.2 | **90.9** | **100.0** | **83.3** |
| ZEPHYRUS-DIRECT | gpt-5.2 | **90.9** | **100.0** | **83.3** |
| Text Only LLM | gpt-5.2 | 80.0 | **100.0** | 66.7 |
| ZEPHYRUS-REFLECTIVE | qwen3-30b | **100.0** | **100.0** | **100.0** |
| ZEPHYRUS-DIRECT | qwen3-30b | 90.9 | 83.3 | **100.0** |
| Text Only LLM | qwen3-30b | 40.0 | **100.0** | 25.0 |

Table 43: Boolean classification metrics for Template ID 25: `Does maximum weather variable within region exceed specified threshold (forecast)`

| Model | LLM | F1 Score(↑) | Precision (%)(↑) | Recall (%)(↑) |
|---|---|---|---|---|
| ZEPHYRUS-REFLECTIVE | gpt-5-mini | 80.0 | **100.0** | 66.7 |
| ZEPHYRUS-DIRECT | gpt-5-mini | **90.9** | **100.0** | **83.3** |
| Text Only LLM | gpt-5-mini | 76.9 | 71.4 | **83.3** |
| ZEPHYRUS-REFLECTIVE | gemini-2.5-flash | 90.9 | 100.0 | 83.3 |
| ZEPHYRUS-DIRECT | gemini-2.5-flash | **100.0** | **100.0** | **100.0** |
| Text Only LLM | gemini-2.5-flash | 66.7 | 75.0 | 60.0 |
| ZEPHYRUS-REFLECTIVE | gpt-oss-120b | **80.0** | **100.0** | **66.7** |
| ZEPHYRUS-DIRECT | gpt-oss-120b | **80.0** | **100.0** | **66.7** |
| Text Only LLM | gpt-oss-120b | 60.0 | 60.0 | 60.0 |
| ZEPHYRUS-REFLECTIVE | gpt-5.2 | **90.9** | **100.0** | **83.3** |
| ZEPHYRUS-DIRECT | gpt-5.2 | **90.9** | **100.0** | **83.3** |
| Text Only LLM | gpt-5.2 | 72.7 | 80.0 | 66.7 |
| ZEPHYRUS-REFLECTIVE | qwen3-30b | 66.7 | **100.0** | 50.0 |
| ZEPHYRUS-DIRECT | qwen3-30b | **92.3** | 85.7 | **100.0** |
| Text Only LLM | qwen3-30b | 33.3 | **100.0** | 20.0 |

Table 44: Boolean classification metrics for Template ID 26: `Does maximum weather variable occur at latitude farther north than in another region (forecast)`

| Model | LLM | F1 Score(↑) | Precision (%)(↑) | Recall (%)(↑) |
|---|---|---|---|---|
| ZEPHYRUS-REFLECTIVE | gpt-5-mini | **88.9** | **100.0** | **80.0** |
| ZEPHYRUS-DIRECT | gpt-5-mini | **88.9** | **100.0** | **80.0** |
| Text Only LLM | gpt-5-mini | 57.1 | **100.0** | 40.0 |
| ZEPHYRUS-REFLECTIVE | gemini-2.5-flash | **88.9** | **100.0** | **80.0** |
| ZEPHYRUS-DIRECT | gemini-2.5-flash | **88.9** | **100.0** | **80.0** |
| Text Only LLM | gemini-2.5-flash | 50.0 | 66.7 | 40.0 |
| ZEPHYRUS-REFLECTIVE | gpt-oss-120b | **88.9** | **100.0** | **80.0** |
| ZEPHYRUS-DIRECT | gpt-oss-120b | **88.9** | **100.0** | **80.0** |
| Text Only LLM | gpt-oss-120b | 57.1 | 66.7 | 50.0 |
| ZEPHYRUS-REFLECTIVE | gpt-5.2 | **88.9** | **100.0** | **80.0** |
| ZEPHYRUS-DIRECT | gpt-5.2 | 75.0 | **100.0** | 60.0 |
| Text Only LLM | gpt-5.2 | 60.0 | 60.0 | 60.0 |
| ZEPHYRUS-REFLECTIVE | qwen3-30b | **75.0** | **100.0** | **60.0** |
| ZEPHYRUS-DIRECT | qwen3-30b | 57.1 | **100.0** | 40.0 |
| Text Only LLM | qwen3-30b | 0.00 | 0.00 | 0.00 |

Table 45: Boolean classification metrics for Template ID 27: `Does maximum weather variable stay above threshold while another stays below (forecast)`

| Model | LLM | F1 Score(↑) | Precision (%)(↑) | Recall (%)(↑) |
|---|---|---|---|---|
| ZEPHYRUS-REFLECTIVE | gpt-5-mini | 75.0 | 75.0 | 75.0 |
| ZEPHYRUS-DIRECT | gpt-5-mini | **88.9** | 80.0 | **100.0** |
| Text Only LLM | gpt-5-mini | 40.0 | **100.0** | 25.0 |
| ZEPHYRUS-REFLECTIVE | gemini-2.5-flash | **75.0** | **75.0** | **75.0** |
| ZEPHYRUS-DIRECT | gemini-2.5-flash | **75.0** | **75.0** | **75.0** |
| Text Only LLM | gemini-2.5-flash | 57.1 | 66.7 | 50.0 |
| ZEPHYRUS-REFLECTIVE | gpt-oss-120b | **75.0** | **75.0** | **75.0** |
| ZEPHYRUS-DIRECT | gpt-oss-120b | **75.0** | **75.0** | **75.0** |
| Text Only LLM | gpt-oss-120b | 0.00 | 0.00 | 0.00 |
| ZEPHYRUS-REFLECTIVE | gpt-5.2 | 75.0 | 75.0 | **75.0** |
| ZEPHYRUS-DIRECT | gpt-5.2 | **75.0** | 75.0 | **75.0** |
| Text Only LLM | gpt-5.2 | 66.7 | **100.0** | 50.0 |
| ZEPHYRUS-REFLECTIVE | qwen3-30b | **85.7** | **100.0** | 75.0 |
| ZEPHYRUS-DIRECT | qwen3-30b | 75.0 | 75.0 | **75.0** |
| Text Only LLM | qwen3-30b | 0.00 | 0.00 | 0.00 |

Table 46: Boolean classification metrics for Template ID 28: `Does maximum weather variable in one region exceed another by specified amount (forecast)`

| Model | LLM | F1 Score(↑) | Precision (%)(↑) | Recall (%)(↑) |
|---|---|---|---|---|
| ZEPHYRUS-REFLECTIVE | gpt-5-mini | **90.9** | 83.3 | **100.0** |
| ZEPHYRUS-DIRECT | gpt-5-mini | **90.9** | 83.3 | **100.0** |
| Text Only LLM | gpt-5-mini | 33.3 | **100.0** | 20.0 |
| ZEPHYRUS-REFLECTIVE | gemini-2.5-flash | 83.3 | 71.4 | **100.0** |
| ZEPHYRUS-DIRECT | gemini-2.5-flash | **90.9** | 83.3 | **100.0** |
| Text Only LLM | gemini-2.5-flash | 33.3 | **100.0** | 20.0 |
| ZEPHYRUS-REFLECTIVE | gpt-oss-120b | **90.9** | 83.3 | **100.0** |
| ZEPHYRUS-DIRECT | gpt-oss-120b | **90.9** | 83.3 | **100.0** |
| Text Only LLM | gpt-oss-120b | 85.7 | **100.0** | 75.0 |
| ZEPHYRUS-REFLECTIVE | gpt-5.2 | **90.9** | 83.3 | **100.0** |
| ZEPHYRUS-DIRECT | gpt-5.2 | **90.9** | 83.3 | **100.0** |
| Text Only LLM | gpt-5.2 | 75.0 | **100.0** | 60.0 |
| ZEPHYRUS-REFLECTIVE | qwen3-30b | **90.9** | 83.3 | **100.0** |
| ZEPHYRUS-DIRECT | qwen3-30b | **90.9** | 83.3 | **100.0** |
| Text Only LLM | qwen3-30b | 50.0 | **100.0** | 33.3 |

Table 47: Boolean classification metrics for Template ID 29: `Does minimum weather variable within region remain above threshold (forecast)`

| Model | LLM | MAE (median)(↓) |
|---|---|---|
| ZEPHYRUS-REFLECTIVE | gpt-5-mini | 191,000 |
| ZEPHYRUS-DIRECT | gpt-5-mini | **182,000** |
| Text Only LLM | gpt-5-mini | 249,000 |
| ZEPHYRUS-REFLECTIVE | gemini-2.5-flash | 183,000 |
| ZEPHYRUS-DIRECT | gemini-2.5-flash | 182,000 |
| Text Only LLM | gemini-2.5-flash | **150,000** |
| ZEPHYRUS-REFLECTIVE | gpt-oss-120b | 3,860 |
| ZEPHYRUS-DIRECT | gpt-oss-120b | **0.00** |
| Text Only LLM | gpt-oss-120b | 1.84e+06 |
| ZEPHYRUS-REFLECTIVE | gpt-5.2 | **0.00** |
| ZEPHYRUS-DIRECT | gpt-5.2 | **0.00** |
| Text Only LLM | gpt-5.2 | 174,000 |
| ZEPHYRUS-REFLECTIVE | qwen3-30b | **0.00** |
| ZEPHYRUS-DIRECT | qwen3-30b | 232,000 |
| Text Only LLM | qwen3-30b | 182,000 |

Table 48: Performance metrics for Template ID 30: `What is the area where multiple weather variables exceed their percentile values (forecast)`

| Model | LLM | MAE (median)(↓) |
|---|---|---|
| ZEPHYRUS-REFLECTIVE | gpt-5-mini | 103,000 |
| ZEPHYRUS-DIRECT | gpt-5-mini | **29,200** |
| Text Only LLM | gpt-5-mini | 349,000 |
| ZEPHYRUS-REFLECTIVE | gemini-2.5-flash | 23,800 |
| ZEPHYRUS-DIRECT | gemini-2.5-flash | **19,200** |
| Text Only LLM | gemini-2.5-flash | 20,400 |
| ZEPHYRUS-REFLECTIVE | gpt-oss-120b | **13,600** |
| ZEPHYRUS-DIRECT | gpt-oss-120b | **13,600** |
| Text Only LLM | gpt-oss-120b | 4.26e+12 |
| ZEPHYRUS-REFLECTIVE | gpt-5.2 | **13,600** |
| ZEPHYRUS-DIRECT | gpt-5.2 | **13,600** |
| Text Only LLM | gpt-5.2 | 15,800 |
| ZEPHYRUS-REFLECTIVE | qwen3-30b | 16,000 |
| ZEPHYRUS-DIRECT | qwen3-30b | **13,600** |
| Text Only LLM | qwen3-30b | 19,100 |

Table 49: Performance metrics for Template ID 31: `What is the area where weather variable exceeds its median value (forecast)`

| Model | LLM | MAE (median)(↓) |
|---|---|---|
| ZEPHYRUS-REFLECTIVE | gpt-5-mini | 149.0 |
| ZEPHYRUS-DIRECT | gpt-5-mini | **96.0** |
| Text Only LLM | gpt-5-mini | 293.0 |
| ZEPHYRUS-REFLECTIVE | gemini-2.5-flash | 171.0 |
| ZEPHYRUS-DIRECT | gemini-2.5-flash | **163.0** |
| Text Only LLM | gemini-2.5-flash | 2,640 |
| ZEPHYRUS-REFLECTIVE | gpt-oss-120b | 103.0 |
| ZEPHYRUS-DIRECT | gpt-oss-120b | **99.9** |
| ZEPHYRUS-REFLECTIVE | gpt-5.2 | **104.0** |
| ZEPHYRUS-DIRECT | gpt-5.2 | **104.0** |
| Text Only LLM | gpt-5.2 | 531.0 |
| ZEPHYRUS-REFLECTIVE | qwen3-30b | 197.0 |
| ZEPHYRUS-DIRECT | qwen3-30b | **163.0** |
| Text Only LLM | qwen3-30b | 11,600 |

Table 50: Performance metrics for Template ID 32: `What is the displacement between centroids of areas with above-median values (forecast)`

| Model | LLM | MAE (median)(↓) |
|---|---|---|
| ZEPHYRUS-REFLECTIVE | gpt-5-mini | **107.0** |
| ZEPHYRUS-DIRECT | gpt-5-mini | 137.0 |
| Text Only LLM | gpt-5-mini | 1.91e+06 |
| ZEPHYRUS-REFLECTIVE | gemini-2.5-flash | **112.0** |
| ZEPHYRUS-DIRECT | gemini-2.5-flash | 164.0 |
| Text Only LLM | gemini-2.5-flash | 426.0 |
| ZEPHYRUS-REFLECTIVE | gpt-oss-120b | 158.0 |
| ZEPHYRUS-DIRECT | gpt-oss-120b | 163.0 |
| Text Only LLM | gpt-oss-120b | **122.0** |
| ZEPHYRUS-REFLECTIVE | gpt-5.2 | 138.0 |
| ZEPHYRUS-DIRECT | gpt-5.2 | **107.0** |
| Text Only LLM | gpt-5.2 | 183.0 |
| ZEPHYRUS-REFLECTIVE | qwen3-30b | **107.0** |
| ZEPHYRUS-DIRECT | qwen3-30b | 136.0 |
| Text Only LLM | qwen3-30b | 9,110 |

Table 51: Performance metrics for Template ID 33: `What is the distance between centroids of maximum weather variable value areas (forecast)`

| Model | LLM | SAE (Q25)(↓) | SAE (Q50)(↓) | SAE (Q75)(↓) | SAE (Q99)(↓) | Err Std(↓) |
|---|---|---|---|---|---|---|
| ZEPHYRUS-REFLECTIVE | gpt-5-mini | **0.00** | 0.02 | **0.09** | **0.95** | **15.1** |
| ZEPHYRUS-DIRECT | gpt-5-mini | **0.00** | **0.01** | 0.13 | 1.01 | **15.1** |
| Text Only LLM | gpt-5-mini | 0.21 | 0.67 | 1.26 | 2.72 | 155.0 |
| ZEPHYRUS-REFLECTIVE | gemini-2.5-flash | **0.00** | **0.02** | **0.09** | **0.95** | **15.1** |
| ZEPHYRUS-DIRECT | gemini-2.5-flash | **0.00** | **0.02** | **0.09** | **0.95** | **15.1** |
| Text Only LLM | gemini-2.5-flash | 0.05 | 0.47 | 1.41 | 5.30 | 510.0 |
| ZEPHYRUS-REFLECTIVE | gpt-oss-120b | **0.00** | **0.02** | **0.09** | **0.95** | **15.1** |
| ZEPHYRUS-DIRECT | gpt-oss-120b | **0.00** | **0.02** | **0.09** | **0.95** | **15.1** |
| Text Only LLM | gpt-oss-120b | 0.16 | 0.21 | 0.50 | 1.24 | 459.0 |
| ZEPHYRUS-REFLECTIVE | gpt-5.2 | **0.00** | **0.02** | **0.09** | **0.95** | **15.1** |
| ZEPHYRUS-DIRECT | gpt-5.2 | **0.00** | **0.02** | **0.09** | **0.95** | **15.1** |
| Text Only LLM | gpt-5.2 | 0.35 | 0.58 | 4.62 | 311.0 | 32.0 |
| ZEPHYRUS-REFLECTIVE | qwen3-30b | **0.00** | **0.02** | **0.09** | **0.95** | **15.1** |
| ZEPHYRUS-DIRECT | qwen3-30b | **0.00** | **0.02** | **0.09** | **0.95** | **15.1** |
| Text Only LLM | qwen3-30b | 0.07 | 0.17 | 1.40 | 7.11e+06 | 203.0 |

Table 52: Standardized Absolute Error (SAE) quantiles for Template ID 34: `What is the maximum difference in weather variable between grid points within region (forecast)`

| Model | LLM | SAE (Q25)(↓) | SAE (Q50)(↓) | SAE (Q75)(↓) | SAE (Q99)(↓) | Err Std(↓) |
|---|---|---|---|---|---|---|
| ZEPHYRUS-REFLECTIVE | gpt-5-mini | 0.01 | **0.03** | **0.03** | **0.13** | **0.58** |
| ZEPHYRUS-DIRECT | gpt-5-mini | **0.00** | **0.03** | **0.03** | **0.13** | 0.61 |
| Text Only LLM | gpt-5-mini | 0.02 | 0.09 | 0.50 | 1.14 | 3.07 |
| ZEPHYRUS-REFLECTIVE | gemini-2.5-flash | **0.01** | **0.03** | **0.03** | **0.13** | **0.58** |
| ZEPHYRUS-DIRECT | gemini-2.5-flash | **0.01** | **0.03** | **0.03** | **0.13** | **0.58** |
| Text Only LLM | gemini-2.5-flash | 0.17 | 0.31 | 1.00 | 4.25 | 17.5 |
| ZEPHYRUS-REFLECTIVE | gpt-oss-120b | **0.01** | **0.03** | **0.03** | **0.13** | 0.58 |
| ZEPHYRUS-DIRECT | gpt-oss-120b | **0.01** | **0.03** | **0.03** | **0.13** | 0.58 |
| Text Only LLM | gpt-oss-120b | 0.30 | 0.30 | 0.30 | 0.30 | **0.00** |
| ZEPHYRUS-REFLECTIVE | gpt-5.2 | **0.01** | **0.03** | **0.03** | **0.13** | **0.58** |
| ZEPHYRUS-DIRECT | gpt-5.2 | **0.01** | **0.03** | **0.03** | **0.13** | **0.58** |
| Text Only LLM | gpt-5.2 | 0.10 | 0.39 | 0.92 | 3.01 | 12.7 |
| ZEPHYRUS-REFLECTIVE | qwen3-30b | 0.02 | **0.03** | 0.07 | **0.13** | **0.56** |
| ZEPHYRUS-DIRECT | qwen3-30b | **0.01** | **0.03** | **0.03** | **0.13** | 0.58 |
| Text Only LLM | qwen3-30b | 7.88 | 15.2 | 15.8 | 17.7 | 110.0 |

Table 53: Standardized Absolute Error (SAE) quantiles for Template ID 35: `What is the minimum weather variable where another variable exceeds median (forecast)`

| Model | LLM | SAE (Q25)(↓) | SAE (Q50)(↓) | SAE (Q75)(↓) | SAE (Q99)(↓) | Err Std(↓) |
|---|---|---|---|---|---|---|
| ZEPHYRUS-REFLECTIVE | gpt-5-mini | **0.10** | **0.23** | 0.45 | **1.54** | **5.84** |
| ZEPHYRUS-DIRECT | gpt-5-mini | 0.16 | **0.23** | 0.35 | 1.82 | 6.80 |
| Text Only LLM | gpt-5-mini | 0.81 | 0.99 | 4.04 | 5.95 | 20.5 |
| ZEPHYRUS-REFLECTIVE | gemini-2.5-flash | **0.15** | **0.25** | 0.75 | 1.49 | 6.42 |
| ZEPHYRUS-DIRECT | gemini-2.5-flash | 0.23 | 0.33 | **0.56** | **1.48** | **5.50** |
| Text Only LLM | gemini-2.5-flash | 0.72 | 2.04 | 2.76 | 3.56 | 18.2 |
| ZEPHYRUS-REFLECTIVE | gpt-oss-120b | **0.11** | **0.23** | 0.50 | 1.55 | **5.93** |
| ZEPHYRUS-DIRECT | gpt-oss-120b | 0.13 | 0.27 | 0.91 | 5.10 | 18.3 |
| Text Only LLM | gpt-oss-120b | 0.56 | 1.25 | 1.95 | 2.00 | 7.77 |
| ZEPHYRUS-REFLECTIVE | gpt-5.2 | 0.15 | **0.44** | 0.85 | 5.18 | 19.0 |
| ZEPHYRUS-DIRECT | gpt-5.2 | **0.13** | **0.44** | 0.96 | **5.10** | 18.2 |
| Text Only LLM | gpt-5.2 | 0.39 | 0.65 | 1.96 | 300.0 | **14.9** |
| ZEPHYRUS-REFLECTIVE | qwen3-30b | **0.20** | **0.33** | **0.52** | **1.80** | **6.55** |
| ZEPHYRUS-DIRECT | qwen3-30b | 0.24 | 0.38 | 1.07 | 6.28 | 23.1 |
| Text Only LLM | qwen3-30b | 1.11 | 2.13 | 5.23 | 19.3 | 98.5 |

Table 54: Standardized Absolute Error (SAE) quantiles for Template ID 36: `What is the difference between maximum weather variables in two regions (forecast)`

| Model | LLM | SAE (Q25)(↓) | SAE (Q50)(↓) | SAE (Q75)(↓) | SAE (Q99)(↓) | Err Std(↓) |
|---|---|---|---|---|---|---|
| ZEPHYRUS-REFLECTIVE | gpt-5-mini | **0.04** | **0.07** | **0.21** | **0.67** | **238.0** |
| ZEPHYRUS-DIRECT | gpt-5-mini | **0.04** | **0.07** | **0.21** | **0.67** | **238.0** |
| Text Only LLM | gpt-5-mini | 0.25 | 1.03 | 2.09 | 5.79 | 1,010 |
| ZEPHYRUS-REFLECTIVE | gemini-2.5-flash | **0.04** | **0.06** | **0.24** | **0.49** | **247.0** |
| ZEPHYRUS-DIRECT | gemini-2.5-flash | **0.04** | 0.12 | 0.32 | 0.68 | 250.0 |
| Text Only LLM | gemini-2.5-flash | 0.41 | 1.04 | 2.23 | 30.9 | 1,130 |
| ZEPHYRUS-REFLECTIVE | gpt-oss-120b | 0.06 | 0.12 | 0.33 | **0.67** | 238.0 |
| ZEPHYRUS-DIRECT | gpt-oss-120b | **0.04** | **0.06** | **0.21** | **0.67** | 238.0 |
| Text Only LLM | gpt-oss-120b | 0.26 | 0.99 | 1.94 | 2.58 | **4.31** |
| ZEPHYRUS-REFLECTIVE | gpt-5.2 | **0.04** | **0.07** | **0.21** | **0.67** | **238.0** |
| ZEPHYRUS-DIRECT | gpt-5.2 | **0.04** | **0.07** | **0.21** | **0.67** | **238.0** |
| Text Only LLM | gpt-5.2 | 0.53 | 0.77 | 1.35 | 19.5 | 772.0 |
| ZEPHYRUS-REFLECTIVE | qwen3-30b | **0.04** | **0.06** | **0.21** | **0.67** | 238.0 |
| ZEPHYRUS-DIRECT | qwen3-30b | **0.04** | 0.10 | 0.36 | 3.10 | **237.0** |
| Text Only LLM | qwen3-30b | 0.23 | 2.16 | 30.6 | 31,500 | 724.0 |

Table 55: Standardized Absolute Error (SAE) quantiles for Template ID 37: `What is the difference in area-weighted mean weather variable between two regions` (forecast)

| Model | LLM | MAE (median)(↓) |
|---|---|---|
| ZEPHYRUS-REFLECTIVE | gpt-5-mini | **36.2** |
| ZEPHYRUS-DIRECT | gpt-5-mini | **36.2** |
| Text Only LLM | gpt-5-mini | 258.0 |
| ZEPHYRUS-REFLECTIVE | gemini-2.5-flash | 135.0 |
| ZEPHYRUS-DIRECT | gemini-2.5-flash | **73.9** |
| Text Only LLM | gemini-2.5-flash | 275.0 |
| ZEPHYRUS-REFLECTIVE | gpt-oss-120b | 73.9 |
| ZEPHYRUS-DIRECT | gpt-oss-120b | **73.8** |
| Text Only LLM | gpt-oss-120b | 1,990 |
| ZEPHYRUS-REFLECTIVE | gpt-5.2 | **36.2** |
| ZEPHYRUS-DIRECT | gpt-5.2 | **36.2** |
| Text Only LLM | gpt-5.2 | 51.9 |
| ZEPHYRUS-REFLECTIVE | qwen3-30b | 302.0 |
| ZEPHYRUS-DIRECT | qwen3-30b | **43.8** |
| Text Only LLM | qwen3-30b | 258.0 |

Table 56: Performance metrics for Template ID 38: `What is the displacement of minimum weather variable location after time window` (forecast)

| Model | LLM | MAE (median)(↓) |
|---|---|---|
| ZEPHYRUS-REFLECTIVE | gpt-5-mini | 5.19 |
| ZEPHYRUS-DIRECT | gpt-5-mini | **4.38** |
| Text Only LLM | gpt-5-mini | 8.66 |
| ZEPHYRUS-REFLECTIVE | gemini-2.5-flash | 4.13 |
| ZEPHYRUS-DIRECT | gemini-2.5-flash | **2.49** |
| Text Only LLM | gemini-2.5-flash | 9.54 |
| ZEPHYRUS-REFLECTIVE | gpt-oss-120b | 6.82 |
| ZEPHYRUS-DIRECT | gpt-oss-120b | **0.88** |
| Text Only LLM | gpt-oss-120b | 21.2 |
| ZEPHYRUS-REFLECTIVE | gpt-5.2 | 3.83 |
| ZEPHYRUS-DIRECT | gpt-5.2 | **2.04** |
| Text Only LLM | gpt-5.2 | 3.24 |
| ZEPHYRUS-REFLECTIVE | qwen3-30b | **4.93** |
| ZEPHYRUS-DIRECT | qwen3-30b | 5.30 |
| Text Only LLM | qwen3-30b | 21.2 |

Table 57: Performance metrics for Template ID 39: `What is the latitude difference between centroids of high weather variable areas (forecast)`

| Model | LLM | SAE (Q25)(↓) | SAE (Q50)(↓) | SAE (Q75)(↓) | SAE (Q99)(↓) | Err Std(↓) |
|---|---|---|---|---|---|---|
| ZEPHYRUS-REFLECTIVE | gpt-5-mini | **0.01** | 0.14 | **0.51** | **0.95** | **28.3** |
| ZEPHYRUS-DIRECT | gpt-5-mini | **0.01** | **0.10** | **0.51** | **0.95** | **28.3** |
| Text Only LLM | gpt-5-mini | 0.53 | 1.16 | 2.99 | 3.79 | 2,880 |
| ZEPHYRUS-REFLECTIVE | gemini-2.5-flash | **0.07** | 0.28 | 0.58 | 0.90 | **19.8** |
| ZEPHYRUS-DIRECT | gemini-2.5-flash | **0.07** | **0.13** | **0.28** | **0.56** | 28.4 |
| Text Only LLM | gemini-2.5-flash | 0.28 | 0.45 | 1.52 | 3.28 | 2,760 |
| ZEPHYRUS-REFLECTIVE | gpt-oss-120b | **0.01** | **0.14** | **0.51** | 2.55 | 28.2 |
| ZEPHYRUS-DIRECT | gpt-oss-120b | **0.01** | **0.14** | **0.51** | **0.90** | 28.3 |
| Text Only LLM | gpt-oss-120b | 1.83 | 1.91 | 9.95 | 17.7 | **10.3** |
| ZEPHYRUS-REFLECTIVE | gpt-5.2 | **0.01** | **0.09** | 0.51 | 0.80 | **28.3** |
| ZEPHYRUS-DIRECT | gpt-5.2 | **0.01** | 0.14 | **0.48** | **0.78** | **28.3** |
| Text Only LLM | gpt-5.2 | 0.14 | 0.80 | 2.00 | 21.1 | 1,460 |
| ZEPHYRUS-REFLECTIVE | qwen3-30b | **0.01** | **0.14** | 0.51 | 0.95 | **28.3** |
| ZEPHYRUS-DIRECT | qwen3-30b | **0.01** | **0.14** | **0.28** | **0.78** | **28.3** |
| Text Only LLM | qwen3-30b | 0.39 | 1.73 | 3.06 | 1.61e+06 | 2,660 |

Table 58: Standardized Absolute Error (SAE) quantiles for Template ID 40: `What is the maximum weather variable difference between two regions (forecast)`

| Model | LLM | SAE (Q25)(↓) | SAE (Q50)(↓) | SAE (Q75)(↓) | SAE (Q99)(↓) | Err Std(↓) |
|---|---|---|---|---|---|---|
| ZEPHYRUS-REFLECTIVE | gpt-5-mini | **0.00** | **0.02** | **0.07** | **0.21** | **5.84** |
| ZEPHYRUS-DIRECT | gpt-5-mini | **0.00** | **0.02** | **0.07** | **0.21** | **5.84** |
| Text Only LLM | gpt-5-mini | 0.44 | 0.70 | 1.95 | 32.9 | 259.0 |
| ZEPHYRUS-REFLECTIVE | gemini-2.5-flash | 0.04 | 0.18 | 4.51 | 10.7 | 3,290 |
| ZEPHYRUS-DIRECT | gemini-2.5-flash | **0.00** | **0.02** | **0.07** | **0.21** | **5.84** |
| Text Only LLM | gemini-2.5-flash | 0.47 | 0.99 | 1.90 | 3.96 | 28.8 |
| ZEPHYRUS-REFLECTIVE | gpt-oss-120b | **0.01** | **0.04** | **0.11** | 1.50 | 6.31 |
| ZEPHYRUS-DIRECT | gpt-oss-120b | **0.01** | **0.04** | **0.11** | 1.50 | 6.31 |
| Text Only LLM | gpt-oss-120b | 0.94 | 0.94 | 0.94 | **0.94** | **0.00** |
| ZEPHYRUS-REFLECTIVE | gpt-5.2 | **0.00** | **0.02** | **0.07** | **0.21** | **5.84** |
| ZEPHYRUS-DIRECT | gpt-5.2 | 0.02 | 0.04 | **0.07** | **0.21** | 6.19 |
| Text Only LLM | gpt-5.2 | 0.26 | 0.46 | 1.33 | 2.70 | 172.0 |
| ZEPHYRUS-REFLECTIVE | qwen3-30b | **0.00** | **0.02** | **0.07** | **0.21** | **5.84** |
| ZEPHYRUS-DIRECT | qwen3-30b | 0.02 | 0.06 | 0.15 | 1.49 | 6.52 |
| Text Only LLM | qwen3-30b | 0.98 | 7.41 | 20.0 | 875,000 | 307.0 |

Table 59: Standardized Absolute Error (SAE) quantiles for Template ID 41: `What is the mean weather variable where another variable exceeds median (forecast)`

| Model | LLM | SAE (Q25)(↓) | SAE (Q50)(↓) | SAE (Q75)(↓) | SAE (Q99)(↓) | Err Std(↓) |
|---|---|---|---|---|---|---|
| ZEPHYRUS-REFLECTIVE | gpt-5-mini | **0.06** | **0.11** | **0.36** | **1.19** | 251.0 |
| ZEPHYRUS-DIRECT | gpt-5-mini | **0.06** | 0.14 | **0.36** | **1.19** | 352.0 |
| Text Only LLM | gpt-5-mini | 0.40 | 0.59 | 1.23 | 17.6 | 1,990 |
| ZEPHYRUS-REFLECTIVE | gemini-2.5-flash | **0.06** | 0.15 | 0.38 | **0.97** | 270.0 |
| ZEPHYRUS-DIRECT | gemini-2.5-flash | **0.06** | **0.14** | **0.36** | 1.19 | **252.0** |
| Text Only LLM | gemini-2.5-flash | 0.55 | 1.20 | 2.42 | 29.2 | 58,500 |
| ZEPHYRUS-REFLECTIVE | gpt-oss-120b | **0.06** | 0.14 | **0.36** | 1.19 | 250.0 |
| ZEPHYRUS-DIRECT | gpt-oss-120b | **0.06** | **0.11** | **0.36** | 1.18 | 251.0 |
| Text Only LLM | gpt-oss-120b | 2.70 | 4.38 | 6.06 | 7.68 | **0.00** |
| ZEPHYRUS-REFLECTIVE | gpt-5.2 | **0.06** | **0.11** | **0.36** | **1.19** | 251.0 |
| ZEPHYRUS-DIRECT | gpt-5.2 | **0.06** | **0.11** | **0.36** | **1.19** | **251.0** |
| Text Only LLM | gpt-5.2 | 0.26 | 0.79 | 1.15 | 1.71 | 1,760 |
| ZEPHYRUS-REFLECTIVE | qwen3-30b | 0.18 | 0.24 | 0.68 | 3.12 | **366.0** |
| ZEPHYRUS-DIRECT | qwen3-30b | **0.09** | **0.17** | **0.41** | **1.31** | 372.0 |
| Text Only LLM | qwen3-30b | 4.58 | 13.7 | 28.6 | 63,500 | 68,300 |

Table 60: Standardized Absolute Error (SAE) quantiles for Template ID 42: `What is the weather variable value where another variable reaches maximum (forecast)`

| Model | LLM | Score (mean)(↑) | Score (median)(↑) |
|---|---|---|---|
| ZEPHYRUS-REFLECTIVE | gpt-5-mini | **0.02** | **0.00** |
| ZEPHYRUS-DIRECT | gpt-5-mini | 0.00 | **0.00** |
| Text Only LLM | gpt-5-mini | 0.00 | **0.00** |
| ZEPHYRUS-REFLECTIVE | gemini-2.5-flash | **0.00** | **0.00** |
| ZEPHYRUS-DIRECT | gemini-2.5-flash | **0.00** | **0.00** |
| Text Only LLM | gemini-2.5-flash | **0.00** | **0.00** |
| ZEPHYRUS-REFLECTIVE | gpt-oss-120b | **0.00** | **0.00** |
| ZEPHYRUS-DIRECT | gpt-oss-120b | **0.00** | **0.00** |
| Text Only LLM | gpt-oss-120b | **0.00** | **0.00** |
| ZEPHYRUS-REFLECTIVE | gpt-5.2 | **0.04** | **0.00** |
| ZEPHYRUS-DIRECT | gpt-5.2 | 0.02 | **0.00** |
| Text Only LLM | gpt-5.2 | 0.00 | **0.00** |
| ZEPHYRUS-REFLECTIVE | qwen3-30b | **0.00** | **0.00** |
| ZEPHYRUS-DIRECT | qwen3-30b | **0.00** | **0.00** |
| Text Only LLM | qwen3-30b | **0.00** | **0.00** |

Table 61: Discussion score metrics for Template ID 43: `Generate comprehensive global climate forecast for temperature and precipitation for next 3 months (forecast)`

| Model | LLM | Score (mean)(↑) | Score (median)(↑) |
|---|---|---|---|
| ZEPHYRUS-REFLECTIVE | gpt-5-mini | **0.40** | **0.42** |
| ZEPHYRUS-DIRECT | gpt-5-mini | 0.12 | 0.10 |
| Text Only LLM | gpt-5-mini | 0.09 | 0.04 |
| ZEPHYRUS-REFLECTIVE | gemini-2.5-flash | **0.17** | **0.15** |
| ZEPHYRUS-DIRECT | gemini-2.5-flash | 0.11 | 0.09 |
| Text Only LLM | gemini-2.5-flash | 0.07 | 0.00 |
| ZEPHYRUS-REFLECTIVE | gpt-oss-120b | **0.16** | **0.09** |
| ZEPHYRUS-DIRECT | gpt-oss-120b | 0.10 | 0.04 |
| Text Only LLM | gpt-oss-120b | 0.03 | 0.00 |
| ZEPHYRUS-REFLECTIVE | gpt-5.2 | **0.43** | **0.46** |
| ZEPHYRUS-DIRECT | gpt-5.2 | 0.26 | 0.30 |
| Text Only LLM | gpt-5.2 | 0.14 | 0.12 |
| ZEPHYRUS-REFLECTIVE | qwen3-30b | **0.10** | 0.02 |
| ZEPHYRUS-DIRECT | qwen3-30b | 0.09 | **0.03** |
| Text Only LLM | qwen3-30b | 0.03 | 0.00 |

Table 62: Discussion score metrics for Template ID 44: `Provide detailed meteorological discussion and forecast for continental United States (forecast)`

| Model | LLM | Score (mean)(↑) | Score (median)(↑) |
|---|---|---|---|
| ZEPHYRUS-REFLECTIVE | gpt-5-mini | **0.43** | **0.41** |
| ZEPHYRUS-DIRECT | gpt-5-mini | 0.25 | 0.22 |
| Text Only LLM | gpt-5-mini | 0.19 | 0.13 |
| ZEPHYRUS-REFLECTIVE | gemini-2.5-flash | **0.17** | 0.08 |
| ZEPHYRUS-DIRECT | gemini-2.5-flash | 0.14 | **0.10** |
| Text Only LLM | gemini-2.5-flash | 0.13 | 0.05 |
| ZEPHYRUS-REFLECTIVE | gpt-oss-120b | **0.26** | **0.25** |
| ZEPHYRUS-DIRECT | gpt-oss-120b | 0.21 | 0.18 |
| Text Only LLM | gpt-oss-120b | 0.07 | 0.00 |
| ZEPHYRUS-REFLECTIVE | gpt-5.2 | **0.47** | **0.49** |
| ZEPHYRUS-DIRECT | gpt-5.2 | 0.28 | 0.27 |
| Text Only LLM | gpt-5.2 | 0.26 | 0.29 |
| ZEPHYRUS-REFLECTIVE | qwen3-30b | **0.11** | 0.02 |
| ZEPHYRUS-DIRECT | qwen3-30b | **0.11** | **0.03** |
| Text Only LLM | qwen3-30b | **0.11** | 0.01 |

Table 63: Discussion score metrics for Template ID 45: `Generate ENSO climate update and outlook based on atmospheric data (forecast)`

| Model | LLM | SAE (Q25)(↓) | SAE (Q50)(↓) | SAE (Q75)(↓) | SAE (Q99)(↓) | Err Std(↓) |
|---|---|---|---|---|---|---|
| ZEPHYRUS-REFLECTIVE | gpt-5-mini | **0.00** | **0.04** | **0.09** | **0.36** | **292.0** |
| ZEPHYRUS-DIRECT | gpt-5-mini | 0.01 | **0.04** | 0.13 | **0.36** | 357.0 |
| Text Only LLM | gpt-5-mini | 0.24 | 0.29 | 0.34 | 0.40 | 549.0 |
| ZEPHYRUS-REFLECTIVE | gemini-2.5-flash | 0.02 | 0.08 | 0.22 | 7.84 | 7,330 |
| ZEPHYRUS-DIRECT | gemini-2.5-flash | **0.01** | **0.06** | **0.16** | 0.68 | **416.0** |
| Text Only LLM | gemini-2.5-flash | 0.24 | 0.29 | 0.34 | **0.40** | 535.0 |
| ZEPHYRUS-REFLECTIVE | gpt-oss-120b | **0.03** | 0.07 | 0.19 | 1.02 | 352.0 |
| ZEPHYRUS-DIRECT | gpt-oss-120b | **0.03** | **0.06** | **0.17** | 1.10 | **286.0** |
| Text Only LLM | gpt-oss-120b | 0.24 | 0.29 | 0.34 | **0.40** | 549.0 |
| ZEPHYRUS-REFLECTIVE | gpt-5.2 | **0.02** | **0.05** | **0.17** | 0.69 | **449.0** |
| ZEPHYRUS-DIRECT | gpt-5.2 | **0.02** | **0.05** | **0.17** | 4,710 | 2.57e+06 |
| Text Only LLM | gpt-5.2 | 0.24 | 0.29 | 0.34 | **0.40** | 549.0 |
| ZEPHYRUS-REFLECTIVE | qwen3-30b | 0.05 | 0.16 | 0.39 | **22.4** | 25,500 |
| ZEPHYRUS-DIRECT | qwen3-30b | 0.04 | **0.13** | **0.30** | 1,170 | 57,900 |
| Text Only LLM | qwen3-30b | **0.00** | 0.28 | 0.36 | 10,500 | **372.0** |

Table 64: Standardized Absolute Error (SAE) quantiles for Template ID 46: `How will weather variable change after specified time given an intervention (counterfactual)`

| Model | LLM | MAE (median)(↓) |
|---|---|---|
| ZEPHYRUS-REFLECTIVE | gpt-5-mini | **0.12** |
| ZEPHYRUS-DIRECT | gpt-5-mini | 0.15 |
| Text Only LLM | gpt-5-mini | 0.27 |
| ZEPHYRUS-REFLECTIVE | gemini-2.5-flash | 0.08 |
| ZEPHYRUS-DIRECT | gemini-2.5-flash | **0.07** |
| Text Only LLM | gemini-2.5-flash | 0.24 |
| ZEPHYRUS-REFLECTIVE | gpt-oss-120b | **0.22** |
| ZEPHYRUS-DIRECT | gpt-oss-120b | 0.24 |
| Text Only LLM | gpt-oss-120b | 0.30 |
| ZEPHYRUS-REFLECTIVE | gpt-5.2 | **0.20** |
| ZEPHYRUS-DIRECT | gpt-5.2 | 0.28 |
| Text Only LLM | gpt-5.2 | 0.29 |
| ZEPHYRUS-REFLECTIVE | qwen3-30b | 0.36 |
| ZEPHYRUS-DIRECT | qwen3-30b | 0.37 |
| Text Only LLM | qwen3-30b | **0.26** |

Table 65: Performance metrics for Template ID 47: `What is the value of the input parameter of the simulator model that produces the simulation output`

| Model | LLM | Correctness (%)(↑) | EMD (km)(↓) | IoU (%)(↑) |
|---|---|---|---|---|
| ZEPHYRUS-REFLECTIVE | gpt-5-mini | 14.3 | 8,626 | **0.00** |
| ZEPHYRUS-DIRECT | gpt-5-mini | 11.9 | **6,572** | **0.00** |
| Text Only LLM | gpt-5-mini | **28.6** | 6,699 | **0.00** |
| ZEPHYRUS-REFLECTIVE | gemini-2.5-flash | 19.0 | **5,409** | **0.00** |
| ZEPHYRUS-DIRECT | gemini-2.5-flash | 11.9 | 5,932 | **0.00** |
| Text Only LLM | gemini-2.5-flash | **28.6** | – | **0.00** |
| ZEPHYRUS-REFLECTIVE | gpt-oss-120b | 14.3 | **5,795** | **0.00** |
| ZEPHYRUS-DIRECT | gpt-oss-120b | 14.3 | 5,918 | **0.00** |
| Text Only LLM | gpt-oss-120b | **28.6** | – | **0.00** |
| ZEPHYRUS-REFLECTIVE | gpt-5.2 | 19.0 | **6,832** | **0.00** |
| ZEPHYRUS-DIRECT | gpt-5.2 | 19.0 | 7,553 | **0.00** |
| Text Only LLM | gpt-5.2 | **28.6** | – | **0.00** |
| ZEPHYRUS-REFLECTIVE | qwen3-30b | 23.8 | **4,443** | **0.00** |
| ZEPHYRUS-DIRECT | qwen3-30b | 19.0 | 6,910 | **0.00** |
| Text Only LLM | qwen3-30b | **28.6** | – | **0.00** |

Table 66: Extreme weather detection metrics for Template ID 48: `Where is a target disaster currently happening`

| Model | LLM | F1 Score(↑) | Precision (%)(↑) | Recall (%)(↑) |
|---|---|---|---|---|
| ZEPHYRUS-REFLECTIVE | gpt-5-mini | 32.4 | 61.1 | 22.0 |
| ZEPHYRUS-DIRECT | gpt-5-mini | 33.1 | **66.7** | 22.0 |
| Text Only LLM | gpt-5-mini | **48.2** | 49.5 | **47.0** |
| ZEPHYRUS-REFLECTIVE | gemini-2.5-flash | **37.3** | 51.9 | **29.2** |
| ZEPHYRUS-DIRECT | gemini-2.5-flash | 33.6 | **62.2** | 23.0 |
| Text Only LLM | gemini-2.5-flash | 7.10 | 33.3 | 4.00 |
| ZEPHYRUS-REFLECTIVE | gpt-oss-120b | **29.8** | **51.2** | **21.0** |
| ZEPHYRUS-DIRECT | gpt-oss-120b | 26.5 | 50.0 | 18.0 |
| Text Only LLM | gpt-oss-120b | 0.00 | 0.00 | 0.00 |
| ZEPHYRUS-REFLECTIVE | gpt-5.2 | **26.2** | **56.7** | **17.0** |
| ZEPHYRUS-DIRECT | gpt-5.2 | **26.2** | **56.7** | **17.0** |
| Text Only LLM | gpt-5.2 | 7.00 | 26.7 | 4.00 |
| ZEPHYRUS-REFLECTIVE | qwen3-30b | 15.6 | 29.4 | 10.6 |
| ZEPHYRUS-DIRECT | qwen3-30b | **33.8** | **55.8** | **24.2** |
| Text Only LLM | qwen3-30b | 0.00 | 0.00 | 0.00 |

Table 67: Boolean classification metrics for Template ID 49: `Check whether the given claim extracted from meteorological report is supported by the data`

