# OpenReview forum: "Zephyrus: An Agentic Framework for Weather Science"
_ICLR.cc/2026/Conference — ICLR 2026 Poster_

### Official Review · Reviewer_nucp · 2025-10-31

**Soundness:** 4
**Presentation:** 3
**Contribution:** 4
**Rating:** 8
**Confidence:** 4

**Summary:**

This is a very. nice study that creates two contributions that are novel and difficult: the first (zephyrusworld) is a system for evaluating weather focused tasks by LLM agents. This is difficult because of the data compute nature of these questions. The second is the
benchmark built on ERA5 combining questions from humans and synthetic creation to cover a wide range of weather probelms with varying difficulty. I'm impressed especially with the hard questions-- this goes beyond what is usually done. Graduate students wrote python code to solve the problems. The authors used these together to evaluate a set of frontier LLMs on the tasks.

**Strengths:**

The strength is the difficulty of the tasks that are created both for the execution environment and for the benchmark itself. Both of these are impressive. These problems are not easy to set up or to create. Expertise is required for both. It is this difficulty that differentiates this paper from the large number of  other benchmarking papers.  I also like the creativity of the tasks--particularly meteorological claim verification, looking for data to support the claims from weather reports.

The evaluation metrics chosen are quantitative and rigorous (eg not employing LLM as judges etc)

The paper also contains a substantive supplementary information that gives lots of detail on tasks, performance breakdowns, etc.

**Weaknesses:**

the number of different tasks is pretty limited -- 46 total of which 30 are synthetic. Yet I acknowledge how hard it is to create tasks and so this is an understandable weakness. I think there is a lot of creativity in the question generation.

Another weakness is that the benchmark was built to a large extent for teh set of tools that the LLM has access to. Thus I can't tell how real the results are: what if the LLM were driving a real GCM, would the results be different? What matters for the quality of ther esults?

10 or so of the tasks are yes/no questions -- evaluation is much easier here, one can just guess ;-) based on climolotagical prior

Baselines aren't given other than the raw text LLM. What about a LLM given access only to climatology? ie to what extent is performance due to memorized patterns versus real skill with data analysis

**Questions:**

I dont understand how your choice of solvers (GCM simulator, geolocator,Forecaster) affect the scores on the tasks. Which of these were important for which tasks?

There could be more ablation studies showing why the scores are what they are. What if you delete tools?

Stronger baselines -- What about a LLM given access only to climatology? ie to what extent is performance due to memorized patterns versus real skill with data analysis. Are there other good baselines?

---

> ### Author Response · Authors · 2025-12-03
>
> We thank the reviewer for their insightful comments and questions. Below, we address the major weaknesses and comments raised:
>
> > the number of different tasks is pretty limited -- 46 total of which 30 are synthetic. Yet I acknowledge how hard it is to create tasks and so this is an understandable weakness. I think there is a lot of creativity in the question generation
>
> We appreciate the acknowledgment of the difficulty and the creativity of the tasks. As noted in responses to other reviewers, our design choice was to prioritize scientific correctness and depth over raw template count. We would like to note that ZephyrusBench is designed to be extensible, with the provided templates and infrastructure serving as a foundation for future community expansion.
>
> > Another weakness is that the benchmark was built to a large extent for teh set of tools that the LLM has access to. Thus I can't tell how real the results are: what if the LLM were driving a real GCM, would the results be different? What matters for the quality of ther esults?
> > I dont understand how your choice of solvers (GCM simulator, geolocator,Forecaster) affect the scores on the tasks. Which of these were important for which tasks?
> > There could be more ablation studies showing why the scores are what they are. What if you delete tools?
>
> We thank the reviewer for their useful suggestion. Based on your suggestion and that of reviewer gtMv31, we ran ablations with one-tool-excluded baselines to better understand the role of the tools for each considered task. Please see the response to review gtMv31 W2 for a detailed discussion of the effect of tools on different task scores.
>
> > 10 or so of the tasks are yes/no questions -- evaluation is much easier here, one can just guess ;-) based on climolotagical prior
>
> We note that while evaluation is easier for these yes/no questions, the answers themselves are quite difficult to guess accurately even with the climatology prior. For example, in the claim verification task (Table 22), none of the models achieve an F1 score meaningfully above 0.5, indicating that they are as good as random guessing.
>
> > Baselines aren't given other than the raw text LLM. What about a LLM given access only to climatology? ie to what extent is performance due to memorized patterns versus real skill with data analysis
>
> Thank you for this suggestion! We have added an additional Climatology tool to all models based that calculates mean, max, min, median, 1%, 5%, 10%, 90%, 95%, 99% quantiles across all dataset variables, grouped by all time, month, day and 6 hour intervals. We report all results with this tool included by default, unless stated otherwise.
>
> > Stronger baselines -- What about a LLM given access only to climatology? ie to what extent is performance due to memorized patterns versus real skill with data analysis. Are there other good baselines?
>
> To address this question, we ran a baseline experiment where the ZephyrusReflective model was given access only to the Climatology and Geolocator tools, without direct access to the dataset or other tools. For gpt-5-mini ZephyrusReflective run on the 500-task subset, we gathered these summary results:
> | Tools               | % Correct | Easy   | Medium | Hard   | Location Acc | EMD     | Valid % |
> | ------------------- | --------- | ------ | ------ | ------ | ------------ | ------- | ------- |
> | all tools           | 59.40%    | 93.83% | 53.07% | 32.08% | 93.22%       | 1675.64 | 99.23%  |
> | only climatology + geolocator | 21.64% | 16.39% | 35.58% | 9.03% | 32.08% | 4633.82 | 93.99% |
>
> Performance drops drastically, indicating that the task answers cannot be inferred solely from climatological priors.

---

### Official Review · Reviewer_gtMv · 2025-10-31

**Soundness:** 3
**Presentation:** 3
**Contribution:** 2
**Rating:** 6
**Confidence:** 3

**Summary:**

The paper introduces Zephyrus, an “agentic” framework designed to enable large language models (LLMs) to interact programmatically with meteorological data. The system includes tools to directly interact with weather and geographic data, and forecasting and simulator models.

The authors also curate a weather reasoning benchmark with 2062 question answer pairs in 46 meteorological tasks. The authors show that their system Zephyrus-Reflective which allows observations and reflections is better than Zephyrus-Direct which only allows the system to execute in one step. Both these systems are better than a text-only baseline on easy and medium tasks but do not show improvement on hard tasks.

**Strengths:**

S1. The paper clearly articulates a gap between numerical weather foundation models and language-based reasoning systems. Positioning the work at this intersection is timely and relevant, especially given the rising interest in LLM-augmented science workflows.

S2. The paper presents a very reasonable set of tools and agentic systems (Zephyrus-Reflective and Zephyrus-Direct) for reasoning over meteorological questions and data and is a solid first step in this direction.

S3. The dataset is well-constructed and leverages a good mix of human expertise and LLM aided construction.

**Weaknesses:**

W1.  The authors do not report error bars or statistical significance. This makes it hard to assess whether observed performance differences are meaningful or consistent across runs.

W2. The observation that adding tools leads to better performance or allowing observations and actions in a multiple steps improves performance is not novel. From my perspective, the main contribution of the work is the incorporation of relevant and helpful tools in the agentic workflow to begin with. Thus, it would be more insightful to see how different baseline tools or the inclusion/exclusion of tools impact performance in Zephyrus-Reflective.

W3. The analysis could be strengthened to understand why the agent succeeds or fails. Given that models all perform roughly equal on the harder subset, are we limited by the tool? agentic function calling? or the LLM reasoning? Such an analysis could make more clear what is needed to improve.

**Questions:**

What is the prompt for the text-only baseline?

Can you explain why text-only seems to be better in Table 16, 17, and 18? The performance trends also appear inconsistent between different models.

---

> ### Author Response · Authors · 2025-12-03
>
> We thank the reviewer for their thorough review and comments. Below, we address the main weaknesses and questions:
>
> > W1. The authors do not report error bars or statistical significance. This makes it hard to assess whether observed performance differences are meaningful or consistent across runs.
>
> We agree that uncertainty estimates are important.  Due to the high API cost of frontier models, we did not run multiple random seeds for all models. However, to estimate the variance in performance across multiple runs, we ran the ZephyrusReflective model with the gpt-5-mini LLM three times on the 500-sample subset, yielding the following metrics:
>
> | Run | % Correct | Easy | Medium | Hard | SAE (Median) | Location Acc | EMD | Valid % | Extreme F1 | Boolean F1 |
> | --- | --- | --- | --- | --- | --- | --- | --- | --- | --- | --- |
> | Run 1 | 59.92% | 92.90% | 53.85% | 28.39% | 0.01 | 90.57% | 1787.12 | 99.40% | 0.00 | 0.52 |
> | Run 2 | 61.52% | 93.44% | 54.81% | 29.68% | 0.01 | 92.45% | 1662.80 | 99.20% | 0.00 | 0.53 |
> | Run 3 | 60.32% | 91.26% | 55.77% | 27.74% | 0.01 | 96.23% | 1776.55 | 99.00% | 0.00 | 0.56 |
>
> Cross-run Variance Summary (mean ± std):
>
> - Questions Correct: 60.59 ± 0.83 (n=3)
> - SAE (median): 0.01 ± 0.00 (n=3)
> - Location Accuracy: 93.08 ± 2.88 (n=3)
> - Emd Score: 1742.16 ± 68.93 (n=3)
> - Valid Outputs: 99.20 ± 0.20 (n=3)
> - Extreme Weather F1: 0.00 ± 0.00 (n=3)
> - Boolean F1: 0.54 ± 0.02 (n=3)
> - Easy Questions Correct: 92.53 ± 1.14 (n=3)
> - Medium Questions Correct: 54.81 ± 0.96 (n=3)
> - Hard Questions Correct: 28.60 ± 0.99 (n=3)
>
> Some variance exists across runs, but it remains minor and within reasonable thresholds.

---

> ### Author Response · Authors · 2025-12-03
>
> > W2. The observation that adding tools leads to better performance or allowing observations and actions in a multiple steps improves performance is not novel. From my perspective, the main contribution of the work is the incorporation of relevant and helpful tools in the agentic workflow to begin with. Thus, it would be more insightful to see how different baseline tools or the inclusion/exclusion of tools impact performance in Zephyrus-Reflective.
>
> In addition to tool-usage statistics as recommended by Reviewer GNmT, we ran ablation experiments with one-tool-excluded baselines to determine the effectiveness of the different tools for the different tasks in the benchmark. These experiments were run on the 500-task subset using the gpt-5-mini LLM and ZephyrusReflective model. Results are summarized here:
>
> | Tools | % Correct | Easy | Medium | Hard | Location Acc | EMD | Valid % |
> | --- | --- | --- | --- | --- | --- | --- | --- |
> | All Tools | 59.92% | 92.90% | 53.85% | 28.39% | 90.57% | 1787.12 | 99.40% |
> | Exclude Forecaster, Simulator | 54.91% | 92.90% | 44.23% | 14.19% | 92.45% | 2031.75 | 97.19% |
> | Exclude Geolocator | 38.68% | 36.07% | 44.23% | 32.90% | 18.87% | 3925.74 | 96.79% |
> | Exclude Forecaster | 59.32% | 91.89% | 33.64% | 37.67% | 94.44% | 1655.26 | 98.20% |
> | Exclude Simulator | 60.52% | 92.90% | 54.81% | 26.45% | 94.34% | 1877.90 | 98.80% |
> | Exclude Ds, Forecaster, Simulator | 21.64% | 16.39% | 35.58% | 9.03% | 32.08% | 4633.82 | 93.99% |
> | Exclude Climatology | 53.31% | 77.05% | 51.92% | 27.74% | 98.11% | 1899.39 | 99.80% |
>
> While the decrease in correctness percentage from excluding the geolocator and climatology tools were expected, we were surprised by the relatively strong correctness percentage when excluding the simulator or forecaster. We examined the per-question performance metrics and found that the presence of multiple tools can be confusing to the model for certain question types.
>
> One performance change is visible for counterfactual simulation (template ID 44), where the climate simulator must be used to answer the question correctly:
> | Tools | % Correct | Valid % |
> | --- | --- | --- |
> | All Tools | 57.89% | 100.00% |
> | Exclude Forecaster, Simulator | 24.56% | 92.98% |
> | Exclude Geolocator | 64.91% | 94.74% |
> | Exclude Forecaster | 79.66% | 96.61% |
> | Exclude Simulator | 50.88% | 96.49% |
> | Exclude Ds, Forecaster, Simulator | 1.75% | 98.25% |
> | Exclude Climatology | 57.89% | 100.00% |
>
> Model performance increases significantly when the forecaster tool is excluded, suggesting that the model often incorrectly attempts to use the forecaster tool for the climate simulation task.
>
> For a basic forecasting task (template ID 6), we confirmed that the forecaster tool is necessary for accurate performance:
> | Tools | % Correct | Valid % | N |
> | --- | --- | --- | --- |
> | All Tools | 47.06% | 100.00% | 17 |
> | Exclude Forecaster, Simulator | 23.53% | 100.00% | 17 |
> | Exclude Geolocator | 41.18% | 100.00% | 17 |
> | Exclude Forecaster | 11.76% | 100.00% | 17 |
> | Exclude Simulator | 47.06% | 100.00% | 17 |
> | Exclude Ds, Forecaster, Simulator | 11.76% | 88.24% | 17 |
> | Exclude Climatology | 35.29% | 100.00% | 17 |
>
> Task performance drops almost 36% when excluding the forecaster tool.
>
> > W3. The analysis could be strengthened to understand why the agent succeeds or fails. Given that models all perform roughly equal on the harder subset, are we limited by the tool? agentic function calling? or the LLM reasoning? Such an analysis could make more clear what is needed to improve.
>
> We thank the reviewer for this suggestion. As noted in the response to Reviewer GNmT, we will highlight some exemplar failure modes of the models on challenging tasks to highlight some shortcomings of the models.

---

> ### Author Response · Authors · 2025-12-03
>
> > What is the prompt for the text-only baseline?
>
> We use the following prompt to force the LLM to provide an answer that we can evaluate consistently, in line with the other methods we benchmark in our work.
>
> ```
> Provide an answer to the following question.
> Do not complain about insufficient data.
> Be brief, and provide a 1-2 sentence answer.
> Always provide your answer in SI units.
> Do not give approximate answers.
> If the question asks for the time offset, you should return the answer in hours from the initial time index.
> For example, if the question asks about a dataset with time interval 6 hours and time indices 12345:12351:1, and you think the answer is index 12350, you should return 30 hours.
> Do NOT return the time index as a timestamp or datetime object.
> Provide a concrete final answer. It is unacceptable to say "Without data I cannot answer this question".
> {question}
> ```
>
> > Can you explain why text-only seems to be better in Table 16, 17, and 18? The performance trends also appear inconsistent between different models.
>
> We thank the review for bringing up this subtle observation. On re-inspection, we found that the original construction of one extreme-weather task had imbalanced labels, with negative samples (no extreme event) significantly outnumbering positive samples. In this setting, the text-only baseline could obtain deceptively high scores by effectively learning to always answer “no” based on prior knowledge rather than data.
> We have since rebalanced the extreme-event dataset to make positive and negative instances more comparable, and updated the extreme-weather question formulation. These changes have removed the misleading advantage of the text-only baseline and made the tasks significantly more difficult.  We will also report Earth Mover’s Distance (EMD) scores for these tasks to capture not only whether an extreme event is detected but also whether it is localized correctly.
> Results for the new extreme weather tasks are available [here](https://freeimage.host/i/fzWntDl).
> The new task formulations are extremely difficult, with models achieving near-zero success rates.
> Regarding Table 18, for the task “Which geographic features experienced unusual weather anomalies compared to baseline”, we have clarified the question formulation. With the update and the newly added Climatology tool, results follow a more expected pattern, with the Text Only LLM unable to achieve above-trivial success. Results are [here](https://freeimage.host/i/fzWaalR)

---

### Official Review · Reviewer_M2k6 · 2025-11-02

**Soundness:** 3
**Presentation:** 4
**Contribution:** 3
**Rating:** 8
**Confidence:** 3

**Summary:**

This paper introduces an agentic framework - Zephyrus - to enable LLMs to interact with weather data via many datasets/tools to address weather science tasks. They also introduces a benchmark (ZephyrusBench) to evaluate how well frontier LLMs can assist with whether science queries/tasks both directly and within their agentic framework. Their agentic framework includes access to different tools to address weather queries including observation data (from ERA5 via WeatherBench2), Geolocator, forecasting models, simulation models, as well as a practical and fast code execution environment.

To create the ZephyrusBench benchmark, they had 2 approaches one that involved human annotation and another that used a semi-synthetic pipeline to generate queries. For the human generated, graduate students come up with 15 task types (5 types each in Easy, Medium, Hard category) and associated question template and solution code. They then substitute the variables in the templates with different parameters e.g. location etc. to generate a wide number (~1.8k) of queries. For the semi-synthetic pipeline they extract claims from papers to formulate queries/tasks and create 31 task types (30 medium, 1 hard) and similar to the human version, substitute variables to create ~300 questions.

They compare performance of the frontier-LLMs in 3 settings: (1) text-only: here the LLM does not get  access to any any external resources/tools, (2) Zephyrus-direct: here the LLM gets to generate the entire code for the solution in a single response turn, but is allowed to get execution feedback and correct errors upto 5 times. (3) Zephyrus-reflective: here the LLM gets to build out the code solution in chunks in a multi-turn fashion, getting the execution outputs for each code chunk which would allow the LLM to plan and break the task into components (and they allow error correction for upto 20 times).

**Strengths:**

* Good weather science agentic framework contribution: The paper introduces a practical agentic framework for weather science that integrates a number of different tools for getting observation data, Geolocator, forecasting models, simulation models, as well as a fast execution environment to run model generated code to compute the solution.

* Interesting and large benchmark to evaluate weather science agents. The paper has a thoughtful approach to curating a set of 46 tasks /templates and over 2100 questions. They used both human annotation to generate queries and solutions, and also an interesting semi-synthetic pipeline to generate some of the questions (see summary).

* Their evaluation set up to compare frontier LLMs without access to external resources, and with access to their agentic framework under different computational resources settings seems sound and the results are meaningful clearly highlighting the value of their dataset and the agentic framework.

* the paper is written well and is easy to follow.

**Weaknesses:**

1. The benchmark creation has humans generate templates and task types to enable semi-automated creation of questions (with suitable solutions). I wonder if this perhaps reduces the diversity that one might observe in realistic settings which may deviate substantially from the templates,

2. The results could be analyzed a few additional ways e.g. based on synthetic tasks and human tasks, and also by 2-3 task-types to see if these chracterizations of the benchmark can provide more insights.

3. It's not entirely clear that there is much of a difference between Zephyris-direct and Zephyrus-reflective agentic settings. Perhaps the small difference could be that Zephyrus-direct gets to make fewer model queries and lower amount of error correction. It's not exactly a major weakness but a more interesting experiment would be to see if the agentic environment would infact benefit from planning as part of the agentic framework or if current "reasoning"/"thinking" models are capable of doing the reasoning fully by themselves.

4. It would be nice to see at least one open source LLM performance in the agentic framework.

**Questions:**

Please address the weaknesses.

1. Since the questions are templated based on tasks, do you see similar performance on most questions of a task type?

2. Can you separate evaluation for synthetic queries vs human queries.

3. Having built an interesting semi-synthetic task/query generation pipeline, why not generate more synthetic tasks/questions? were there any issues? Is there value in having more queries?

4. Related to weakness-3. It appears that there may not be quite as much of a difference between the Zephyrus-direct and Zephyrus-reflective agents and the small difference we see may be attributed to differences in # of LLM queries and error-corrections allowed in both settings. So, question: what if you gave Zephyrus-direct ~100 error correction attempts (to compensate both increasing LLM queries and opportunities for error feedback), would it be on-par or better than Zephyrus-reflective?

---

> ### Author Response · Authors · 2025-12-03
>
> We thank the reviewer for their insightful comments and questions. Below we address the main weaknesses and comments brought up:
>
> > 1. The benchmark creation has humans generate templates and task types to enable semi-automated creation of questions (with suitable solutions). I wonder if this perhaps reduces the diversity that one might observe in realistic settings which may deviate substantially from the templates
>
>
> We would like to mention that in general, it is challenging to automate the creation of diverse weather tasks that were simultaneously (1) interesting to a weather scientist (2) programmatically verifiable (3) sufficiently diverse and different from one another. The most reliable way of doing this was through the human-written templates, which were constructed in consultation with domain experts. To partially compensate for the limited number of templates, we systematically vary parameters (regions, time ranges, thresholds, etc.) so that each template spans a family of related but non-identical queries. As noted in the response to Reviewer GNmT, per-template results demonstrate non-trivial variability across instances, indicating that the benchmark is not simply testing a single pattern repeated 46 times.
>
>
>
> > 2. The results could be analyzed a few additional ways e.g. based on synthetic tasks and human tasks, and also by 2-3 task-types to see if these characterizations of the benchmark can provide more insights.
>
> > 2. Can you separate evaluation for synthetic queries vs human queries.
>
>
>
> Thanks! Based on your suggestion, we report the results for synthetic and human-generated tasks separately
>
> ### Results on human-generated tasks
>
> | Model           | LLM            | % Correct | % Valid Outputs | Discussion Score | Boolean F1 |
> |----------------|----------------|----------:|----------------:|-----------------:|-----------:|
> | Zephyrus-Reflective | gpt-5-mini     |     60.8 |            94.5 |            0.30 |        -- |
> | Zephyrus-Direct     | gpt-5-mini     |     59.2 |            94.1 |            0.18 |        -- |
> | Text Only LLM   | gpt-5-mini     |     10.2 |            94.1 |            0.09 |        -- |
> | Zephyrus-Reflective | gpt-5-nano     |     48.6 |            93.7 |            0.23 |        -- |
> | Zephyrus-Direct     | gpt-5-nano     |     50.4 |            93.4 |            0.11 |        -- |
> | Text Only LLM   | gpt-5-nano     |     10.0 |            93.2 |            0.13 |        -- |
> | Zephyrus-Reflective | gpt-5.1        |     57.9 |            91.9 |            0.39 |        -- |
> | Zephyrus-Direct     | gpt-5.1        |     55.2 |            94.4 |            0.23 |        -- |
> | Text Only LLM   | gpt-5.1        |     11.2 |            93.7 |            0.20 |        -- |
> | Zephyrus-Reflective | gpt-oss-120b   |     52.9 |            93.8 |            0.18 |        -- |
> | Zephyrus-Direct     | gpt-oss-120b   |     54.1 |            90.3 |            0.14 |        -- |
> | Text Only LLM   | gpt-oss-120b   |     10.8 |            84.8 |            0.04 |        -- |
>
> ### Results on Synthetically-generated tasks
> | Model           | LLM            | % Correct | % Valid Outputs | Discussion Score | Boolean F1 |
> |----------------|----------------|----------:|----------------:|-----------------:|-----------:|
> | Zephyrus-Reflective | gpt-5-mini     | 60.8 | 97.3 | -- | 0.81 |
> | Zephyrus-Direct     | gpt-5-mini     | 64.5 | 95.7 | -- | 0.83 |
> | Text Only LLM   | gpt-5-mini     | 44.2 | 98.3 | -- | 0.59 |
> | Zephyrus-Reflective | gpt-5-nano     | 62.1 | 95.3 | -- | 0.82 |
> | Zephyrus-Direct     | gpt-5-nano     | 61.5 | 92.7 | -- | 0.80 |
> | Text Only LLM   | gpt-5-nano     | 49.7 | 97.7 | -- | 0.63 |
> | Zephyrus-Reflective | gpt-5.1        | 63.4 | 97.7 | -- | 0.83 |
> | Zephyrus-Direct     | gpt-5.1        | 62.0 | 97.7 | -- | 0.83 |
> | Text Only LLM   | gpt-5.1        | 41.0 | 98.7 | -- | 0.54 |
> | Zephyrus-Reflective | gpt-oss-120b   | 62.8 | 98.0 | -- | 0.84 |
> | Zephyrus-Direct     | gpt-oss-120b   | 62.7 | 95.0 | -- | 0.83 |
> | Text Only LLM   | gpt-oss-120b   | 44.7 | 86.0 | -- | 0.58 |
>
>
> > It would be nice to see at least one open source LLM performance in the agentic framework.
>
> We agree. Our current draft already includes results for the open-source model gpt-oss-120b in Zephyrus. We are additionally running experiments with more open-source models to make our evaluation more comprehensive.

---

> ### Author Response · Authors · 2025-12-03
>
> > 3. It's not entirely clear that there is much of a difference between Zephyris-direct and Zephyrus-reflective agentic settings. Perhaps the small difference could be that Zephyrus-direct gets to make fewer model queries and lower amount of error correction. It's not exactly a major weakness but a more interesting experiment would be to see if the agentic environment would infact benefit from planning as part of the agentic framework or if current "reasoning"/"thinking" models are capable of doing the reasoning fully by themselves.
>
> We clarify that the only difference is not just that Direct makes fewer model queries, but also that it lacks the ability to reflect on the outputs of the program it generates.
> In preliminary experiments, we tried adding explicit “planning steps” (e.g., separate calls to generate a plan and then implement it) but found that for the strong models we evaluated, this explicit planning often duplicated implicit reasoning and did not significantly improve performance, so we removed it for simplicity.
>
>
> > 1. Since the questions are templated based on tasks, do you see similar performance on most questions of a task type?
>
> In the appendix (Tables 7–22), we report per-template results, and most templates show a non-degenerate “spread” in performance (i.e., far from trivially 0% or 100%).
>
> > 3. Having built an interesting semi-synthetic task/query generation pipeline, why not generate more synthetic tasks/questions? were there any issues? Is there value in having more queries?
>
> The semi-synthetic pipeline is powerful but still requires humans in the loop:
> Many auto-generated tasks are either scientifically trivial, redundant with existing templates, or subtly incorrect.
> Humans are required to vet both the claims and the corresponding solution code for correctness and interest.
>
>
> This manual vetting step limits how far we can scale the semi-synthetic portion for this paper. We agree that there is value in more question templates and view ZephyrusBench as a foundation we can grow in follow-up work.
> > 4. Related to weakness-3. It appears that there may not be quite as much of a difference between the Zephyrus-direct and Zephyrus-reflective agents and the small difference we see may be attributed to differences in # of LLM queries and error-corrections allowed in both settings. So, question: what if you gave Zephyrus-direct ~100 error correction attempts (to compensate both increasing LLM queries and opportunities for error feedback), would it be on-par or better than Zephyrus-reflective?
>
> In our current setup, Zephyrus-Reflective is allowed up to 20 code-generation/execution steps. To make the comparison fairer, we have updated Zephyrus-Direct with an expanded error-correction budget (20 attempts), matching Zephyrus-Reflective’s step limit. Our latest results contain the expanded error-correction budget.
>
> We will also highlight questions which were answered correctly by Zephyrus-Reflective but incorrectly by Zephyrus-Direct to highlight how the self-reflection loop can be beneficial for solving certain complex tasks in the updated version of the paper.

---

### Official Review · Reviewer_GNmT · 2025-11-03

**Soundness:** 3
**Presentation:** 3
**Contribution:** 3
**Rating:** 6
**Confidence:** 3

**Summary:**

This paper presents Zephyrus, an AI agent that can perform tasks in weather science such as identifying weather events from data, an API / code execution server and a new benchmark.

The LLM based agent is connected to a code execution server, that provides access to several weather science related APIs.
The agent uses either single step code generation/execution or multistep, with several execution / result steps before producing the result

The benchmark is produced from two types of task templates:

* manual (grad student) written pairs of template + reference implementation
* semi-synthetic: mines patterns from weather related texts to produce templates and solution code. Templates are human verified

Actual benchmark examples are generated by setting variables such as time and location in the
templates.

Evaluation is done conditional on the answer type, for example locations are resolved and compared with earth movers distance. Text based answer are compared after deconstruction into sets of claims.

Experimental results confirm, as expected, that allowing several agentic steps improves results and that any access to code execution is much better than a non-agentic baseline. An interesting outcome is that this difference gets smaller as the tasks get harder.

**Strengths:**

Significant contributions: Agent, Framework + Benchmark

Overall well written paper with good related work section

**Weaknesses:**

While the overall number of examples in the benchmark is over 2k, these are based on only 46 templates. It is not clear to this reviewer whether having several examples per task improves evaluation strength.

Given that this work evaluates a complex agentic system, it would be interesting to see more in-depth analysis of the behaviour and comparison between common problems between models

Given that the number of examples is not too high, evaluating top line models like GPT5 or o3, Claude Opus or Gemini Pro would be interesting and not too cost intensive.

**Questions:**

From what I understand, all problems for a template can be solved by a similar generated code with filled in variables. Does this mean that the performance of the evaluated models correlated strongly within a template cluster?

Can you provide more statistics on how the APIs were used? Which problems occured, if code errors were correctly fixed from the feedback, etc.

---

> ### Author Response · Authors · 2025-12-03
>
> We thank the reviewer for their insightful review and comments. Below, we address the major weaknesses and questions:
>
> > While the overall number of examples in the benchmark is over 2k, these are based on only 46 templates. It is not clear to this reviewer whether having several examples per task improves evaluation strength.
> > From what I understand, all problems for a template can be solved by a similar generated code with filled in variables. Does this mean that the performance of the evaluated models correlated strongly within a template cluster?
>
> We believe that it is beneficial to have several examples per task for evaluation due to the following reasons:
> 1. We would like to evaluate the ability of models to robustly handle a wide range of input parameters (location, time interval, thresholds, etc.). This helps mitigate biases toward specific locations or dates.
> 2. Multiple instantiations per template allow us to estimate how likely a model is to succeed on that type of question. In the appendix (Tables 7–22), we already report per-template results, and most templates show a non-degenerate “spread” in performance (i.e., far from trivially 0% or 100%).
> 3. For some task families such as report generation or extreme weather prediction, the same template can generate instances that require different code blocks or strategies. Thus, we would like for the evaluation protocol to be as general as possible across all tasks.
>
> > Given that this work evaluates a complex agentic system, it would be interesting to see more in-depth analysis of the behaviour and comparison between common problems between models
> > Can you provide more statistics on how the APIs were used?  Which problems occured, if code errors were correctly fixed from the feedback, etc.
>
> We will update the manuscript with some examples to highlight the failure modes of both Zephyrus-Direct and Zephyrus-Reflective on difficult tasks considered in our benchmark.
>
> We have added tool-usage statistics gathering as a feature in the framework. For a run of ZephyrusReflective over the full dataset (2199 samples) with GPT-5-Mini, we gathered the following summary statistics:
>
> #### **Tool Call Counts**
>   geolocator: 287655
>
>   Dataset access: 3970
>
>   simulator: 2907
>
>   climatology: 2558
>
>   forecaster: 2381
>
> #### **Error Analysis**
> Tasks with code execution errors: 220 (10.0%)
> Errors eventually corrected: 218 (99.1% of errors)
> Errors exceeded attempts allowed: 2
>
> Note that tools can be called multiple times or in a loop by the agent within each code execution block, hence the larger count of tool calls than tasks.
>
>
>
>
> > Given that the number of examples is not too high, evaluating top line models like GPT5 or o3, Claude Opus or Gemini Pro would be interesting and not too cost intensive.
>
> In addition to the models already reported, we have run experiments for gpt-5.1 and are running evaluations on the stronger frontier models mentioned by the reviewer. We will include in the manuscript the latest generation of Gemini (gemini-3-pro) and Claude (sonnet-4.5) models.
>
> A summary comparison of gpt-5.1 results compared to gpt-5-mini is available here, both run in ‘high’ reasoning mode:
>
> | Method | Model | % Correct | Easy | Medium | Hard | SAE (Median) | Location Acc | EMD | Valid % | Extreme F1 | Boolean F1 |
> | --- | --- | --- | --- | --- | --- | --- | --- | --- | --- | --- | --- |
> | Text Only LLM | gpt-5-mini | 17.52% | 9.40% | 30.74% | 8.67% | 0.33 | 14.12% | 6063.02 | 99.45% | 0.00 | 0.49 |
> | Zephyrus-Reflective | gpt-5-mini | 59.40% | 93.83% | 53.07% | 32.08% | 0.01 | 93.22% | 1675.64 | 99.23% | 0.01 | 0.57 |
> | Zephyrus-Direct | gpt-5-mini | 58.40% | 92.93% | 55.62% | 26.83% | 0.01 | 88.95% | 1842.58 | 98.68% | 0.00 | 0.60 |
> | Text Only LLM | gpt-5.1 | 17.11% | 8.72% | 30.12% | 9.96% | 0.33 | 13.56% | 5707.77 | 99.00% | 0.00 | 0.22 |
> | Zephyrus-Reflective | gpt-5.1 | 56.85% | 91.81% | 55.33% | 25.61% | 0.02 | 91.53% | 1147.59 | 97.09% | 0.01 | 0.56 |
> | Zephyrus-Direct | gpt-5.1 | 55.76% | 92.08% | 55.12% | 22.38% | 0.02 | 89.27% | 1148.95 | 99.18% | 0.01 | 0.55 |
>
> Our preliminary results indicate that larger models do not necessarily correlate with better performance on our benchmark. We plan to examine this in more detail with experiments on other LLM families.

---

### Author Response · Authors · 2025-12-03
**Overall Response**

We are extremely grateful for the thoughtful reviews provided for our paper. We have worked to address the questions, concerns and suggestions raised by the reviewers in our response. We are running further experiments and plan to incorporate these results into our next version of the manuscript.

We categorize our changes in response to reviewer feedback as follows:
1. Data regeneration + model re-runs.
2. Evaluating with frontier + more open-source LLMs
3. Ablation Studies with different tools
4. Tool-use and error-correction statistics
5. Addition of new tool (Climatology)


### Data Regeneration + Model Re-runs
We have incorporated several changes in the prompts of many tasks to make the questions clearer and more explicit, and conducted further quality-assurance on both human and semi-synthetic task templates. Following Reviewer gtMv31’s observation about the extreme weather event, we reframed the task to make the setting clearer in the problem formulation, and also rebalanced the dataset to counteract the model’s tendency to guess the answer.

The new dataset statistics are as follows:
| Difficulty | Human-Gen. Tasks | Human-Gen. Samples | Synthetic Tasks | Synthetic Samples | Total Samples |
| --- | --- | --- | --- | --- | --- |
| Easy | 8 | 745 | 0 | 0 | 745 |
| Medium | 3 | 189 | 29 | 299 | 488 |
| Hard | 8 | 775 | 1 | 191 | 966 |
| **Total** | **19** | **1,709** | **30** | **491** | **2,199** |

We also built and included a climatology tool following Reviewer nucp31’s suggestion. These extensive changes necessitated a new run of the experiments. While experiments with all models are currently ongoing, we report the results for the OpenAI family of models on the new dataset in the table below.
| Method | Model | % Correct | Easy | Medium | Hard | SAE (Median) | Location Acc | EMD | Valid % | Extreme F1 | Boolean F1 |
| --- | --- | --- | --- | --- | --- | --- | --- | --- | --- | --- | --- |
| Text Only LLM | gpt-5-mini | 17.52% | 9.40% | 30.74% | 8.67% | 0.33 | 14.12% | 6063.02 | 99.45% | 0.00 | 0.49 |
| Zephyrus-Reflective | gpt-5-mini | 59.40% | 93.83% | 53.07% | 32.08% | 0.01 | 93.22% | 1675.64 | 99.23% | 0.01 | 0.57 |
| Zephyrus-Direct | gpt-5-mini | 58.40% | 92.93% | 55.62% | 26.83% | 0.01 | 88.95% | 1842.58 | 98.68% | 0.00 | 0.60 |
| Text Only LLM | gpt-5-nano | 18.89% | 11.54% | 34.43% | 8.80% | 0.33 | 15.82% | 5226.16 | 98.50% | 0.00 | 0.38 |
| Zephyrus-Reflective | gpt-5-nano | 50.34% | 83.09% | 49.39% | 19.02% | 0.06 | 84.18% | 1463.42 | 98.27% | 0.00 | 0.56 |
| Zephyrus-Direct | gpt-5-nano | 51.43% | 84.74% | 47.11% | 22.25% | 0.03 | 78.33% | 1314.37 | 97.77% | 0.00 | 0.53 |
| Text Only LLM | gpt-5.1 | 17.11% | 8.72% | 30.12% | 9.96% | 0.33 | 13.56% | 5707.77 | 99.00% | 0.00 | 0.22 |
| Zephyrus-Reflective | gpt-5.1 | 56.85% | 91.81% | 55.33% | 25.61% | 0.02 | 91.53% | 1147.59 | 97.09% | 0.01 | 0.56 |
| Zephyrus-Direct | gpt-5.1 | 55.76% | 92.08% | 55.12% | 22.38% | 0.02 | 89.27% | 1148.95 | 99.18% | 0.01 | 0.55 |
| Text Only LLM | openai_gpt-oss-120b | 17.57% | 10.34% | 30.12% | 8.80% | 0.32 | 12.99% | 4103.02 | 89.71% | 0.00 | 0.21 |
| Zephyrus-Reflective | openai_gpt-oss-120b | 53.35% | 89.40% | 56.15% | 18.24% | 0.04 | 82.49% | 1920.53 | 98.77% | 0.02 | 0.56 |
| Zephyrus-Direct | openai_gpt-oss-120b | 52.44% | 88.72% | 53.07% | 17.34% | 0.03 | 72.32% | 1878.24 | 95.45% | 0.00 | 0.56 |

A complete set of performance statistics is available [here](https://freeimage.host/i/fzGtn2a)
Correctness percentage plots are available [here](https://freeimage.host/i/fzMniKu)

### Evaluating with frontier + more open-source LLMs

As seen in the previous table, we include the results with GPT-5.1 (with think tokens enabled).   In addition, we are also conducting experiments with the latest generation of Gemini (gemini-3-pro) and Claude (sonnet-4.5) models. For open-source results, we will add results from the Qwen3 family in addition to our existing gpt-oss-120b results.

### Ablation Studies with Different Tools

Following the suggestions of Reviewers gtMv31 and nucp31, we conduct ablation studies where we exclude one or more tools from our framework. We use a subset of 500 examples taken from the full ZephyrusBench dataset for evaluation. This subset is referred to as the “500 sample subset” in our other comments. Statistics:
| Difficulty | Human-Gen. Tasks | Human-Gen. Samples | Synthetic Tasks | Synthetic Samples | Total Samples |
| --- | --- | --- | --- | --- | --- |
| Easy | 8 | 183 | 0 | 0 | 183 |
| Medium | 3 | 41 | 29 | 63 | 104 |
| Hard | 8 | 156 | 1 | 57 | 213 |
| **Total** | **19** | **380** | **30** | **120** | **500** |
The results are summarized in our response to Reviewer gtMv31.

---

> ### Author Response · Authors · 2025-12-03
>
> ### Tool-use and error-correction statistics
> Following the suggestion from Reviewer GNmT, we report the tool use statistics and error correction attempt statistics for the Zephyrus Reflective (GPT-5-mini) model in our response to reviewer GNmT.
>
> ### Addition of new tool (Climatology)
> Following the suggestion from Reviewer nucp31, we have created and added a new Climatology tool that precomputes various quantities from historical data and enables models to access them with a simple API. More specifically, the climatology tool provides access to mean, max, min, median, 1%, 5%, 10%, 90%, 95%, 99% quantiles across all dataset variables, grouped by all time, month, day and 6 hour intervals, calculated on the reference period 1979-2000. All our experiments (unless otherwise specified) include the climatology tool.

---

### Meta-Review · Area_Chair_FHdC · 2026-01-07

**Summary:**

This submission introduces Zephyrus, an agentic framework that enables LLMs to interact programmatically with meteorological data, and ZephyrusBench, a challenging benchmark comprising 46 weather-science task templates with 2.2K+ questions. Reviewers broadly agree that the work is well-motivated, technically sound, and clearly written, with particular strengths in the difficulty and scientific realism of the tasks, the integration of domain-specific tools, and rigorous automatic evaluation metrics.

Key concerns across reviews focused on (i) the limited number of task templates, (ii) depth of analysis explaining agent successes/failures, (iii) novelty of agentic gains, (iv) lack of uncertainty estimates, and (v) insufficient baselines and tool ablations. The authors responded convincingly: they regenerated and rebalanced parts of the dataset, added a Climatology tool, reran experiments, reported tool-use and error-correction statistics, conducted extensive ablation studies, separated human vs. synthetic task performance, evaluated open-source and stronger frontier models, and provided variance estimates across multiple runs on a representative subset.

The added ablations clarify the role of individual tools, showing when forecasting or simulation is essential and when extra tools can confuse the agent. Error analysis demonstrates that most code failures are automatically corrected. Variance across runs is small, supporting robustness of reported trends. While gains on the hardest tasks remain limited, this is consistent across models and underscores the benchmark’s difficulty rather than a flaw in evaluation.

Overall, reviewers agree that this is a solid and useful contribution, particularly as an extensible framework and benchmark for weather-science agents, with remaining limitations clearly acknowledged and partially mitigated.

**Reviewer Concerns:**

Concerns substantially addressed:
1/ Lack of tool-use and error-correction analysis (addressed with detailed statistics showing high correction rates).
2/ Absence of ablations (addressed via extensive one-tool-excluded studies clarifying which tools matter for which tasks).
3/ Confusing text-only baseline behavior (resolved by identifying and correcting label imbalance in extreme-weather tasks).
4/ Missing uncertainty estimates (addressed via multi-run variance analysis on a 500-sample subset).
5/ Limited baselines (addressed by adding open-source models and a climatology-only baseline demonstrating large performance drops).
6/ Requests to separate human vs. synthetic task performance (addressed with disaggregated results).

Concerns partially/not fully addressed:
1/ The limited number of task templates remains, though justified by task difficulty and human vetting requirements.
2/ Performance on the hardest tasks remains low across models, with analysis pointing to inherent difficulty rather than a clear single bottleneck (LLM vs. tools vs. agent design).

**Reviewer Scores:**

Reviewer GNmT: Original score: 6,  Likely 7
Rationale: Requests for tool statistics, stronger models, and deeper analysis were directly addressed with new experiments and quantitative evidence.

Reviewer M2k6:
Original score: 8, Likely to remain the same
Rationale: Already positive; requested analyses (synthetic vs. human split, open-source models, agent comparison) were added, reinforcing the original assessment.

Reviewer gtMv:
Original score: 6, Likely 7
Rationale: Key concerns about variance, ablations, and misleading baseline results were resolved through additional experiments and dataset fixes.

Reviewer nucp:
Original score: 8, Likely to remain unchanged
Rationale: Strongly positive initially; additional ablations and climatology-only baselines directly addressed remaining questions.

---

### Decision · Program_Chairs · 2026-01-26

Accept (Poster)